# IMPROVING FAIRNESS VIA FEDERATED LEARNING

## ABSTRACT

Recently, lots of algorithms have been proposed for learning a fair classifier from centralized data. However, how to privately train a fair classifier on decentralized data has not been fully studied yet. In this work, we first propose a new theoretical framework, with which we analyze the value of federated learning in improving fairness. Our analysis reveals that federated learning can strictly boost model fairness compared with all non-federated algorithms. We then theoretically and empirically show that the performance tradeoff of FEDAVG-based fair learning algorithms is strictly worse than that of a fair classifier trained on centralized data. To resolve this, we propose FEDFB, a private fair learning algorithm on decentralized data with a modified FEDAVG protocol. Our extensive experimental results show that FEDFB significantly outperforms existing approaches, sometimes achieving a similar tradeoff as the one trained on centralized data.

## 1 INTRODUCTION

As machine learning is now used to make critical decisions that affect human life, culture, and rights, fair learning has recently received increasing attention. Various fairness notions have been introduced in the past few years (Dwork et al., 2012; Hardt et al., 2016; Zafar et al., 2017b;a; Kearns et al., 2018; Friedler et al., 2016). Among various fairness notions, *group fairness* is the most studied one (Hardt et al., 2016; Zafar et al., 2017a). Group fairness requires the classifier to treat different groups similarly, where groups are defined with respect to sensitive attributes such as gender and race. One of the most commonly used group fairness notions is *demographic parity*, which requires that different groups are equally likely to receive desirable outcomes.

There has been a large amount of work in training fair classifiers (Zafar et al., 2017c; Hardt et al., 2016; Roh et al., 2021), and almost all of these studies assume that the learner has access to the entire training data. Unfortunately, this is *not* the case in many critical applications. To see this, consider a scenario where multiple data owners (*e.g.* courts or financial institutions) have their own private data. Even if they are willing to coordinate with the other institutions to obtain a single model that works well on the combined data, they cannot directly share their data with the others due to the privacy act. This precisely sets the core question we aim to answer in this paper – *how can we privately train a fair classifier on decentralized data?* To answer this, we first study three existing approaches: Unfederated Fair Learning (UFL), Federated Fair Learning via FEDAVG (FFL via FEDAVG), and Centralized Fair Learning (CFL). See Fig. 1 for illustration.

**Unfederated Fair Learning (UFL) and Centralized Fair Learning (CFL)** UFL is the most naïve yet most private approach. As the name indicates, this strategy refers to a scenario where multiple data owners simply decide to *not* coordinate. Instead, each of them learns a fair model on its local data to serve its own users. This approach is completely private as the participating data owners share nothing with the others. However, the overall performance of UFL is expected to be poor, because each data owner may have a highly biased view of the entire data distribution, making their locally trained classifiers fair only on a biased subset of the data, but not on the entire data. To evaluate the performance of this approach, we consider the randomized classifier that makes a prediction using a randomly chosen local classifier. Note that this can be viewed as a random customer model, *i.e.*, a user drawn from the overall data distribution picks and visits one of the institutions, uniformly at random. Another extreme approach is CFL, where a fair model is trained on the pooled data. We expect CFL to achieve the best performance tradeoff, at the cost of no privacy.

**Federated Fair Learning via FEDAVG (FFL via FEDAVG)** FFL via FEDAVG applies *federated learning* (Konečnỳ et al., 2017) together with existing fair learning algorithms. Federated learning is

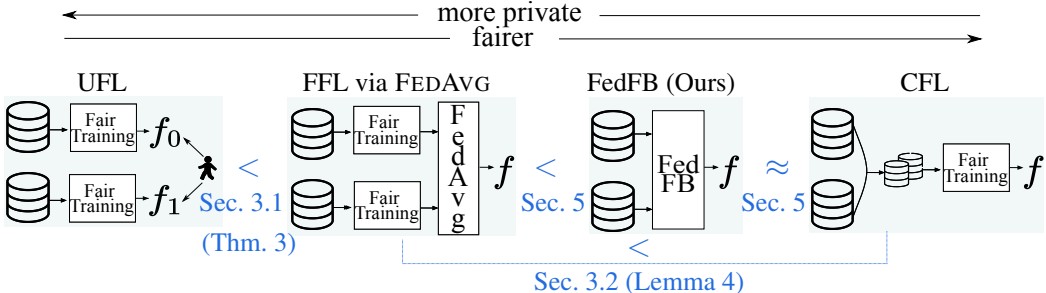

**Figure 1: A high-level illustration of various approaches to fair learning on decentralized data and our contributions.** Assuming two data owners, from left to right, we show UFL (Unfederated Fair Learning), FFL via FEDAVG (Federated Fair Learning via FEDAVG), FEDFB (ours), and CFL (Centralized Fair Learning). In UFL, each data owner trains a locally fair model on its own data, and the customer picks one of them at random. FFL via FEDAVG applies FEDAVG together with off-the-shelf fair training algorithms for local training. Our proposed solution FedFB consists of a modified FEDAVG protocol with a custom-designed fair learning algorithm. CFL is the setting where a fair model is trained on the pooled data. In this work, we theoretically characterize the strict ordering between the existing approaches and empirically demonstrate the superior performance of FedFB.

a distributed learning framework, using which many data owners can collaboratively train a model under the orchestration of a central server while keeping their data decentralized. For instance, under FEDAVG, the standard aggregation protocol for federated learning, the central server periodically computes a weighted average of the locally trained model parameters. If each data owner runs a fair learning algorithm on its own data and these locally trained models are aggregated via FEDAVG, then one might hope to obtain a model that is accurate and fair on the overall data distribution. We call this approach *Federated Fair Learning via* FEDAVG (FFL via FEDAVG).

**Goal and Main Contributions**   The performances of these approaches have not been rigorously analyzed in the literature. In the first place, it has been unknown whether there is any strict performance gap between UFL and FFL via FEDAVG. This makes it unclear whether or not federated learning is necessary at all for decentralized fair learning. The performance comparison between FFL via FEDAVG and CFL also remains unclear. Can FFL via FEDAVG always match the performance of CFL? If not, can we develop a better federated learning approach for decentralized fair learning? Inspired by these open questions, this work rigorously analyzes the performance of the existing approaches and proposes a new solution to decentralized fair learning. Our major contributions can be summarized as follows:

- We develop a theoretical framework for analyzing various approaches for decentralized fair learning. Using this, we prove the strict ordering between the existing approaches, *i.e.*, under some mild conditions, UFL < FFL via FEDAVG < CFL, w.r.t. their fairness-accuracy tradeoffs.
- Improving upon the state-of-the-art algorithm for (centralized) fair learning (Roh et al., 2021), we design FEDFB, a novel approach to learning fair classifiers via federated learning.
- Via extensive experiments, we show that (1) our theoretical findings hold under more general settings, and (2) FEDFB significantly outperforms the existing approaches on various datasets and achieves similar performance as CFL.

To the best of our knowledge, our work is the first theoretical performance comparison of various approaches to fair learning on decentralized data. Moreover, it characterizes the necessity of federated learning for improved fairness-accuracy tradeoff, and we expect this to expedite the adoption of federated learning-based approaches. Our proposed solution FEDFB achieves state-of-the-art performance on many datasets, sometimes achieving a similar tradeoff as the one trained on centralized data.

## 2   RELATED WORK

**Model Fairness** Among various algorithms for fair training (Zemel et al., 2013; Jiang & Nachum, 2020; Zafar et al., 2017c;a; Hardt et al., 2016; Roh et al., 2021; 2020), the current state-of-the-art is FairBatch (Roh et al., 2021), which reweights the samples by solving a bi-level optimization problem, whose inner optimizer is the standard training algorithm and outer optimizer aims to find the best weights attached to groups of samples for the sake of model fairness.

**Federated Learning** Unlike traditional, centralized machine learning approaches, federated learning keeps the data decentralized throughout training, reducing the privacy risks involved in traditional approaches (Konečnÿ et al., 2017; McMahan et al., 2017). FEDAVG (McMahan et al., 2017) is the

first and most widely used federated learning algorithm. The idea is to iteratively compute a weighted average of the local model parameters, with the weights proportional to the local datasets' sizes. Prior work (Li et al., 2020b) has shown that FEDAVG provably converges under some mild conditions. The design of our proposed algorithm FEDFB is also based on that of FEDAVG.

**Federated Fair Learning for Client Parity** There have been only a few attempts in achieving fairness under the federated setting. Moreover, the definition of "fairness" used in the existing federated learning work is slightly different from the standard notion used in the centralized setting. One popular definition of fairness in the federated setting is that all clients (*i.e.* data owners) achieve similar accuracies (or loss values), which we call *client parity*, and several algorithms have been proposed to achieve this goal (Li et al., 2021; 2020a; Mohri et al., 2019; Yue et al., 2021; Zhang et al., 2020a). To compare our methods with existing federated fair learning algorithms designed for client parity, we also extend our FEDFB such that it can also achieve client parity instead of the standard notion of group fairness. In Sec. 5, we will show that FEDFB can achieve as good client parity as the existing algorithms, though FEDFB is not specifically designed for client parity.

**Federated Fair Learning for Group Fairness** A few very recent studies (Ezzeldin et al., 2021; Rodríguez-Gálvez et al., 2021; Chu et al., 2021; Du et al., 2021; Cui et al., 2021), conducted concurrently with our work, also aim at achieving group fairness under federated learning. In particular, Du et al. (2021), Rodríguez-Gálvez et al. (2021) and Chu et al. (2021) mimic the centralized fair learning setting by exchanging information for each local update. In contrast, our FEDFB requires much fewer communication rounds, ensuring higher privacy and lower communication costs. Simlar to FEDFB, Ezzeldin et al. (2021) employs FEDAVG and a reweighting mechanism to

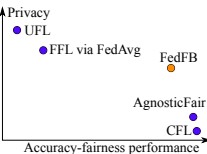

**Figure 2:** Comparison of various fair learning methods..

achieve group fairness. However, FAIRFED only applies to the case with one single binary sensitive attribute, while Rodríguez-Gálvez et al. (2021) and Chu et al. (2021) are not applicable to demographic parity. Therefore, we summarize the comparison of UFL, FFL via FEDAVG, CFL, FEDFB and AGNOSTICFAIR (Du et al., 2021) in terms of performance and privacy in Fig. 2. There is also work that aims at achieving local fairness for each data owner (Cui et al., 2021). This is in contrast to our work, which instead focuses on achieving global fairness in the overall data distribution. Our setting is more appropriate in domains such as criminal justice and social welfare.

## 3 PERFORMANCE ANALYSIS OF UFL, FFL VIA FEDAVG AND CFL

In Sec. 3.1, we first show the necessity of federation by proving that FFL via FEDAVG can achieve strictly higher fairness than UFL. We then prove the limitation of FFL via FEDAVG by comparing its performance with an oracle bound of CFL in Sec. 3.2. These two results together imply that federated learning is necessary, but there exists a limit on what can be achieved by FEDAVG-based approaches. We will present informal theoretical statements, deferring the formal versions and proofs to Sec. A.

**Problem Setting** Denote $[N] := \{0, 1, \ldots, N-1\}$ for any $N \in \mathbb{Z}^+$. We assume $I$ clients, which have the same amount of data. We further assume a simple binary classification setting with a binary sensitive attribute, *i.e.*, $x \in \mathcal{X} = \mathbb{R}$ is the input feature, $y \in \mathcal{Y} = [1]$ is the outcome and $a \in \mathcal{A} = [A] = [1]$ is the binary sensitive attribute. Assume $x$ is a continuous variable. The algorithm we will develop later in Sec. 4 will be applicable to general settings.

We now introduce parameters for describing the data distribution. Let $y \mid x \sim \text{Bern}(\eta(x))$ for all client $i$, where $\eta(\cdot) : \mathcal{X} \to [0, 1]$ is a strictly monotone increasing function. Assume $x \mid a = a, i = i \sim \mathcal{P}_a^{(i)}$, $a \mid i = i \sim \text{Bern}(q_i)$, where $i$ is the index of the client, $\mathcal{P}_a^{(i)}$ is a distribution, and $q_i \in [0, 1]$ for $a = 0, 1, i \in [I]$. Let $\mathcal{F} = \{f : \mathcal{X} \times \mathcal{A} \to [0, 1]\}$. Given $f \in \mathcal{F}$ and data sample $(x, a)$, we consider the following randomized classifier: $\hat{y} \mid x, a \sim \text{Bern}(f(x, a))$.

Using these definitions, we now define demographic parity, specialized for a binary sensitive attribute:

**Definition 1** (Demographic Parity (binary cases)). $\mathbb{P}(\hat{y} = 1 \mid a = 0) = \mathbb{P}(\hat{y} = 1 \mid a = 1)$.

To measure how *unfair* a classifier is with respect to demographic parity (DP), we measure *DP disparity*, *i.e.* the absolute difference between the two positive prediction rates:

$$\text{DP Disp}(f) = |\mathbb{P}(\hat{y} = 1 \mid a = 0) - \mathbb{P}(\hat{y} = 1 \mid a = 1)|.$$

**Figure 3: Fundamental limitation of UFL in terms of fairness range.** We visualize the DP disparity of UFL as a function of local unfairness budgets on three simple Gaussian distributions. The blue horizontal plane is of perfect fairness. The pink horizontal plane visualizes the value of $\delta$ (the lowest DP disparity that UFL can achieve). (a) (Example 1) On a distribution that satisfies the conditions of Corollary 2. (b, c) On distributions that do not satisfy the conditions of Corollary 2. In all three cases, the UFL cannot achieve perfect fairness, *i.e.*, $\delta > 0$.

### 3.1 NECESSITY OF FEDERATION: FFL VIA FEDAVG IS STRICTLY BETTER THAN UFL

**UFL Optimization** We now present the optimization problem solved in the UFL scenario. Here, for analytical tractability, we will assume the population limit, *i.e.*, the true data distribution is used in optimization. Recall that clients do not coordinate in this scenario, and each of them solves their own optimization problem to train a locally fair classifier. In particular, each client $i \in [I]$ first sets its own local fairness constraint $\varepsilon_i \in [0, 1]$, and solves the following constrained optimization problem:

$$\min_{f \in \mathcal{F}} \mathbb{P}(\hat{y} \neq y \mid i = i), \text{ s.t. } |\mathbb{P}(\hat{y} = 1 \mid a = 0, i = i) - \mathbb{P}(\hat{y} = 1 \mid a = 1, i = i)| \leq \varepsilon_i. \quad (\text{UFL}(i, \varepsilon_i))$$

We denote by $f_i^{\varepsilon_i}$ the solution to UFL$(i, \varepsilon_i)$. Recall that the overall performance of UFL is defined as the performance of the following mixture of $I$ classifiers:

$$\hat{y} \mid x, a \sim \text{Bern}(f_i^{\varepsilon_i}(x, a)), \text{ w.p. } 1/I = \text{Bern}\left(\sum_{i \in [I]} f_i^{\varepsilon_i}(x, a)/I\right) = \text{Bern}(f_{\boldsymbol{\varepsilon}}^{\text{UFL}}),$$

where $f_{\boldsymbol{\varepsilon}}^{\text{UFL}} := \sum_{i \in [I]} f_i^{\varepsilon_i}(x, a)/I$, with $\boldsymbol{\varepsilon} = (\varepsilon_0, \ldots, \varepsilon_{I-1})$.

The question here is whether the resulting classifier $f_{\boldsymbol{\varepsilon}}^{\text{UFL}}$ obtained by UFL can achieve an arbitrary level of fairness. The following lemma shows that UFL cannot achieve a high enough fairness level beyond a certain threshold.

**Lemma 1** ((Informal) Achievable fairness range of UFL). *Let $q_i = q \in (0, 1)$ for all $i \in [I]$. Under certain conditions, there exists a certain DP disparity threshold $\delta > 0$ such that $\min_{\boldsymbol{\varepsilon} \in [0,1]^I} DP\ Disp(f_{\boldsymbol{\varepsilon}}^{\text{UFL}}) > \delta$.*

A critical condition of Lemma 1 is that the distribution of insensitive attribute $x \mid a$ is highly heterogeneous on different clients. (See the detailed statements in Sec. A.2.2 and Sec. A.4.) Therefore, Lemma 1 implies that UFL fails to achieve strict fairness requirements even without data heterogeneity on the distribution of sensitive attribute $a$. The following corollary provides an example satisfying the conditions of Lemma 1.

**Corollary 2** (Informal). *Let $I = 2$ and $q_0 = q_1 = 0.5$, $\eta(x) = \frac{1}{1+e^{-x}}$, $\mathcal{P}_a^{(i)} = \mathcal{N}(\mu_a^{(i)}, \sigma^{(i)2})$, where $i = 0, 1$. Under certain assumptions, if one client has much larger variance than the other, there exists $\delta > 0$ such that $\min_{\varepsilon_0, \varepsilon_1 \in [0,1]} DP\ Disp(f_{\boldsymbol{\varepsilon}}^{\text{UFL}}) > \delta$.*

Note that the condition that one client has a much larger variance than the other contributes to the "high data heterogeneity" requirement. We provide the explicit form of $\delta$ in Corollary 9 in Sec. A.2.2. Corollary 2 implies that under a limiting case of Gaussian distribution, UFL cannot achieve high fairness requirements. In Sec. 5.1, we will numerically demonstrate the same claim holds for more general cases. Next, we give a specific example that satisfies the conditions of Corollary 2, which is visualized in Fig. 3(a).

**Example 1.** *Let $\mu_0^{(0)} = \mu_1^{(0)} = 0, \mu_0^{(1)} = 3, \mu_1^{(1)} = -1, \sigma^{(0)} = 70$ and $\sigma^{(1)} = 1$. Then, $\delta \approx 0.21$.*

**FFL via FEDAVG Optimization** FFL via FEDAVG enables federated training of a fair classifier on decentralized data. In Sec. A.3.2, we show that the classifier obtained by FFL via FEDAVG is equivalent to $f_{\boldsymbol{\varepsilon}}^{\text{FFL via FedAvg}}$, the solution to the following constrained optimization problem.

$$\min_{f \in \mathcal{F}} \mathbb{P}(\hat{y} \neq y)$$

$$\text{s.t.} \quad |\mathbb{P}(\hat{y} = 1 \mid a = 0, i = i) - \mathbb{P}(\hat{y} = 1 \mid a = 1, i = i)| \leq \varepsilon_i. \quad (\text{FFL via FEDAVG}(\varepsilon))$$

The following theorem asserts that FFL via FEDAVG can achieve a *strictly* higher fairness than UFL.

**Figure 4: Accuracy-fairness tradeoff curves of CFL, FFL via FEDAVG, and UFL for two clients cases.** Here $q_i$ denotes the proportion of group 1 in client $i \in \{0, 1\}$, so $|q_1 - q_0|$ captures the data heterogeneity of the distribution. The green dotted vertical line describes the lower bound on DP disparity FFL via FEDAVG can achieve, and the orange dotted vertical line describes the lower bound on DP disparity UFL can achieve. As predicted in Thm. 3 and Lemma 4, FFL via FEDAVG's maximum fairness is strictly higher than that of UFL but strictly lower than that of CFL. Moreover, the tradeoff curves are strictly ordered in the same order.

**Theorem 3** ((Informal) Fundamental values of federated learning for fairness)**.** *Let* $q_i = q \in (0, 1)$ *for all* $i \in [I]$. *Under certain conditions,* $\min_{\boldsymbol{\varepsilon} \in [0,1]^I} DP\,Disp(f_{\boldsymbol{\varepsilon}}^{UFL}) > \min_{\boldsymbol{\varepsilon} \in [0,1]^I} DP\,Disp(f_{\boldsymbol{\varepsilon}}^{FFL\ via\ FedAvg}) = 0$.

Similar to Lemma 1, the technical assumptions include high enough heterogeneity of $x \mid a$ across the clients. Thm. 3 shows that even with *just* two clients ($I = 2$), a non-trivial gap exists between non-federated algorithms and federated algorithms in their fairness performances. More details are provided in Sec. A.4. The theorem asserts that, under certain distributional assumptions, by using the optimal local fairness budgets $\varepsilon_i$, FFL via FEDAVG can achieve perfect fairness, while UFL cannot.

**Remark 1.** *Extending Thm. 3 to general cases where* $q_i$ *are not all the same remains open. In particular, the analysis of FFL via* FEDAVG *for those cases remains open. However, we conjecture that our lemma holds for more general cases, and we numerically support our conjecture in Sec. 5.*

**Remark 2.** *There is a stark difference between this phenomenon and the well-known gain of federated learning due to an increased sample size, which is almost negligible with a few number of clients. Our finding on this untapped gain in fairness can better support the need for federated learning even between a small number of clients, which is the case for most cross-silo federated learning scenarios.*

### 3.2 FFL VIA FEDAVG IS STRICTLY WORSE THAN CFL

While we showed that FFL via FEDAVG can achieve perfect fairness on certain distributions, it is still unclear whether or not this is the case for *every* distribution. In this section, we first present the optimization problem for CFL, whose achievable fairness region can serve as an upper bound on that of all other federated learning algorithms. We then show the existence of data distributions on which FFL via FEDAVG achieves a strictly worse fairness performance than CFL. This implies a strict gap between the performance tradeoff of FFL via FEDAVG and that of CFL.

**CFL Optimization** We consider the same problem setting as Sec. 3.1. We now model the CFL scenario as the following constrained optimization problem:

$$\min_{f \in \mathcal{F}} \mathbb{P}(\hat{y} \neq y), \text{ s.t. } |\mathbb{P}(\hat{y} = 1 \mid a = 0) - \mathbb{P}(\hat{y} = 1 \mid a = 1)| \leq \varepsilon. \qquad (\text{CFL}(\varepsilon))$$

Denote the solution to CFL($\varepsilon$) as $f_{\varepsilon}^{\text{CFL}}$. It is clear that $f_{\varepsilon}^{\text{CFL}}$ achieves the best accuracy-fairness tradeoff, at the cost of no privacy. The following lemma shows that there exists some distribution such that FFL via FEDAVG is strictly worse than CFL when the distribution of sensitive attribute $a$ is heterogeneous ($q_i$ are not all the same).

**Lemma 4** ((Informal) A strict gap between FFL via FEDAVG and CFL)**.** *When there exist* $i \neq j \in [I]$ *s.t.* $q_i \neq q_j$, *there exist a distribution such that* $\min_{\boldsymbol{\varepsilon} \in [0,1]^I} DP\,Disp(f_{\boldsymbol{\varepsilon}}^{FFL\ via\ FedAvg}) > \min_{\varepsilon} DP\,Disp(f_{\varepsilon}^{CFL}) = 0$.

**Remark 3.** *A strict gap exists for* certain *distributions, but not for all distributions.*

### 3.3 NUMERICAL COMPARISONS OF ACCURACY-FAIRNESS TRADEOFFS

One limitation of our current theoretical results is that they only compare the maximum achievable fairness. Note that such analysis reveals how the tradeoff of accuracy and fairness behaves as the fairness level increases, but it fails at fully characterizing the entire tradeoff curve. Extending our theoretical results to fully characterize such tradeoffs is highly non-trivial, so we leave it as future

work. Instead, we numerically solve each of the optimization problems and visualize the tradeoff curves achieved by different algorithms.

Shown in Fig. 4 are the tradeoff curves for two clients cases. Let $x \mid a = 0, i = 0 \sim \mathcal{N}(3,1), x \mid a = 1, i = 0 \sim \mathcal{N}(5,1), x \mid a = 0, i = 1 \sim \mathcal{N}(1,1), x \mid a = 1, i = 1 \sim \mathcal{N}(-1,1), a \mid i = 0 \sim \text{Bern}(q_0), a \mid i = 1 \sim \text{Bern}(q_1), \eta(x) = \frac{1}{1+e^{-x}}$, and vary the values of $q_0$ and $q_1$. Note that $|q_1 - q_0|$ captures the heterogeneity of the sensitive data $a$, which increases from left to right. First, one can observe that UFL<FFL via FEDAVG< CFL in terms of the achievable fairness range, as predicted by our theory. Furthermore, we also observe an increasing gap between the tradeoff curves as the data heterogeneity increases. Theoretical understanding of this phenomenon remains open.

## 4 FEDFB FOR IMPROVED FEDERATED FAIR LEARNING

Our findings in the previous section imply that federated learning is necessary, but the current FEDAVG-based approach might not be the best approach. Can we design a federated learning algorithm that is strictly better than FEDAVG-based approaches? In this section, we propose a new federated learning algorithm for fair learning, which we dub FEDFB (short for Federated FairBatch). Our approach is based on the state-of-the-art (centralized) fair learning algorithm FB (short for FairBatch) (Roh et al., 2021) and has a few desirable theoretical guarantees. Later in Sec. 5, we empirically show that FEDFB outperforms FFL via FEDAVG and closely matches the CFL's tradeoff on various datasets.

---

**Algorithm 1:** FEDFB algorithm

---

**ClientUpdate**$(i, \boldsymbol{w}, \boldsymbol{\lambda})$**:**
  Update $\boldsymbol{w}^{(i)}$ according to the sample weights $\boldsymbol{\lambda}$;
  $L_{y,a}^{(i)} \leftarrow \sum_{(i,y,a)=(i,y,a)} \ell(\hat{y}, y; \boldsymbol{w})$, $\forall (y,a)$;
  Send $\boldsymbol{w}^{(i)}, L_{y,a}^{(i)}(\boldsymbol{w})$ for all $(y,a)$ to server via a SecAgg protocol;

**ServerExecutes:**
  **for** *each iteration* **do**
    Clients perform updates;
    $\boldsymbol{w} \leftarrow \text{SecAgg}(\{\boldsymbol{w}^{(i)}\})$;
    $\boldsymbol{L}_{y,a} \leftarrow \text{SecAgg}(\{L_{y,a}^{(i)}\}), \forall (y,a)$;
    $\boldsymbol{\lambda} \leftarrow \texttt{Update}(\boldsymbol{\lambda}, \boldsymbol{L}_{y,a})$;
    Broadcast $\boldsymbol{w}$ and $\boldsymbol{\lambda}$ to clients;
  **output:** $\boldsymbol{w}$

---

We first provide a brief review of the FB algorithm. FB solves a bi-level optimization problem to learn a fair classifier on centralized data. The inner optimization problem solves a weighted empirical risk minimization problem where samples from different groups are reweighted by different weights. The outer optimization problem optimizes the weights used for the inner problem, with the goal of minimizing the unfairness of the classifier. FB works for various group fairness definitions including demographic parity, equalized odds, and equalized opportunity. For the case of demographic parity, the algorithm reduces to the following simple yet intuitive algorithm. The algorithm starts with equal weights for two different groups. After training a model with the initial weights, it computes the sign of the difference between the two positive prediction rates $\mathbb{P}(\hat{y} = 1 \mid a = 0) - \mathbb{P}(\hat{y} = 1 \mid a = 1)$. If this quantity is zero, then DP Disp is zero, so the sample weights are not updated. If this is positive, it decreases the weights for the samples whose $y = 1, a = 0$ and increases the weights for the samples whose $y = 1, a = 1$ so that after retraining, $\mathbb{P}(\hat{y} = 1 \mid a = 0)$ decreases and $\mathbb{P}(\hat{y} = 1 \mid a = 1)$ increases. And vice versa for the other case.

FEDFB is a simple modification of the original FB, which closely simulates the centralized FB applied to the entire data. Recall that under the FEDAVG protocol, clients periodically share their locally trained model parameters with the server. Our modification is based on the following simple observation: *the bi-level structure of* FB *naturally fits the hierarchical structure of federated learning.*

More specifically, note that if the clients also share their group-specific positive prediction rates, then the centralized server can immediately reconstruct the difference between the two positive prediction rates, measured on the entire data distribution. Therefore, the update rules for the outer optimization of the original FB algorithm can be implemented at the central server given the extra information. Then, the central server can broadcast the updated group weights with the clients, which can then locally train models with the newly reweighted samples.

This precisely describes the essence of FEDFB, and shown in Alg. 1 is the pseudocode of the FEDFB framework. Note that the update rule for group weights (denoted by $\boldsymbol{\lambda}$ in the pseudocode) only requires the sum of the group losses, which enables secure aggregation. In Sec. B.1, we present the detailed description of the algorithm, which consists of the local training algorithm with

**Table 1: Comparison of accuracy and DP disparity on the synthetic, Adult, COMPAS, and Bank datasets.** FEDFB significantly outperforms the other approaches on all the tested datasets, sometimes nearly matching the performance of CFL. Note that FFL via FEDAVG sometimes gets a strictly worse performance than FEDAVG. This can be explained by noting that the average of two fair models may not be fair at all.

| | SYNTHETIC | | ADULT | | COMPAS | | BANK | |
| METHOD | ACC.($\uparrow$) | DP DISP.($\downarrow$) | ACC.($\uparrow$) | DP DISP.($\downarrow$) | ACC.($\uparrow$) | DP DISP.($\downarrow$) | ACC.($\uparrow$) | DP DISP.($\downarrow$) |
|---|---|---|---|---|---|---|---|---|
| FEDAVG | .886±.003 | .406±.009 | .829±.012 | .153±.022 | .655±.009 | .167±.037 | .898±.001 | .026±.003 |
| UFL | .727±.194 | .248±.194 | .825±.008 | .034±.028 | .620±.019 | .088±.055 | .892±.002 | .014±.006 |
| FFL VIA FEDAVG | .823±.102 | .305±.131 | .801±.043 | .123±.071 | .595±.005 | .059±.009 | .893±.000 | .017±.001 |
| **FEDFB (OURS)** | .613±.007 | **.011±.009** | .765±.001 | **.001±.001** | .542±.001 | **.001±.001** | .883±.000 | **.000±.000** |
| CFL | .726±.009 | .028±.016 | .816±.010 | .045±.024 | .616±.033 | .036±.028 | .883±.000 | **.000±.000** |

reweighted samples, the model/loss aggregation protocol, the group-weight update algorithm, and the model/group-weight distribution protocol. We highlight a few advantages of FEDFB. First, it provably converges under some mild technical conditions. We proved it by leveraging the analysis tools for federated learning and FB – See Thm. 23 for more details. Second, our algorithm is a strict improvement of FB even in centralized data cases. The original FB algorithm was not applicable if the sensitive attributes are not binary. We made appropriate changes to the algorithm (with theoretical guarantees) so that it can also handle more general cases. Thus, we use our version of FB by default for fair learning in the rest of this paper.

One can note that under FEDFB, clients exchange additional information with the server by communicating real-valued loss values in addition to the model parameters. To limit the information leakage, we also consider a variant of FEDFB, which exchanges the quantized loss values. For instance, "FEDFB(10bits)" means each loss value is uniformly quantized using 10 bits. Such a loss quantization scheme limits the amount of additional information shared between the clients and the server, at the cost of potentially inaccurate group weight updates.

## 5 EXPERIMENTS

In this section, we numerically study the performance of UFL, FFL via FEDAVG, and CFL for more general cases, and evaluate the empirical performance of FEDFB. We investigate the fundamental limitation of UFL under general Gaussian distribution. We compare the accuracy-fairness tradeoff of UFL, FFL via FEDAVG, and CFL by numerically solving UFL($i, \varepsilon_i$), FFL via FEDAVG($\varepsilon$), and CFL($\varepsilon$). More specifically, we first characterize the solutions to these problems up to an unknown scalar, which can be numerically optimized. See Sec. A for more details. Moreover, we evaluate FEDFB on both demographic parity and client parity. In each simulation study, we report the summary statistics across five replications. Similar to the experimental settings used in (Roh et al., 2020), we train all algorithms using a two-layer ReLU neural network with four hidden neurons to evaluate the performance of FEDFB for the non-convex case. The results for logistic regression are provided in Sec. C. We also investigate the empirical relationship between the performance of FEDFB and the number of clients and incorporate differential privacy to further strengthen the power of FEDFB. More implementation details are included in Sec. C.

### 5.1 LIMITATION OF UFL ON GENERAL CASES

The first experiment examines the fairness range of UFL under a more general Gaussian distribution, which does not satisfy the conditions of Corollary 2. For instance, if the variance of two clients is similar, then the conditions do not hold. However, we still conjecture that the same phenomenon holds for more general distributions, and we corroborate our conjecture with numerical experiments. Shown in Fig. 3(b,c) are the numerically computed lower bound on UFL's achievable fairness. In particular, for (b), we let $x \mid a = 0, i = 0 \sim \mathcal{N}(10, 0.2^2), x \mid a = 1, i = 0 \sim \mathcal{N}(9.8, 0.2^2), x \mid a = 0, i = 1 \sim \mathcal{N}(0.2, 0.2^2), x \mid a = 1, i = 1 \sim \mathcal{N}(0, 0.2^2), a \sim \text{Bern}(0.2)$, and for (c), we let $x \mid a = 0, i = 0 \sim \mathcal{N}(3, 1), x \mid a = 1, i = 0 \sim \mathcal{N}(5, 1), x \mid a = 0, i = 1 \sim \mathcal{N}(1, 1), x \mid a = 1, i = 1 \sim \mathcal{N}(-1, 1), a \sim \text{Bern}(0.5)$. For both cases, we set $\eta(x) = \frac{1}{1+e^{-x}}$. It is easy to check that these distributions do not satisfy the conditions of Corollary 2. In particular, the distribution (b) corresponds to the case that the same group is favored on both clients, and the positive rates of each group in different clients are distinctive. The distribution (c) represents the case that different groups are favored on two clients. In both cases, we can see that UFL fails to achieve perfect fairness, *i.e.*, $\delta > 0$. We also observe that $\delta$ is large on the distribution (c), where different groups are favored on two clients. This supports our conjecture that UFL's fairness performance is strictly limited not only on certain data distributions but also on more general ones.

### 5.2 ACCURACY-FAIRNESS TRADEOFFS OF UFL, FFL VIA FEDAVG AND CFL

The second experiment extends the experiments conducted in Sec. 3.3. We assess the relationship between the data heterogeneity and the gap between the three fair learning scenarios with three clients.

**Table 2: Comparison of accuracy and DP disparity on the synthetic dataset with varying heterogeneity.** FEDFB achieves good performance on all the tested levels of heterogeneity. This is because by design, FEDFB closely matches the operation of CFL, whose performance is independent of data heterogeneity.

| METHOD | LOW DATA HETEROGENEITY | | MEDIUM DATA HETEROGENEITY | | HIGH DATA HETEROGENEITY | |
|---|---|---|---|---|---|---|
| | ACC.(↑) | DP DISP.(↓) | ACC.(↑) | DP DISP.(↓) | ACC.(↑) | DP DISP.(↓) |
| **FEDFB (OURS)** | .669±.040 | .058±.042 | .613±.007 | **.011±.009** | .627±.019 | .030±.026 |
| CFL | .726±.009 | **.028±.016** | .726±.009 | .028±.016 | .726±.009 | **.028±.016** |

As shown in Fig. 8, FFL via FEDAVG is observed to achieve a strictly worse tradeoff than CFL and a strictly higher maximum fairness value than UFL. The results corroborate the benefit and limitation of FEDAVG-based federated learning in improving fairness. A very interesting observation is that UFL is observed to obtain a strictly higher accuracy than FFL via FEDAVG. Indeed, this could be attributed to the fact that the average of locally fair models might not be fair to any sub-distribution, while UFL at least ensures that each component of the mixture classifier is fair on some sub-distribution.

### 5.3 FEDFB EVALUATION ON DEMOGRAPHIC PARITY

We assess the empirical performance of FEDFB for both convex and non-convex cases on four datasets and the performance of FEDFB under different data heterogeneity. We focus on demographic parity and report DP disparity $= \max_{a \in [A]} |\mathbb{P}(\hat{y} = 1 \mid \mathrm{a} = a) - \mathbb{P}(\hat{y} = 1)|$, where $A$ is the number of groups. Note that this is slightly different from the definition we used in the previous sections, which was used specifically for the case of one binary sensitive attribute.

**Baselines** We employ three types of baselines: (1) decentralized non-fair training (FEDAVG); (2) decentralized fair training (UFL, FFL via FEDAVG); (3) centralized fair training (AGNOSTICFAIR, CFL). Here, for all the algorithms that involve fair training, we use our improved version of FB, which is the state-of-the-art fair learning algorithm on centralized data. for fairer comparison and better performance, the implementation of UFL, FFL via FEDAVG, and CFL are all based on FB. Note that UFL is absolutely private, CFL violates the privacy policy, FEDAVG, FFL via FEDAVG, and FEDFB share some information at communication rounds without directly sharing the data, and AGNOSTICFAIR exchanges information for each local update. To have a fairer comparison between FFL via FEDAVG and FEDFB, we also equalize their differential privacy guarantees (Dwork, 2008) and compare their performances. See Sec. C for more details. We also report the performance of FAIRFED, a recently proposed algorithm for achieving demographic parity for binary sensitive groups in the federated setting (Ezzeldin et al., 2021), and AGNOSTICFAIR in Sec. C.

**Datasets** **(synthetic)** We follow Roh et al. (2021) for data generation, but with a slight modification to make the dataset more unbalanced. To study the empirical relationship between accuracy, fairness, and data heterogeneity, we split the dataset in different ways to obtain desired levels of data heterogeneity. More details are given in Sec. C.2. **(real)** We use three benchmark datasets: Adult (Dua & Graff, 2017) with 48,842 samples, COMPAS (ProPublica, 2021) with 7,214 samples, and Bank (Moro et al., 2014) with 45,211 samples. We follow Du et al. (2021)'s method to preprocess and split Adult into two clients and Jiang & Nachum (2020)'s method to preprocess COMPAS and Bank. Then, we split COMPAS into two clients based on age and split Bank into three clients based on the loan decision. Note that all the datasets are split in heterogeneous ways.

**Results** We present the results for two-layer ReLU neural networks in Table 1 and leave the results for logistic regression in Sec. C. Table 1 reports the test accuracy and DP disparity of four baselines and FEDFB. We see a substantial fairness improvement obtained by FEDFB. As expected, the resulting fairness level of FEDFB is close to that of CFL and AGNOSTICFAIR. Besides, we observe the poor performance of UFL and FFL via FEDAVG, which is due to the fundamental limitation of UFL and FFL via FEDAVG. Table 2 reports the accuracy and fairness of each method under different data heterogeneity. FedFB is observed to be robust to data heterogeneity. This agrees with our expectation as FEDFB mimics the operation of CFL – which is not affected by data heterogeneity – as

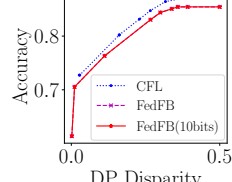

**Figure 5: Accuracy-fairness tradeoff curves on the synthetic dataset.** FEDFB nearly matches the performance of CFL.

much as possible by design. We make a more thorough comparison between CFL and FEDFB by plotting the accuracy-fairness tradeoff curves in Fig. 5. To demonstrate the performance gain does not come at the cost of privacy loss, we restrict FEDFB to only exchange 10 bits of information per communication round. Fig. 5 showcases the benefit of FEDFB in terms of accuracy, fairness and privacy. In Table 7 and Table 8 in Sec. C, we compare FEDFB and FAIRFED. One can observe that FEDFB can achieve a strictly improved fairness than FAIRFED. Also, FEDFB is observed to be more

**Figure 6: Comparison of accuracy and Client Parity (CP) disparity on the synthetic, Adult, COMPAS, and Bank datasets.** Even though our algorithm is not specifically designed for CP, it closely matches the performance of the state-of-the-art fair federated learning algorithms designed for CP.

robust to data heterogeneity. Table 9 in Sec. C shows that FEDFB achieves similar performance as AGNOSTICFAIR, at much lower cost of privacy.

## 5.4 FEDFB EVALUATION ON CLIENT PARITY

We evaluate the performance of FEDFB in achieving client parity (CP) and compare it with the state-of-the-art algorithms for CP. We will measure CP disparity $= \max_{i \neq j \in [I]} |L^{(i)} - L^{(j)}|$. Here $L^{(i)}$ is the loss in $i$th client, see Sec. B.4 for more detail.

**Baselines** We consider GIFAIR (Yue et al., 2021), Q-FFL (Li et al., 2020a), DITTO (Li et al., 2021), and the unconstrained baseline FEDAVG (McMahan et al., 2017). GIFAIR and Q-FFL are the most similar ones to FEDFB. Similar to FEDFB, both GIFAIR and Q-FFL propose a modified aggregation protocol, under which clients share some additional information with the central server, which then accordingly adjust the objective function used for the next round of training. The key difference is that while FEDFB optimizes the coefficients for the primary objective terms (*i.e.* sample reweighting) by solving a bi-level optimization problem, GIFAIR updates the coefficient for the penalty terms, and Q-FFL implicitly updates the weights on each objective term based on nonlinear behaviors of polynomial functions, which is equivalent to the $\alpha$-fairness algorithm used in networking (Mo & Walrand, 2000). The DITTO algorithm combines multitask learning with federated learning to learn a personalized classifier for each client, improving the accuracy of the clients with low accuracy.

**Datasets** We use the same datasets as Sec. 5.3, but split the datasets according to their sensitive attributes to simulate the same setting as assumed by GIFAIR, Q-FFL, and DITTO.

**Results** Fig. 6 shows that FEDFB offers competitive and stable performances in mitigating the model bias, especially in the high fairness region. Although Q-FFL achieves better accuracy and fairness on the synthetic data, under strict fairness constraint, FEDFB and its private variant nearly achieves the highest accuracy on the other three datasets.

## 6 CONCLUSIONS

**Summary** We have investigated how one can achieve group fairness under a decentralized setting. For the first time in the literature, we developed a theoretical framework for decentralized fair learning algorithms and analyzed the performance of UFL, FFL via FEDAVG, and CFL. As a result, we provide novel insights that (1) federated learning can significantly boost model fairness even with only a handful number of participating clients, and (2) FEDAVG-based federated fair learning algorithms are strictly worse than the oracle upper bound of CFL. To close the gap between FEDAVG-based fair learning algorithms and CFL, we propose FEDFB, a new federated fair learning algorithm. The key idea behind FEDFB is that each client shares extra information about the unfairness of its local classifier with the server, which then computes the optimal samples weights that need to be used for the following round of local training. Our extensive experimental results demonstrate that our proposed solution FEDFB achieves state-of-the-art performance, while still ensuring data privacy.

**Open questions** **(Theory)** While we characterized some fundamental limits on tradeoffs of various approaches, there still remains a large number of open questions. First, as we briefly mentioned in Sec. 3.3, full theoretical characterization of accuracy-fairness tradeoff still remains open. Our current theoretical results only study the extreme ends of the tradeoff curves. Moreover, as shown in Sec. 5.2, some of our experimental results reveal a highly nontrivial phenomenon. Studying this phenomenon and identifying the exact relationship between various learning algorithms is an interesting open problem. Furthermore, a three-way tradeoff between accuracy, fairness, and privacy remains widely open. **(Algorithm)** It remains open whether or not our proposed solution FEDFB can be applied for achieving different fairness notions used in the federated setting. In particular, *proportional fairness*, *i.e.*, clients who contribute more should receive more rewards (Zhang et al., 2020b; Lyu et al., 2020b;a), is another popular notion of fairness used in the federated setting, and our current FEDFB cannot handle it. Extending FEDFB to handle proportional fairness is one future research direction.

ETHICS STATEMENT

This work will improve the well-being of individuals in our society by solving ethical issues of AI, such as the implicit discrimination in machine learning algorithms and the privacy of sensitive data. Our theoretical and empirical findings assert that a fair machine learning model can be reliably trained on decentralized data without compromising much privacy.

REPRODUCIBILITY STATEMENT

We have released our implementation in anonymous github[1], which contains the code for all the experiments, including the datasets we use and the implementation of data processing steps. For the theoretical results, we provide all the necessary assumptions and proof in the Appendix.

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

| Symbol | Meaning | Symbol | Meaning | Symbol | Meaning |
|--------|---------|--------|---------|--------|---------|
| x | feature | $\llbracket \cdot \rrbracket$ | indicator function | $\varepsilon_i$ | bias in client $i$ |
| a | sensitive attribute | $f$ | randomized classifier | $q$ | a $\sim$ Bern$(q)$ |
| i | client index | $\mathcal{P}_a^{(i)}$ | distribution | DP Disp$(f)$ | unfairness of $f$ |
| $\hat{y}$ | predicted class | $\eta(x)$ | $\mathbb{P}(y=1 \mid x=x)$ | $f_{\varepsilon_0,\varepsilon_1}^{\text{UFL}}$ | UFL classifier |

**Table 3:** Commonly used notations.

## A APPENDIX - UFL, FFL VIA FEDAVG, CFL ANALYSIS

In this section, we provide the concrete analysis for UFL, FFL via FEDAVG, and CFL. For illustration purposes, we will start by discussing two clients cases, and then extend the analysis into more clients cases in Sec. A.4. We begin with the analysis of CFL in Sec. A.1. Then we analyze UFL and FFL via FEDAVG. To be specific, in Sec. A.2, we analyze the limitation of UFL and present the formal version of Lemma 1 under two clients cases and Corollary 2. In Sec. A.3, we analyze FFL via FEDAVG and compare it with UFL and CFL, then we present the formal two-client version of Thm. 3 and Lemma 4. All the multi-client statements are included in Sec. A.4 We summarize the commonly used notations in Table 3 .

### A.1 CFL ANALYSIS

In this section, we analyze the CFL classifier $f_\varepsilon^{\text{CFL}}$ given in CFL$(\varepsilon)$. We mainly derive the solution of CFL$(\varepsilon)$ in Lemma 5. In Lemma 6 and Lemma 7 we summarize the properties of $f_\varepsilon^{\text{CFL}}$.

**Lemma 5.** *Let* $q \in (0,1)$. *Define* $g(\cdot) : [-\max(q, 1-q), \max(q, 1-q)] \to [-1, 1]$ *as*

$$g(\lambda) = \int_{[\eta^{-1}(\frac{1}{2} - \frac{\lambda}{2(1-q)}), +\infty]} \mathrm{d}\mathcal{P}_0 - \int_{[\eta^{-1}(\frac{1}{2} + \frac{\lambda}{2q}), +\infty]} \mathrm{d}\mathcal{P}_1,$$

*then* $f_\varepsilon^{CFL} = \{\llbracket s(x,a) > 0 \rrbracket + \alpha \llbracket s(x,a) = 0 \rrbracket : \alpha \in [0,1]\}$, *where* $s(x,0) = 2\eta(x) - 1 + \frac{\lambda}{1-q}$, $s(x,1) = 2\eta(x) - 1 - \frac{\lambda}{q}$, $\lambda = g^{-1}(\text{sign}(g(0))\min\{\varepsilon, |g(0)|\})$. *Here we denote the indicator function as* $\llbracket E \rrbracket : \llbracket E \rrbracket = 1$ *if* $E$ *is true, zero otherwise.*

*Proof.* The proof is similar as Menon & Williamson (2018). To solve CFL$(\varepsilon)$, we first write the error rate and the fairness constraint as a linear function of $f$. Let $p_a(\cdot)$ be the pdf of $\mathcal{P}_a$, where $a = 0, 1$. Denote the joint distribution of x and a as $p_{x,a}(x, a)$. Note that

$$\mathbb{P}(\hat{y} \neq y)$$
$$= \int_{\mathcal{X}} \sum_{a \in \mathcal{A}} [f(x,a)(1-\eta(x)) + (1-f(x,a))\eta(x)] \, p_{x,a}(x,a) \, \mathrm{d}x$$
$$= \mathbb{E}_{x,a} f(x,a)(1 - 2\eta(x)) + \mathbb{P}(y = 1)$$

and

$$\mathbb{P}(\hat{y} = 1 \mid a = 0) - \mathbb{P}(\hat{y} = 1 \mid a = 1)$$
$$= \int_{\mathcal{X}} f(x,0) p_0(x) \, \mathrm{d}x - \int_{\mathcal{X}} f(x,1) p_1(x) \, \mathrm{d}x$$
$$= \int_{\mathcal{X}} \sum_{a \in \mathcal{A}} \llbracket a = 0 \rrbracket f(x,0) \frac{p_{x,a}(x,a)}{\mathbb{P}(a=0)} \, \mathrm{d}x - \int_{\mathcal{X}} \sum_{a \in \mathcal{A}} \llbracket a = 1 \rrbracket f(x,1) \frac{p_{x,a}(x,a)}{\mathbb{P}(a=1)} \, \mathrm{d}x$$
$$= \mathbb{E}_{x,a} \left[ f(x,0) \frac{\llbracket a = 0 \rrbracket}{1-q} - f(x,1) \frac{\llbracket a = 1 \rrbracket}{q} \right].$$

Consequently, our goal becomes solving

$$\min_{f \in \mathcal{F}} \mathbb{E}_{x,a} f(x,a)(1 - 2\eta(x)) + \mathbb{P}(y = 1)$$
$$\text{s.t. } \left| \mathbb{E}_{x,a} \left[ f(x,0) \frac{\llbracket a=0 \rrbracket}{1-q} - f(x,1) \frac{\llbracket a=1 \rrbracket}{q} \right] \right| \leq \varepsilon. \tag{1}$$

Denote the function that minimizes the error rate (ERM) as $\tilde{f} \in \mathcal{F}$. It is easy to see that,

$$\tilde{f}(x) \in \{[\![\eta(x) > 1/2]\!] + \alpha [\![\eta(x) = 1/2]\!] : \alpha \in [0, 1]\} .$$

Next, consider the following three cases. In particular, we provide the proof for $|\text{MD}(\tilde{f})| \leq \varepsilon$ and $\text{MD}(\tilde{f}) > \varepsilon$. The proof for $\text{MD}(\tilde{f}) < -\varepsilon$ is similar as the proof for $\text{MD}(\tilde{f}) > \varepsilon$.

**Case 1.** $|MD(\tilde{f})| \leq \varepsilon$: *ERM is already fair.*

The solution to (1) and CFL($\varepsilon$) is $\tilde{f}$.

**Case 2.** $MD(\tilde{f}) > \varepsilon$: *ERM is favoring group 0 over group 1.*

We will show that solving (1) is equivalent to solving an unconstrained optimization problem.

First, we will prove by contradiction that the solution $f^\star \in \mathcal{F}$ to (1) satisfies $\text{MD}(f^\star) = \varepsilon$. We use $f^\star \in \mathcal{F}$ to denote the solution of (1). Suppose the above claim does not hold. Then we have $\text{MD}(f^\star) < \varepsilon$. To show the contradiction, we construct a $f' \in \mathcal{F}$ that satisfies the fairness constraint and has a lower error rate than that of $f^\star$. Let $f'$ be a linear combination of $f^\star$ and $\tilde{f}$:

$$f' = af^\star + (1 - a)\tilde{f},$$

where $a = \frac{\text{MD}(\tilde{f}) - \varepsilon}{\text{MD}(\tilde{f}) - \text{MD}(f^\star)} \in (0, 1)$. Then we obtain

$$\text{MD}(f') = a\text{MD}(f^\star) + (1 - a)\text{MD}(\tilde{f}) = \frac{\varepsilon(\text{MD}(\tilde{f}) - \text{MD}(f^\star))}{\text{MD}(\tilde{f}) - \text{MD}(f^\star)} = \varepsilon.$$

Denote the error rate $\mathbb{P}\{\hat{y} \neq y\}$ as $e : \mathcal{F} \to [0, 1]$. Then the error rate of $f'$ is

$$e(f') = ae(f^\star) + (1 - a)e(\tilde{f}) < e(f^\star),$$

which is inconsistent to the optimality assumption of $f^\star$. Therefore, $\text{MD}(f^\star) = \varepsilon$.

Now, solving CFL($\varepsilon$) is equivalent to solving

$$\min_{f \in \mathcal{F}} \mathbb{E}_{x,a} f(x, a)(1 - 2\eta(x)) + \mathbb{P}(y = 1)$$
$$\text{s.t. } \left| \mathbb{E}_{x,a} \left[ f(x, 0)\frac{[\![a=0]\!]}{1-q} - f(x, 1)\frac{[\![a=1]\!]}{q} \right] \right| = \varepsilon.$$

Furthermore, the optimization problem above is also equivalent to

$$\min_{f \in \mathcal{F}} \quad \mathbb{E}_{x,a} f(x, a)(1 - 2\eta(x)) - \lambda \mathbb{E}_{x,a} \left( f(x, 0)\frac{[\![a = 0]\!]}{1 - q} - f(x, 1)\frac{[\![a = 1]\!]}{q} \right) \qquad (2)$$

$$\text{s.t. } \left| \mathbb{E}_{x,a} \left[ f(x, 0)\frac{[\![a = 0]\!]}{1 - q} - f(x, 1)\frac{[\![a = 1]\!]}{q} \right] \right| = \varepsilon, \qquad (3)$$

for all $\lambda \in \mathbb{R}$.

Next, our goal is to select a suitable $\lambda$ such that the constrained optimization problem above becomes an unconstrained problem, *i.e.*, we will select a suitable $\lambda$ such that the minimizer to the unconstrained optimization problem (2) satisfies equality constraint (3).

Note that

$$\mathbb{E}_{x,a} f(x, a)(1 - 2\eta(x)) - \lambda \mathbb{E}_{x,a} \left( f(x, 0)\frac{[\![a = 0]\!]}{1 - q} - f(x, 1)\frac{[\![a = 1]\!]}{q} \right)$$
$$= \mathbb{E}_{x,a} f(x, a) \left( 1 - 2\eta(x) - \lambda \frac{[\![a = 0]\!]}{1 - q} + \lambda \frac{[\![a = 1]\!]}{q} \right),$$

then the solution to unconstrained optimization problem (2) is

$$\bar{f} \in \{[\![s(x, a) > 0]\!] + \alpha [\![s(x, a) = 0]\!] : \alpha \in [0, 1]\} ,$$

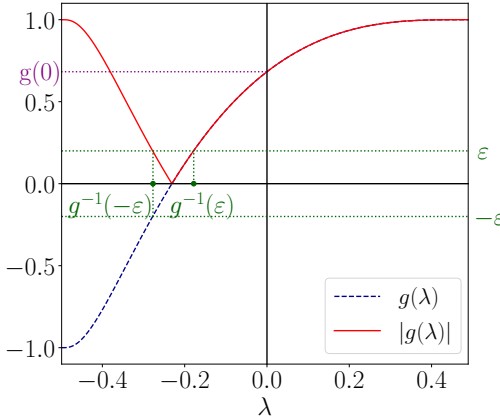

**Figure 7:** Visualization of $g(\lambda)$ and $|g(\lambda)|$. When $g(0) = \text{MD}(\tilde{f}) > \varepsilon \geq 0$, the corresponding $\lambda$ of the best classifier is $\lambda = g^{-1}(\epsilon) < 0$.

where

$$s(x, a) = \begin{cases} -1 + 2\eta(x) + \frac{\lambda}{1-q} & a = 0 \\ -1 + 2\eta(x) - \frac{\lambda}{q} & a = 1 \end{cases}.$$

Since the range of $\eta(x)$ is [0,1], $\bar{f}$ with $\lambda > \max(q, 1-q)$ is no different from $\bar{f}$ with $\lambda = \max(q, 1-q)$; $\bar{f}$ with $\lambda < -\max(q, 1-q)$ is no different from $\bar{f}$ with $\lambda = -\max(q, 1-q)$. Therefore, the only thing left is to find the $\lambda \in [-\max(q, 1-q), \max(q, 1-q)]$ such that $\bar{f}$ satisfies the constraint (3).

Now consider the mean difference between the positive rate of two groups:

$$\text{MD}(\bar{f}) = \mathbb{P}(\hat{\text{y}} = 1 \mid \text{a} = 0) - \mathbb{P}(\hat{\text{y}} = 1 \mid \text{a} = 1)$$

$$= \int_{-\infty}^{+\infty} [\![\eta(x) > \frac{1}{2} - \frac{\lambda}{2(1-q)}]\!] \, d\mathcal{P}_0 - \int_{-\infty}^{+\infty} [\![\eta(x) > \frac{1}{2} + \frac{\lambda}{2q}]\!] \, d\mathcal{P}_1$$

$$= \int_{[\eta^{-1}(\frac{1}{2} - \frac{\lambda}{2(1-q)}), +\infty]} d\mathcal{P}_0 - \int_{[\eta^{-1}(\frac{1}{2} + \frac{\lambda}{2q}), +\infty]} d\mathcal{P}_1$$

$$= g(\lambda).$$

Note that $g(\cdot) : [-\max(q, 1-q), \max(q, 1-q)] \to [-1, 1]$ is a strictly monotone increasing function. Consequently, if and only if $\lambda = g^{-1}(\varepsilon)$, $\bar{f}$ satisfies (3). Recall that optimization problem (2) with constraint (3) is equivalent as CFL($\varepsilon$). Thus, let $\lambda = g^{-1}(\varepsilon)$ and $\bar{f}$ is the solution of CFL($\varepsilon$).

**Case 3.** $MD(\tilde{f}) < -\varepsilon$: *ERM is favoring group 1 over group 0.*

Similarly, like Case 2, we obtain that the solution CFL($\varepsilon$) is

$$\bar{f} \in \{[\![s(x, a) > 0]\!] + \alpha[\![s(x, a) = 0]\!] : \alpha \in [0, 1]\},$$

where

$$s(x, a) = \begin{cases} -1 + 2\eta(x) + \frac{\lambda}{1-q} & a = 0 \\ -1 + 2\eta(x) - \frac{\lambda}{q} & a = 1 \end{cases},$$

and $\lambda = g^{-1}(-\varepsilon)$. Combining all the cases above yields the desired conclusion. The proof is now complete. $\qquad\square$

**Remark 4.** *Select $\alpha = 0$, then the solution to CFL($\varepsilon$) can be written as*

$$f(x, a) = \begin{cases} [\![\eta(x) > \frac{1}{2} - \frac{\lambda}{2(1-q)}]\!] & a = 0 \\ [\![\eta(x) > \frac{1}{2} + \frac{\lambda}{2q}]\!] & a = 1 \end{cases}. \tag{4}$$

*Therefore, Lemma 5 implies that the best classifier of the CFL problem is equivalent to simply applying a constant threshold to the class-probabilities for each value of the sensitive feature.*

Lemma 5 suggests the following property of the solution to CFL($\varepsilon$).

**Lemma 6.** *If $f$ and $g$ are two solutions to CFL($\varepsilon$), then $f = g$ almost everywhere.*

For illustration purposes, we denote

$$\lambda_\varepsilon^{\text{CFL}} = g^{-1}(\text{sign}(g(0)) \min\{\varepsilon, |g(0)|\}). \tag{5}$$

Below we summarize some useful properties of $\lambda_\varepsilon^{\text{CFL}}$.

**Lemma 7.** *The sign of $\lambda_\varepsilon^{CFL}$ and $MD(f_\varepsilon^{CFL})$ are determined by $g(0)$.*

1. *If $\varepsilon < |g(0)|$, then $MD(f_\varepsilon^{CFL}) = \lambda_\varepsilon^{CFL} \neq 0$, and $g(\lambda_\varepsilon^{CFL}) = \text{sign}(g(0))\varepsilon$. If $|g(0)| \leq \varepsilon$, then $\lambda_\varepsilon^{CFL} = 0$ and $MD(f_\varepsilon^{CFL}) = g(\lambda_\varepsilon^{CFL}) = g(0)$.*

2. *If $g(0) > 0$ or $g(0) < 0$, then for any $\varepsilon \geq 0$, we have $\lambda \leq 0$ or $\lambda \geq 0$, respectively.*

*Proof.* The first property follows directly from the definition of $\lambda_\varepsilon^{\text{CFL}}$. Next, we prove the second property.

When $\varepsilon > |g(0)|$, we have $\lambda = 0$ and the first property holds. When $g(0) > \varepsilon > 0$, by the definition of $\lambda_\varepsilon^{\text{CFL}}$, we have $\lambda = g^{-1}(\varepsilon) < g^{-1}(g(0)) = 0$. When $g(0) < -\varepsilon < 0$, we have $\lambda = g^{-1}(-\varepsilon) > g^{-1}(g(0)) = 0$. Combining all the cases above yields the desired conclusion. □

## A.2 UFL ANALYSIS

With the analysis of CFL in Sec. A.1, in this section, we analyze the UFL classifier UFL($i, \varepsilon_i$) for the case of two clients. For illustration purpose, with $I = 2$, we denote $f_{\varepsilon_0, \varepsilon_1}^{\text{UFL}} = f_{\boldsymbol{\varepsilon}}^{\text{UFL}} = (f_0^{\varepsilon_0} + f_1^{\varepsilon_1})/2$. In Sec. A.2.1 we introduce some notations for the UFL classifier $f_{\varepsilon_0, \varepsilon_1}^{\text{UFL}}$ that follows from Lemma 5. In Sec. A.2.2 we analyze the limitation of $f_{\varepsilon_0, \varepsilon_1}^{\text{UFL}}$ as stated in Sec. 3.1. To be more specific, we present the two clients' version of Lemma 1, formal version of Corollary 2 and conclude their proof. In Sec. A.2.3, we analyze the performance gap between $f_{\varepsilon_0, \varepsilon_1}^{\text{UFL}}$ and CFL classifier $f_\varepsilon^{\text{CFL}}$.

### A.2.1 PROBLEM SETTING

By Lemma 5, the solution to UFL($i, \varepsilon_i$) is

$$f_i^{\varepsilon_i}(x, a) = \begin{cases} [\![\eta(x) > \frac{1}{2} - \frac{\lambda_{\varepsilon_i}^{\text{UFL}_i}}{2(1-q)}]\!] & a = 0 \\ [\![\eta(x) > \frac{1}{2} + \frac{\lambda_{\varepsilon_i}^{\text{UFL}_i}}{2q}]\!] & a = 1 \end{cases},$$

where the associated $\lambda_{\varepsilon_i}^{\text{UFL}_i}$ is defined as

$$\lambda_{\varepsilon_i}^{\text{UFL}_i} = g_i^{-1}(\text{sign}(g_i(0)) \min(\varepsilon_i, |g_i(0)|)) \tag{6}$$

and

$$g_i(\lambda) = \int_{[\eta^{-1}(\frac{1}{2} - \frac{\lambda}{2(1-q)}), +\infty)} d\mathcal{P}_0^{(i)} - \int_{[\eta^{-1}(\frac{1}{2} + \frac{\lambda}{2q}), +\infty)} d\mathcal{P}_1^{(i)}.$$

Note that $g_i(\lambda)$ is the mean difference on $i$th client $\mathbb{E}_{\mathbf{x} \sim \mathcal{P}_0^{(i)}} f(\mathbf{x}, 0) - \mathbb{E}_{\mathbf{x} \sim \mathcal{P}_1^{(i)}} f(\mathbf{x}, 1)$ of the classifier of the form (4). Now, the demographic disparity for $f_{\varepsilon_0, \varepsilon_1}^{\text{UFL}}$ can be written as

$$\text{DP Disp}(f_{\varepsilon_0, \varepsilon_1}^{\text{UFL}}) = \left| \frac{1}{2} \sum_{i=0}^{1} [\mathbb{P}(\hat{y} = 1 \mid a = 0, i = i) - \mathbb{P}(\hat{y} = 1 \mid a = 1, i = i)] \right|$$

$$= \left| \frac{1}{4} \sum_{i,j=0}^{1} \left[ \mathbb{E}_{\mathbf{x} \sim \mathcal{P}_0^{(i)}} f_j^{\varepsilon_j}(\mathbf{x}, 0) - \mathbb{E}_{\mathbf{x} \sim \mathcal{P}_1^{(i)}} f_j^{\varepsilon_j}(\mathbf{x}, 1) \right] \right|$$

$$= \left| \frac{1}{4} \left( g_0(\lambda_{\varepsilon_0}^{\text{UFL}_0}) + g_0(\lambda_{\varepsilon_1}^{\text{UFL}_1}) + g_1(\lambda_{\varepsilon_0}^{\text{UFL}_0}) + g_1(\lambda_{\varepsilon_1}^{\text{UFL}_1}) \right) \right|.$$

For ease of notation, we define local mean difference on $i$th client as $\text{MD}_i(f) = \mathbb{P}(\hat{y} = 1 \mid a = 0, i = i) - \mathbb{P}(\hat{y} = 1 \mid a = 1, i = i)$ and local demographic disparity on $i$th client as

DP $\mathrm{Disp}_i(f) = |\mathrm{MD}_i(f)|$, where $i = 0, 1$. Since the overall distribution of the data samples is $\mathrm{x} \mid \mathrm{a} = a \sim \mathcal{P}_a = \mathcal{P}_a^{(0)}/2 + \mathcal{P}_a^{(1)}/2$, $a = 0, 1$, $g$ (see the definition of $g$ in Lemma 5) and $g_0, g_1$ has the following relation:

$$g(\lambda) = \frac{1}{2}g_0(\lambda) + \frac{1}{2}g_1(\lambda).$$

### A.2.2 LIMITATION OF UFL

In this section, we mainly analyze the limitation of UFL in Lemma 8, which shows that $f_{\varepsilon_0,\varepsilon_1}^{\mathrm{UFL}}$ can not achieve 0 demographic disparity in certain cases. Corollary 9 is a specific example of Lemma 8.

**Lemma 8** (Formal version of Lemma 1 under two clients cases). *Let $q \in (0, 1)$. Let $c = \min\{|g_0(0)|, |g_1(0)|\}$. Define $\psi : [0, c] \times [0, c] \to [-1, 1]$ as*

$$\psi(\varepsilon_0, \varepsilon_1) = MD(f_{\varepsilon_0,\varepsilon_1}^{UFL}) = \frac{1}{4}g_0(g_1^{-1}(\mathrm{sign}(g_1(0))\varepsilon_1)) + \frac{1}{4}g_1(g_0^{-1}(\mathrm{sign}(g_0(0))\varepsilon_0))$$
$$+ \frac{1}{4}\mathrm{sign}(g_0(0))\varepsilon_0 + \frac{1}{4}\mathrm{sign}(g_1(0))\varepsilon_1. \tag{7}$$

*If $g_0(0)g_1(0) < 0$ and $\psi(\varepsilon_0, \varepsilon_1)(g_0(0) + g_1(0)) > 0$ for all $\varepsilon_0, \varepsilon_1 \in [0, c]$, then for all $\varepsilon_0, \varepsilon_1 \in [0, 1]$, DP Disp$(f_{\varepsilon_0,\varepsilon_1}^{UFL}) \geq \delta = \min\{|\psi(\varepsilon_0, \varepsilon_1)| : \varepsilon_0, \varepsilon_1 \in [0, c]\} > 0$.*

*Proof.* Define $\delta = \min\{|\psi(\varepsilon_0, \varepsilon_1)| : \varepsilon_0, \varepsilon_1 \in [0, c]\}$. The goal is to show that the demographic disparity has a positive lower bound. Note that the mean difference can be expressed as

$$\mathrm{MD}(f_{\varepsilon_0,\varepsilon_1}^{\mathrm{UFL}}) = \frac{1}{4}\left(g_0(\lambda_{\varepsilon_0}^{\mathrm{UFL}_0}) + g_0(\lambda_{\varepsilon_1}^{\mathrm{UFL}_1}) + g_1(\lambda_{\varepsilon_0}^{\mathrm{UFL}_0}) + g_1(\lambda_{\varepsilon_1}^{\mathrm{UFL}_1})\right). \tag{8}$$

In the following proof, we will show, the mean difference cannot reach 0.

Without any loss of generality, assume $|g_0(0)| < |g_1(0)|$. First we consider $g_1(0) > 0$. We will discuss $g_1(0) < 0$ later. By $g_0(0)g_1(0) < 0$ and $\psi(\varepsilon_0, \varepsilon_1)(g_0(0) + g_1(0)) > 0$ for all $\varepsilon_0, \varepsilon_1 \in [0, c]$, we have $g_0(0) < 0$ and $\psi(\varepsilon_0, \varepsilon_1) > 0$ for all $\varepsilon_0, \varepsilon_1 \in [0, c]$.

First, we will prove that UFL achieves its lowest mean difference when $\varepsilon_0, \varepsilon_1 \in [0, c]$. In what follows, we consider five different cases to derive the desired result.

**Case 1.** $\varepsilon_0 > |g_0(0)|, \varepsilon_1 > |g_1(0)|$: *ERM is fair on both clients.*

By (6), we have $\lambda_{\varepsilon_0}^{\mathrm{UFL}_0} = \lambda_{\varepsilon_1}^{\mathrm{UFL}_1} = 0$. Recall $g_i(\cdot)$ is a monotone increasing function, we combine $g_1(0) > 0$ and Lemma 7 to have $g_0(g_1^{-1}(0)) < g_0(0) < 0$. Applying the above conclusion yields

$$(8) = \frac{1}{2}g_0(0) + \frac{1}{2}g_1(0) > 2\left(\frac{1}{4}g_0(0) + \frac{1}{4}g_1(0) + \frac{1}{4}g_0(g_1^{-1}(0))\right) = 2\psi(g_0(0), 0) \geq \delta.$$

**Case 2.** $\varepsilon_0 \leq |g_0(0)|, \varepsilon_1 > |g_1(0)|$: *ERM is unfair on client 0, but fair on client 1.*

Applying (6) results in $\lambda_{\varepsilon_1}^{\mathrm{UFL}_1} = 0$. By the fact that $g_i(\cdot)$ is a strictly monotone increasing function, we have $\lambda_{\varepsilon_0}^{\mathrm{UFL}_0} = g_0^{-1}(-\varepsilon_0) > g_0^{-1}(g_0(0)) = 0$. Applying the above conclusion yields

$$(8) = -\frac{1}{4}\varepsilon_0 + \frac{1}{4}g_1(0) + \frac{1}{4}g_0(0) + \frac{1}{4}g_1(\lambda_{\varepsilon_0}^{\mathrm{UFL}_0}) \quad (\lambda_{\varepsilon_0}^{\mathrm{UFL}_0} > 0, g_1(\lambda_{\varepsilon_0}^{\mathrm{UFL}_0}) > g_1(0), g_0(0) < -\varepsilon_0)$$
$$> \frac{1}{2}g_0(0) + \frac{1}{2}g_1(0) > 2\psi(g_0(0), 0) \geq \delta.$$

**Case 3.** $\varepsilon_0 \leq |g_0(0)|, \varepsilon_1 \leq |g_1(0)|$: *ERM is unfair on both client 0 and client 1.*

Applying (6) we have $\lambda_{\varepsilon_0}^{\mathrm{UFL}_0} = g_0^{-1}(-\varepsilon_0), \lambda_{\varepsilon_1}^{\mathrm{UFL}_1} = g_1^{-1}(\varepsilon_1)$. Then we have

$$(8) = \frac{1}{4}\left(-\varepsilon_0 + \varepsilon_1 + g_0(g_1^{-1}(\varepsilon_1)) + g_1(g_0^{-1}(-\varepsilon_0))\right)$$
$$\geq \frac{1}{4}\left(-\varepsilon_0 + \varepsilon_0 + g_0(g_1^{-1}(\varepsilon_0)) + g_1(g_0^{-1}(-\varepsilon_0))\right) = \psi(\varepsilon_0, \varepsilon_0) \geq \delta.$$

**Case 4.** $\varepsilon_0 > |g_0(0)|, \varepsilon_1 \le |g_0(0)|$: *ERM is fair on client 0 and very unfair on client 1.*

By (6), we have $\lambda_{\varepsilon_0}^{\mathrm{UFL}_0} = 0, \lambda_{\varepsilon_1}^{\mathrm{UFL}_1} = g_1^{-1}(\varepsilon_1) > g_1^{-1}(0)$. Then we obtain

$$(8) = \frac{1}{4}\left(g_0(0) + g_0(\lambda_{\varepsilon_1}^{\mathrm{UFL}_1}) + g_1(0) + \varepsilon_1\right)$$
$$> (g_0(0) + g_0(g_1^{-1}(0)) + g_1(0))/4 = \psi(g_0(0), 0) \ge \delta.$$

**Case 5.** $\varepsilon_0 > |g_0(0)|, |g_0(0)| \le \varepsilon_1 < |g_1(0)|$: *ERM is fair on client 0 and unfair on client 1.*

Applying (6) implies $\lambda_{\varepsilon_1}^{\mathrm{UFL}_1} = g_1^{-1}(\varepsilon_1) > g_1^{-1}(0)$. Therefore,

$$(8) = \frac{1}{4}\left(g_0(0) + g_0(g_1^{-1}(\varepsilon_1)) + \varepsilon_1 + g_1(0)\right)$$
$$> (g_0(0) + g_1(0) + g_0(g_1^{-1}(0)) + \varepsilon_1)/4 > \psi(g_0(0), 0) \ge \delta.$$

Combining all the cases above, we conclude that when $g_1(0) > 0$ DP Disp$(f_{\varepsilon_0,\varepsilon_1}^{\mathrm{UFL}}) \ge \delta = \min\{|\psi(\varepsilon_0,\varepsilon_1)| : \varepsilon_0, \varepsilon_1 \in [0, c]\} > 0$ for all $\varepsilon_0, \varepsilon_1 \in [0, 1]$.

Now we consider $g_1(0) < 0$, by the setting $|g_0(0)| < |g_1(0)|$ and the assumption $g_0(0)g_1(0) < 0$ and $\psi(\varepsilon_0,\varepsilon_1)(g_0(0) + g_1(0)) > 0$, we have $g_0(0) > 0$ and $\psi(\varepsilon_0,\varepsilon_1) < 0$ for all $\varepsilon_0, \varepsilon_1 \in [0, c]$. Following similar computation above, Case 1 - Case 5 become:

Case 1. $\varepsilon_0 > |g_0(0)|, \varepsilon_1 > |g_1(0)|$. Now we have $0 < g_0(0) < g_0(g_1^{-1}(0))$, thus

$$(8) < 2\left(\frac{1}{4}g_0(0) + \frac{1}{4}g_1(0) + \frac{1}{4}g_0(g_1^{-1}(0))\right) = 2\psi(g_0(0), 0) \le -\delta.$$

Case 2. $\varepsilon_0 \le |g_0(0)|, \varepsilon_1 > |g_1(0)|$. Now we have $g_0(0) > \varepsilon_0, g_1(\lambda_{\varepsilon_0}^{\mathrm{UFL}_0}) < g_1(0)$, thus

$$(8) < \frac{1}{2}g_0(0) + \frac{1}{2}g_1(0) < 2\psi(g_0(0), 0) \le -\delta.$$

Case 3. In this case we have

$$(8) = \frac{1}{4}\left(\varepsilon_0 - \varepsilon_1 + g_0(g_1^{-1}(-\varepsilon_1)) + g_1(g_0^{-1}(\varepsilon_0))\right) = \psi(\varepsilon_0, \varepsilon_1) \le -\delta.$$

Case 4. Now we have $\lambda_{\varepsilon_1}^{\mathrm{UFL}_1} = g_1^{-1}(-\varepsilon_1) < g_1^{-1}(0)$, thus

$$(8) < (g_0(0) + g_0(g_1^{-1}(0)) + g_1(0))/4 = \psi(g_0(0), 0) \le -\delta.$$

Case 5. Now we have $\lambda_{\varepsilon_1}^{\mathrm{UFL}_1} = g_1^{-1}(-\varepsilon_1) < g_1^{-1}(0)$, thus

$$(8) = (g_0(0) + g_1(0) + g_0(g_1^{-1}(0)) - \varepsilon_1)/4 < \psi(g_0(0), 0) \le -\delta.$$

Then we conclude the proof.

$\square$

**Remark 5.** *Note that $c$ is the smallest local demographic disparity the ERM achieves on clients. The condition $g_0(0)g_1(0) < 0$ implies that the ERM is favoring different groups in different clients. The condition $\psi(\varepsilon_0,\varepsilon_1)(g_0(0) + g_1(0)) > 0$ for all $\varepsilon_0, \varepsilon_1 \in [0, c]$ implies that $f_{\varepsilon_0,\varepsilon_1}^{UFL}$ favors the same group as ERM when the constraint is very tight. If the conditions above hold, Lemma 8 suggests that there exists a lower bound of all the demographic disparity that UFL can achieve. In particular, if the conditions above hold, UFL fails to achieve perfect demographic parity.*

Among the conditions of Lemma 8, $g_0(0)g_1(0) < 0$ can be satisfied by the distribution with high data heterogeneity. To demonstrate the condition $\psi(\varepsilon_0,\varepsilon_1)(g_0(0) + g_1(0)) > 0$ for all $\varepsilon_0, \varepsilon_1 \in [0, c]$ can be satisfied, we consider a limiting Gaussian case. The following corollary serves as an example that satisfies the conditions of Lemma 8, and provides a more explicit expression of the lowest demographic disparity UFL can reach in the Gaussian case.

**Corollary 9** (Formal version of Corollary 2). *Let $q = 0.5$, $\eta(x) = \frac{1}{1+e^{-x}}$, $\mathcal{P}_a^{(i)} = \mathcal{N}(\mu_a^{(i)}, \sigma^{(i)2})$ where $(\mu_0^{(0)} - \mu_1^{(0)})(\mu_0^{(1)} - \mu_1^{(1)}) < 0$. Then $DP\ Disp(f_{\varepsilon_0,\varepsilon_1}^{UFL}) \geq \delta \approx \frac{1}{4}|g_0(0) + g_1(0)| > 0$ for all $\varepsilon_0, \varepsilon_1 \in [0, 1]$ if one of the following condition holds:*

1. *$\sigma^{(0)} \gg \sigma^{(1)}, |\mu_0^{(0)}|, |\mu_1^{(0)}|, |\mu_0^{(1)}|, |\mu_1^{(1)}|$ and $\mu_0^{(1)} > \mu_1^{(1)}$: client 0 has much larger variance than client 1, and client 1 is favoring group 0;*

2. *$\sigma^{(1)} \gg \sigma^{(0)}, |\mu_0^{(0)}|, |\mu_1^{(0)}|, |\mu_0^{(1)}|, |\mu_1^{(1)}|$ and $\mu_0^{(0)} > \mu_1^{(0)}$: client 1 has much larger variance than client 0, and client 0 is favoring group 0.*

*Proof.* In this example, note that local mean difference function of $\lambda$ can be written as:

$$g_i(\lambda) = \Phi\left(\frac{\eta^{-1}(\frac{1}{2} + \lambda) - \mu_1^{(i)}}{\sigma^{(i)}}\right) - \Phi\left(\frac{\eta^{-1}(\frac{1}{2} - \lambda) - \mu_0^{(i)}}{\sigma^{(i)}}\right), \tag{9}$$

where $\Phi(\cdot)$ is the CDF of the standard Gaussian distribution.

We only provide the proof for condition 1, and the proof for condition 2 is similar. Assume condition 1 holds. By $(\mu_0^{(0)} - \mu_1^{(0)})(\mu_0^{(1)} - \mu_1^{(1)}) < 0$ and $\mu_0^{(1)} - \mu_1^{(1)} > 0$, we have $\mu_0^{(0)} - \mu_1^{(0)} < 0$. By (9), we have $g_0(0) < 0$ and $g_1(0) > 0$. Consequently, combining Lemma 7 and (7) yields

$$\psi(\varepsilon_0, \varepsilon_1) = \frac{1}{4}\left(g_0(g_1^{-1}(\varepsilon_1)) + g_1(g_0^{-1}(-\varepsilon_0)) - \varepsilon_0 + \varepsilon_1\right).$$

First we show that $\psi(\varepsilon_0, \varepsilon_1)$ reachs its minimum either at $(c, 0)$ or $(0, c)$ by taking the derivative of $\psi$, where $c = \min\{|g_0(0)|, |g_1(0)|\}$ is the smallest local demographic disparity. And then, we will estimate the minimum of $\psi$ on $[0, 1] \times [0, 1]$.

We take the derivative of $\psi$ with respect to $\varepsilon_i$ and get

$$\frac{\partial\psi}{\partial\varepsilon_i}(\varepsilon_0, \varepsilon_1) = \text{sign}(g_i(0))\left(1 + \frac{g_{1-i}'(g_i^{-1}(\text{sign}(g_i(0))\varepsilon_i))}{g_i'(g_i^{-1}(\text{sign}(g_i(0))\varepsilon_i))}\right)/4.$$

By condition 1, we have $g_0(0) = \Phi(-\frac{\mu_1^{(0)}}{\sigma^{(0)}}) - \Phi(-\frac{\mu_0^{(0)}}{\sigma^{(0)}}) \approx 0$, thus $|g_0(0)| \ll |g_1(0)|$ and $c = |g_0(0)|$. Since $g_i$ are increasing function, $i = 0, 1$, we have $g_i'(\cdot) > 0$, and thus $\frac{\partial\psi}{\partial\varepsilon_0} < 0, \frac{\partial\psi}{\partial\varepsilon_1} > 0$. Therefore, $\psi$ reaches its extreme value at $(0, c)$ and $(c, 0)$.

Now, we evaluate

$$\psi(0, c) = (g_0(g_1^{-1}(c)) + g_1(g_0^{-1}(0)) - g_0(0))/4 > \frac{g_1(0) - g_0(0) + g_0(g_1^{-1}(c))}{4},$$

where the inequality comes from $g_0(0) < 0$ to have $g_1(g_0^{-1}(0)) > g_1(0)$. And at $(c, 0)$, we have

$$\psi(c, 0) = (g_0(g_1^{-1}(0)) + g_1(0) + g_0(0))/4.$$

In what follows, we will show that $\min\{\psi(c, 0), \psi(0, c)\} \approx \delta = \frac{g_1(0) + g_0(0)}{4}$ by proving that $0 > g_0(g_1^{-1}(c)) > g_0(g_1^{-1}(0)) \approx 0$.

Consider $\psi(c, 0)$ and $\psi(0, c)$. Since $g_1(0) > c, g_0(0) < 0$, we have

$$g_0(g_1^{-1}(0)) < g_0(g_1^{-1}(c)) < g_0(0) < 0.$$

Therefore, the only thing left is to show $g_0(g_1^{-1}(0)) \approx 0$. We divide the rest of the proof into the following three cases.

**Case 1.** $\mu_1^{(1)} < \mu_0^{(1)} < 0$: *on client 1, the local classifier is favoring group 0 over group 1, and the positive rate of both groups are under $\frac{1}{2}$.*

Clearly, under this case, we have $\int_{[\eta^{-1}(\frac{1}{2}),\infty)} d\mathcal{P}_0^{(1)} = \int_{[0,\infty)} d\mathcal{P}_0^{(1)} < \frac{1}{2}$. We select $\lambda' < 0$ such that $\eta^{-1}(\frac{1}{2} + \lambda') = \mu_1^{(1)}$. Then we have $\int_{[\eta^{-1}(\frac{1}{2}+\lambda'),\infty)} d\mathcal{P}_1^{(1)} = \int_{[\mu_1^{(1)},\infty)} d\mathcal{P}_1^{(1)} = \frac{1}{2}$, while $\int_{[\eta^{-1}(\frac{1}{2} - \frac{\lambda'}{2(1-q)}),\infty)} d\mathcal{P}_0^{(1)} < 0$. Thus we get $g_1(\lambda') < 0$. Combining $g_1(0) > 0$ and intermediate value theorem results in $\lambda' < g_1^{-1}(0) < 0$. Then we obtain

$$
\begin{aligned}
\mu_1^{(1)} &= \eta^{-1}(\frac{1}{2} + \lambda') < \eta^{-1}(\frac{1}{2} + g_1^{-1}(0)) < 0 \\
-\mu_1^{(1)} &= \eta^{-1}(\frac{1}{2} - \lambda') > \eta^{-1}(\frac{1}{2} - g_1^{-1}(0)) > 0 \quad (\eta(x) - \frac{1}{2} \text{ is odd})
\end{aligned}
\tag{10}
$$

By plugging (10) into (9), we have the other side of $g_0(g_1^{-1}(0)) < 0$ is bounded by $g_0(\lambda') = \Phi(\frac{\mu_1^{(1)} - \mu_1^{(1)}}{\sigma^{(1)}}) - \Phi(\frac{-\mu_1^{(1)} - \mu_0^{(1)}}{\sigma^{(1)}})$. Since $\sigma^{(1)} \gg |\mu_1^{(1)}|, |\mu_1^{(0)}|$, we get $g_0(g_1^{-1}(0)) \approx 0$.

**Case 2.** $0 < \mu_1^{(1)} < \mu_0^{(1)}$: *on client 1, the local classifier is favoring group 0 over group 1, and the positive rate of both groups are above $\frac{1}{2}$.*

This proof of this case is similar to Case 1.

**Case 3.** $\mu_1^{(1)} < 0 < \mu_0^{(1)}$: *with respect to the local classifier trained by client 1, the positive rate of group 0 is above $\frac{1}{2}$ while that of group 1 is under $\frac{1}{2}$.*

Without any loss of generality, we assume $|\mu_1^{(1)}| < |\mu_0^{(1)}|$. Select $\lambda'' < 0$ such that $\eta(\frac{1}{2} - \lambda'') = \mu_0^{(1)}$. Clearly $\int_{[\eta^{-1}(\frac{1}{2}-\lambda''),+\infty)} d\mathcal{P}_0^{(1)} = \int_{[\mu_0^{(1)},+\infty)} d\mathcal{P}_0^{(1)} = \frac{1}{2}$, while

$$
\int_{[\eta^{-1}(\frac{1}{2}+\lambda''),+\infty)} d\mathcal{P}_1^{(1)} = \int_{[-\mu_0^{(1)},+\infty)} d\mathcal{P}_1^{(1)} > \int_{[-\mu_1^{(1)},+\infty)} d\mathcal{P}_1^{(1)} > \frac{1}{2}.
$$

Consequently, we get $g_1(\lambda'') < 0$ and $\lambda'' < g_1^{-1}(0) < 0$. Then we draw the same conclusion as (10). Therefore, $g_0(g_1^{-1}(0)) \approx 0$.

Combining all three cases above, we get $\delta > 0$. Then applying Lemma 8 we complete the proof. $\quad\square$

### A.2.3 COMPARISON BETWEEN UFL AND CFL

In this section, we compare the performance of UFL and CFL. In Lemma 10 and Lemma 11, we illustrate the conditions for UFL to have the same performance as CFL. In Lemma 12, Lemma 13 and Lemma 14, we illustrate the scenarios when CFL outperforms UFL.

To do the comparison, first we introduce some additional notations. Define the accuracy of a classifier $f$ as

$$
\text{Acc}(f) = \mathbb{P}(\hat{y} = y) = \mathbb{P}(y = 0)\mathbb{E}_{x,a|y=0}[1 - f(x,a)] + \mathbb{P}(y = 1)\mathbb{E}_{x,a|y=0}f(x,a).
$$

Given the required global demographic disparity $\varepsilon$, define the performance of $f_{\varepsilon_0,\varepsilon_1}^{\text{UFL}}$ as:

$$
\text{UFL}(\varepsilon_0, \varepsilon_1; \varepsilon) = \begin{cases} \text{Acc}(f_{\varepsilon_0,\varepsilon_1}^{\text{UFL}}) & \text{DP Disp}(f_{\varepsilon_0,\varepsilon_1}^{\text{UFL}}) \le \varepsilon \\ 0 & o.w. \end{cases},
$$

and define performance of $f_\varepsilon^{\text{CFL}}$ as:

$$
\text{CFL}(\varepsilon) = \text{Acc}(f_\varepsilon^{\text{CFL}}).
$$

Now we are able to compare the performance between UFL and CFL with the metric $\text{UFL}(\varepsilon_0, \varepsilon_1; \varepsilon)$ and $\text{CFL}(\varepsilon)$. In particular, we will show that, under some mild conditions, $\max_{\varepsilon_0,\varepsilon_1} \text{UFL}(\varepsilon_0, \varepsilon_1; \varepsilon) < \text{CFL}(\varepsilon)$, which implies the gap between UFL and CFL is inevitable.

We begin with the following two lemmas, which describe the cases that $\text{UFL}(\varepsilon_0, \varepsilon_1; \varepsilon) = \text{CFL}(\varepsilon)$.

**Lemma 10.** *Let $q \in (0,1)$. Given an UFL classifier $f_{\varepsilon_0,\varepsilon_1}^{UFL}$ such that DP Disp($f_{\varepsilon_0,\varepsilon_1}^{UFL}$) $\leq \varepsilon$ and a CFL classifier $f_{\varepsilon}^{CFL}$, we have $UFL(\varepsilon_0,\varepsilon_1;\varepsilon) \leq CFL(\varepsilon)$. The equality holds if and only if $\lambda_{\varepsilon_0}^{UFL_0} = \lambda_{\varepsilon_1}^{UFL_1} = \lambda_{\varepsilon}^{CFL}$, where $\lambda_{\varepsilon}^{CFL}$ is defined in (5), $\lambda_{\varepsilon_0}^{UFL_0}, \lambda_{\varepsilon_1}^{UFL_1}$ are defined in (6).*

*Proof.* The proof is straightforward. Clearly, since $f_{\varepsilon}^{\text{CFL}}$ is the optimizer to CFL($\varepsilon$), we have UFL($\varepsilon_0,\varepsilon_1;\varepsilon$) $\leq$ CFL($\varepsilon$). By Lemma 5, according to the form of the solution to CFL($\varepsilon$), $f_{\varepsilon_0,\varepsilon_1}^{\text{UFL}}$ is the solution to CFL($\varepsilon$) if and only if $\lambda_{\varepsilon_0}^{\text{UFL}_0} = \lambda_{\varepsilon_1}^{\text{UFL}_1} = \lambda_{\varepsilon}^{\text{CFL}}$. Thus complete the proof.

$\square$

**Lemma 11.** *Let $q \in (0,1)$. If the ERM is already fair, i.e., $\varepsilon \geq |g(0)|$, then*

$$\max_{\varepsilon_0,\varepsilon_1} UFL(\varepsilon_0,\varepsilon_1;\varepsilon) = CFL(\varepsilon).$$

*Proof.* Since the ERM is already fair, $f_{\varepsilon}^{\text{CFL}}$ is ERM= $[\![\eta(x) > 1/2]\!]$. Therefore, we take $\varepsilon_0 = \varepsilon_1 = 1$, and $f_{\varepsilon_0,\varepsilon_1}^{\text{UFL}}$ also equals to ERM. Thus we conclude the lemma.

$\square$

The next two lemmas describes the cases that UFL($\varepsilon_0,\varepsilon_1;\varepsilon$) < CFL($\varepsilon$).

**Lemma 12.** *Let $q \in (0,1)$. If $g_0(0)g_1(0) < 0$, $\max_{\varepsilon_0,\varepsilon_1} UFL(\varepsilon_0,\varepsilon_1;\varepsilon) < CFL(\varepsilon)$ for all $\varepsilon < |g(0)|$.*

*Proof.* In this proof, we only consider the case that $g_1(0) > 0$, $|g_1(0)| \geq |g_0(0)|$. The proof for $g_1(0) > 0$ or $|g_1(0)| \geq |g_0(0)|$ is similar. Next, we divide the proof into two cases.

**Case 1.** $\max_{\varepsilon_0,\varepsilon_1} UFL(\varepsilon_0,\varepsilon_1;\varepsilon) = 0$: *UFL cannot achieve $\varepsilon$ global demographic disparity.*

The conclusion holds.

**Case 2.** $\max_{\varepsilon_0,\varepsilon_1} UFL(\varepsilon_0,\varepsilon_1;\varepsilon) > 0$: *UFL can achieve $\varepsilon$ global demographic disparity.*

Since $\varepsilon < |g(0)|$, by Lemma 7, we have $\lambda_{\varepsilon}^{\text{CFL}} \neq 0$. Next, we solve $f_{\varepsilon_0,\varepsilon_1}^{\text{UFL}}$ by solving the local version of CFL($\varepsilon$). Combining Lemma 7, $g_1(0) > 0$ and $g_0(0) < 0$ yields $\lambda_{\varepsilon_0}^{\text{UFL}_0} \geq 0, \lambda_{\varepsilon_1}^{\text{UFL}_1} \leq 0$. If $\lambda_{\varepsilon_0}^{\text{UFL}_0} = \lambda_{\varepsilon_1}^{\text{UFL}_1}$, then $\lambda_{\varepsilon_0}^{\text{UFL}_0} = \lambda_{\varepsilon_1}^{\text{UFL}_1} = 0 \neq \lambda_{\varepsilon}^{\text{CFL}}$. Thus, we conclude the lemma by applying Lemma 10. $\square$

**Remark 6.** *Lemma 12 implies that if ERM is favoring different groups in different clients, there exists an inevitable gap between the performance of UFL and that of CFL.*

**Lemma 13.** *Let $q \in (0,1)$. Let $\tau = \min\left\{|g(0)|, \max\{\text{sign}(g(0))g(g_0^{-1}(0)), \text{sign}(g(0))g(g_1^{-1}(0))\}\right\}$.*

*If $g_0(0)g_1(0) > 0$, we have*

$$\max_{\varepsilon_0,\varepsilon_1} UFL(\varepsilon_0,\varepsilon_1;\varepsilon) \begin{cases} = CFL(\varepsilon) & \text{for all } \varepsilon \geq \tau \\ < CFL(\varepsilon) & \text{o.w.} \end{cases}.$$

*Proof.* Without any loss of generality, assume $g_0(0), g_1(0) > 0$. Then by Lemma 7, we have $\lambda_{\varepsilon_0}^{\text{UFL}_0} \leq 0, \lambda_{\varepsilon_1}^{\text{UFL}_1} \leq 0$ for all $\varepsilon_0, \varepsilon_1 \in [0,1]$, and $g(0) = (g_0(0) + g_1(0))/2 > 0$.

To study the performance of UFL when $g_0(0)g_1(0) < 0$, recall that we use $f_i^{\varepsilon_i}$ to denote the local classifier trained by client $i$ in UFL analysis. Therefore, $f_i^0$ is the local classifier trained by client $i$ that achieves perfect local fairness.

Next, we discuss two cases to prove the result. In Case 1, we will show that $\tau = 0$, and then prove that $\max_{\varepsilon_0,\varepsilon_1} \text{UFL}(\varepsilon_0,\varepsilon_1;\varepsilon) = \text{CFL}(\varepsilon)$; in Case 2, we will show that $\tau > 0$, and then prove that $\max_{\varepsilon_0,\varepsilon_1} \text{UFL}(\varepsilon_0,\varepsilon_1;\varepsilon) < \text{CFL}(\varepsilon)$ when $\varepsilon < \tau$.

**Case 1.** $f_0^0 = f_1^0$: *the two local classifiers that achieve perfect local fairness are equal.*

When $\varepsilon \geq g(0)$, the conclusion holds by directly applying Lemma 11. Therefore, in what follows, we focus on the case that $\varepsilon < g(0)$.

Next, we will first, show that when $\varepsilon = 0$, $\lambda_0^{\mathrm{UFL}_0} = \lambda_0^{\mathrm{UFL}_1} = \lambda_0^{\mathrm{CFL}}$, which implies that $f_0^0 = f_1^0 = f_0^{\mathrm{CFL}}$.

Since $f_0^0 = f_1^0$, we have $\lambda_0^{\mathrm{UFL}_0} = \lambda_0^{\mathrm{UFL}_1}$, and thus $g_0^{-1}(0) = g_1^{-1}(0)$. Consequently, we get

$$g(\lambda_0^{\mathrm{UFL}_0}) = g(\lambda_0^{\mathrm{UFL}_1}) = g(g_0^{-1}(0)) = \frac{g_0(g_0^{-1}(0)) + g_1(g_1^{-1}(0))}{2} = 0 = g(\lambda_0^{\mathrm{CFL}}).$$

By the monotonicity of $g$, we have $\lambda_0^{\mathrm{UFL}_0} = \lambda_0^{\mathrm{UFL}_1} = \lambda_0^{\mathrm{CFL}}$ and $\tau = 0$. Next, we will show that, for $\varepsilon \neq 0$, there also exists $\varepsilon_0, \varepsilon_1 \in [0, 1]$ such that $\lambda_{\varepsilon_0}^{\mathrm{UFL}_0} = \lambda_{\varepsilon_1}^{\mathrm{UFL}_1} = \lambda_\varepsilon^{\mathrm{CFL}}$, which implies $f_{\varepsilon_0}^{\mathrm{UFL}_0} = f_{\varepsilon_0}^{\mathrm{UFL}_1} = f_\varepsilon^{\mathrm{CFL}}$.

Consider $\varepsilon \neq 0$. Select $\varepsilon_i = g_i(\lambda_\varepsilon^{\mathrm{CFL}}), i = 0, 1$. By Lemma 5 and the monotonicity of $g$, we have $\lambda_0^{\mathrm{CFL}} < \lambda_\varepsilon^{\mathrm{CFL}} < 0$. Therefore, $\varepsilon_i = g_i(\lambda_\varepsilon^{\mathrm{CFL}}) > g_i(\lambda_0^{\mathrm{CFL}}) = 0$. By Lemma 5, we get $\lambda_{\varepsilon_i}^{\mathrm{UFL}_i} = g_i^{-1}(\varepsilon_i)$ for $i = 0, 1$. By the selection of $\varepsilon_i$, we have $\lambda_\varepsilon^{\mathrm{CFL}} = g_i^{-1}(\varepsilon_i)$. Therefore, $\lambda_\varepsilon^{\mathrm{CFL}} = \lambda_{\varepsilon_0}^{\mathrm{UFL}_0} = \lambda_{\varepsilon_1}^{\mathrm{UFL}_1}$ when $\varepsilon \neq 0$.

Combining all the discussion above yields $f_{\varepsilon_0,\varepsilon_1}^{\mathrm{UFL}} = f_\varepsilon^{\mathrm{CFL}}$ for all $\varepsilon < g(0)$, thus we conclude $\max_{\varepsilon_0,\varepsilon_1} \mathrm{UFL}(\varepsilon_0, \varepsilon_1; \varepsilon) = \mathrm{CFL}(\varepsilon)$ for all $\varepsilon < g(0)$. Consequently, the lemma holds under Case 1.

**Case 2.** $f_0^0 \neq f_1^0$: *the two local classifiers that achieve perfect local fairness are different.*

The key idea of this proof is: when $\mathrm{MD}_0(f_\varepsilon^{\mathrm{CFL}}), \mathrm{MD}_1(f_\varepsilon^{\mathrm{CFL}}) \geq 0$, then we can always select $\varepsilon_0 = g_0(\lambda_\varepsilon^{\mathrm{CFL}}) = \mathrm{MD}_0(f_\varepsilon^{\mathrm{CFL}}) > 0, \varepsilon_1 = g_1(\lambda_\varepsilon^{\mathrm{CFL}}) = \mathrm{MD}_1(f_\varepsilon^{\mathrm{CFL}}) > 0$ such that $\lambda_{\varepsilon_0}^{\mathrm{UFL}_0} = g_0^{-1}(\varepsilon_0) = \lambda_\varepsilon^{\mathrm{CFL}}$ and $\lambda_{\varepsilon_1}^{\mathrm{UFL}_1} = g_1^{-1}(\varepsilon_1) = \lambda_\varepsilon^{\mathrm{CFL}}$, thus $f_{\varepsilon_0,\varepsilon_1}^{\mathrm{UFL}} = f_\varepsilon^{\mathrm{CFL}}$; when $\mathrm{MD}_0(f_\varepsilon^{\mathrm{CFL}})\mathrm{MD}_1(f_\varepsilon^{\mathrm{CFL}}) = g_0(\lambda_\varepsilon^{\mathrm{CFL}})g_1(\lambda_\varepsilon^{\mathrm{CFL}}) < 0$, however, by Lemma 7, for all $\varepsilon_0, \varepsilon_1 \in [0, 1] \times [0, 1]$, we have $g_0(\lambda_{\varepsilon_0}^{\mathrm{UFL}_0})g_1(\lambda_{\varepsilon_1}^{\mathrm{UFL}_1}) > 0 > g_0(\lambda_\varepsilon^{\mathrm{CFL}})g_1(\lambda_\varepsilon^{\mathrm{CFL}})$, thus there exist $i \in \{0, 1\}$ such that $\lambda_{\varepsilon_i}^{\mathrm{UFL}_i} \neq \lambda_\varepsilon^{\mathrm{CFL}}$ and $f_{\varepsilon_0,\varepsilon_1}^{\mathrm{UFL}} \neq f_\varepsilon^{\mathrm{CFL}}$. Next, we will give rigorous proof.

Since $f_0^0 \neq f_1^0$, we have $\lambda_0^{\mathrm{UFL}_0} \neq \lambda_0^{\mathrm{UFL}_1}$. By Lemma 5, we get $g_0^{-1}(0) \neq g_1^{-1}(0)$. Without any loss of generality, assume $g_0^{-1}(0) < g_1^{-1}(0)$, which implies $g_1(g_0^{-1}(0)) < g_1(g_1^{-1}(0)) = 0$ and $g_0(g_0^{-1}(0)) = 0 < g_0(g_1^{-1}(0))$. Thus we get

$$g_1(g_0^{-1}(0)) < 0 < g_0(g_1^{-1}(0)).$$

Combining the inequality above and $g = \frac{g_0 + g_1}{2}$ yields

$$g(g_0^{-1}(0)) = g_1(g_0^{-1}(0)) < 0 = g(g^{-1}(0)) < g_0(g_1^{-1}(0)) = g(g_1^{-1}(0)). \tag{11}$$

Thus we have

$$\tau = \max\{\mathrm{sign}(g(0))g(g_0^{-1}(0)), \mathrm{sign}(g(0))g(g_1^{-1}(0))\} = g(g_1^{-1}(0)).$$

When $\varepsilon \geq |g(0)|$, by Lemma 11, clearly we have $\max_{\varepsilon_0,\varepsilon_1} \mathrm{UFL}(\varepsilon_0, \varepsilon_1; \varepsilon) = \mathrm{CFL}(\varepsilon)$.

For the other case $\varepsilon < |g(0)|$, by Lemma 7 we have $\lambda = g^{-1}(\varepsilon)$. Similar to Case 1, in order to achieve $\lambda_{\varepsilon_0}^{\mathrm{UFL}_0} = \lambda_{\varepsilon_1}^{\mathrm{UFL}_1} = \lambda_\varepsilon^{\mathrm{CFL}}$, we select $\varepsilon_i = g_i(\lambda_\varepsilon^{\mathrm{CFL}})$.

When $\varepsilon < g(g_1^{-1}(0))$,

$$g_1(\lambda_\varepsilon^{\mathrm{CFL}}) = g_1(g^{-1}(\varepsilon)) < g_1(g^{-1}(g(g_1^{-1}(0)))) = 0.$$

Since $g_1(0) > 0$, by Lemma 7 we have $g_1(\lambda_{\varepsilon_1}^{\mathrm{UFL}_1}) \geq 0$, from the monotonicity of $g_1$ we conclude $\lambda_{\varepsilon_1}^{\mathrm{UFL}_1} \neq \lambda_\varepsilon^{\mathrm{CFL}}$. By Lemma 10, we have $\max_{\varepsilon_0,\varepsilon_1} \mathrm{UFL}(\varepsilon_0, \varepsilon_1; \varepsilon) < \mathrm{CFL}(\varepsilon)$ for all $\varepsilon \leq \tau$.

When $\varepsilon \geq g(g_1^{-1}(0))$, applying (11) we have

$$g_i(0) > \varepsilon_i = g_i(\lambda) = g_i(g^{-1}(\varepsilon)) \geq g_i(g^{-1}(g(g_i^{-1}(0)))) = 0,$$

where the first inequality comes from Case 1. Thus, $\lambda_i = g^{-1}(\varepsilon_i) = \lambda$ as desired, and we obtain $f_{\varepsilon_0,\varepsilon_1}^{\mathrm{UFL}} = f_\varepsilon^{\mathrm{CFL}}$.

Combining both cases above yields the desired conclusion. $\qquad \square$

**Remark 7.** *Recall that we use $f_i^{\varepsilon_i}$ to denote the local classifier trained by client $i$ in UFL analysis. In the expression of $\tau$:*

$$\min\left\{|g(0)|, \max\{\operatorname{sign}(g(0))g(g_0^{-1}(0)), \operatorname{sign}(g(0))g(g_1^{-1}(0))\}\right\},$$

*$|g(0)|$ is the demographic disparity of ERM, $\operatorname{sign}(g(0))g(g_0^{-1}(0))$ is the demographic disparity of local classifier $f_0^0$, and $\operatorname{sign}(g(0))g(g_1^{-1}(0))$ is the demographic disparity of local classifier $f_1^0$. According to the proof of Lemma 13, we obtain $\max\{\operatorname{sign}(g(0))g(g_0^{-1}(0)), \operatorname{sign}(g(0))g(g_1^{-1}(0))\} > 0$ if and only if two local classifiers which achieves perfect local fairness is equal, i.e., $f_0^0 = f_1^0$. Therefore, Lemma 13 implies that, if the ERM is favoring the same group on different clients and the two local classifiers which achieve perfect local fairness are unequal, then UFL performs strictly worse than CFL when the required demographic disparity is smaller than a certain value.*

So far we assume $a \sim \operatorname{Bern}(q)$ for both client 0 and client 1. When both clients do not share the same $q$, we can conclude that CFL outperforms UFL in the following lemma.

**Lemma 14.** *Assume $a \sim \operatorname{Bern}(q_i)$ in client $i$, and $q_0 \neq q_1 \in (0,1)$. Then $\max_{\varepsilon_0,\varepsilon_1} UFL(\varepsilon_0, \varepsilon_1; \varepsilon) < CFL(\varepsilon)$ for all $\varepsilon < |g(0)|$.*

*Proof.* We assemble the dataset from two clients to have $x \mid a = a \sim \mathcal{P}_a = \frac{q_0}{q_0+q_1}\mathcal{P}_a^{(0)} + \frac{q_1}{q_0+q_1}\mathcal{P}_a^{(1)}$, $a = 0, 1$. We let $f_\varepsilon^{\text{CFL}}$ be the solution to CFL($\varepsilon$), with $q = \frac{q_0+q_1}{2}$:

$$f_\varepsilon^{\text{CFL}} = [\![s(x,a) > 0]\!],$$

$$\text{where } s(x,0) = 2\eta(x) - 1 + \frac{\lambda_\varepsilon^{\text{CFL}}}{1-q}, \quad s(x,1) = 2\eta(x) - 1 - \frac{\lambda_\varepsilon^{\text{CFL}}}{q}.$$

Given an UFL classifier $f_{\varepsilon_0,\varepsilon_1}^{\text{UFL}} = (f_0^{\varepsilon_0} + f_1^{\varepsilon_1})/2$ such that DP $\operatorname{Disp}(f_{\varepsilon_0,\varepsilon_1}^{\text{UFL}}) \leq \varepsilon$, the solution reads

$$f_i^{\varepsilon_i} = [\![s_i(x,a) > 0]\!],$$

$$\text{where } s_i(x,0) = 2\eta(x) - 1 + \frac{\lambda_{\varepsilon_i}^{\text{UFL}_i}}{1-q_i}, \quad s_i(x,1) = 2\eta(x) - 1 - \frac{\lambda_{\varepsilon_i}^{\text{UFL}_i}}{q_i}.$$

We prove the lemma by contradiction argument. If $\operatorname{Acc}(f_{\varepsilon_0,\varepsilon_1}^{\text{UFL}}) = \text{CFL}(\varepsilon)$, then $f_{\varepsilon_0,\varepsilon_1}^{\text{UFL}}$ is a solution to CFL($\varepsilon$) with $q = \frac{q_0+q_1}{2}$. Since $\varepsilon < |g(0)|$, by Lemma 7 we have $\lambda_\varepsilon^{\text{CFL}} \neq 0$. Without any loss of generality, assume $\lambda_\varepsilon^{\text{CFL}} < 0$. Below we discuss three cases.

**Case 1.** $\lambda_{\varepsilon_0}^{UFL_0} = \lambda_{\varepsilon_1}^{UFL_1} = 0$: *the UFL classifier is ERM.*

In this case, $f_0^{\varepsilon_0} = f_1^{\varepsilon_1}$. We have $f_{\varepsilon_0,\varepsilon_1}^{\text{UFL}}(x,0) = 1$ for $\eta(x) > \frac{1}{2}$, and $f_\varepsilon^{\text{CFL}} = 0$ for $\eta(x) < (1 - \lambda_\varepsilon^{\text{CFL}}/(1-q))/2$. By Lemma 6, $f_{\varepsilon_0,\varepsilon_1}^{\text{UFL}}$ is not a solution to CFL($\varepsilon$).

**Case 2.** $\lambda_{\varepsilon_0}^{UFL_0} \neq 0$ or $\lambda_{\varepsilon_1}^{UFL_1} \neq 0$, and $\lambda_{\varepsilon_0}^{UFL_0}\lambda_{\varepsilon_1}^{UFL_1} = 0$: *the UFL classifier is not ERM, but one of the local classifier is ERM.*

Without any loss of generality, let $\lambda_{\varepsilon_0}^{\text{UFL}_0} = 0$, $\lambda_{\varepsilon_1}^{\text{UFL}_1} < 0$. Then $f_0^{\varepsilon_0}(x,0) = 1$ for $\eta(x) > \frac{1}{2}$, while $f_1^{\varepsilon_1}(x,0) = 0$ for $\eta(x) < (1 - \lambda_{\varepsilon_0}^{\text{UFL}_0}(1-q_1))/2$. Thus we get

$$f_{\varepsilon_0,\varepsilon_1}^{\text{UFL}}(x,0) = \frac{1}{2} \text{ for } \frac{1}{2} < \eta(x) < (1 - \lambda_{\varepsilon_1}^{\text{UFL}_1}(1-q_1))/2.$$

By Lemma 6, $f_{\varepsilon_0,\varepsilon_1}^{\text{UFL}}$ is not a solution to CFL($\varepsilon$).

**Case 3.** $\lambda_{\varepsilon_0}^{UFL_0}\lambda_{\varepsilon_1}^{UFL_1} \neq 0$: *The local classifiers are not ERM.*

When $\frac{\lambda_{\varepsilon_0}^{\text{UFL}_0}}{1-q_0} \neq \frac{\lambda_{\varepsilon_1}^{\text{UFL}_1}}{1-q_1}$, without loss of generality, let $\frac{\lambda_{\varepsilon_0}^{\text{UFL}_0}}{1-q_0} > \frac{\lambda_{\varepsilon_1}^{\text{UFL}_1}}{1-q_1}$. Then by the same argument in Case 2, we have

$$f_{\varepsilon_0,\varepsilon_1}^{\text{UFL}}(x,0) = \frac{1}{2} \text{ for } \frac{1 - \frac{\lambda_{\varepsilon_0}^{\text{UFL}_0}}{1-q_0}}{2} < \eta(x) < \frac{1 - \frac{\lambda_{\varepsilon_1}^{\text{UFL}_1}}{1-q_1}}{2}.$$

By Lemma 6, $f_{\varepsilon_0,\varepsilon_1}^{\text{UFL}}$ is not a solution to CFL($\varepsilon$).

When $\frac{\lambda_{\varepsilon_0}^{\text{UFL 0}}}{q_0} \neq \frac{\lambda_{\varepsilon_1}^{\text{UFL 1}}}{q_1}$, similarly, $f_{\varepsilon_0,\varepsilon_1}^{\text{UFL}}$ is not a solution to CFL($\varepsilon$).

When $\frac{\lambda_{\varepsilon_0}^{\text{UFL 0}}}{q_0} = \frac{\lambda_{\varepsilon_1}^{\text{UFL 1}}}{q_1}$ and $\frac{\lambda_{\varepsilon_0}^{\text{UFL 0}}}{1-q_0} = \frac{\lambda_{\varepsilon_1}^{\text{UFL 1}}}{1-q_1}$, since $\lambda_{\varepsilon_0}^{\text{UFL 0}} \lambda_{\varepsilon_1}^{\text{UFL 1}} \neq 0$, we have

$$\frac{q_0}{\lambda_{\varepsilon_0}^{\text{UFL 0}}} = \frac{q_1}{\lambda_{\varepsilon_1}^{\text{UFL 1}}}, \quad \frac{1-q_0}{\lambda_{\varepsilon_0}^{\text{UFL 0}}} = \frac{1-q_1}{\lambda_{\varepsilon_1}^{\text{UFL 1}}},$$

which leads to $\frac{1}{\lambda_{\varepsilon_0}^{\text{UFL 0}}} = \frac{1}{\lambda_{\varepsilon_1}^{\text{UFL 1}}}$ and thus $q_0 = q_1$. This contradicts with the assumption that $q_0 \neq q_1$.

Combining all three cases yields desired conclusion.

$\square$

### A.3 FFL VIA FEDAVG ANALYSIS

In this section, we analyze FFL via FEDAVG for the case of two clients. For purpose of illustration, with $I = 2$, we denote $f_{\varepsilon_0,\varepsilon_1}^{\text{FFL via FedAvg}} = f_{\varepsilon}^{\text{FFL via FedAvg}}$ to be the solution to FFL via FEDAVG($\varepsilon$). In Sec. A.3.1, we present a formal version of Thm. 3 and show that compared to UFL, FFL via FEDAVG has strictly higher fairness. In Sec. A.3.2 we derive the solution to FFL via FEDAVG($\varepsilon$), and show it is equivalent to FFL via FEDAVG. In Sec. A.3.3, we analyze the limitation of FFL via FEDAVG and present a formal version of Lemma 4.

#### A.3.1 IMPROVE FAIRNESS VIA FEDERATED LEARNING

Different to UFL, FFL via FEDAVG can reach any $\varepsilon$ demographic disparity:

**Theorem 15** (Formal version of Thm. 3 under two clients cases). *Let $q \in (0,1)$. For all $\varepsilon \in [0,1]$, there exists $\varepsilon_0, \varepsilon_1 \in [0,1]$ such that DP Disp($f_{\varepsilon_0,\varepsilon_1}^{FFL\ via\ FedAvg}$) $\leq \varepsilon$. Thus under the condition in Lemma 8, we have*

$$\min_{\varepsilon_0,\varepsilon_1 \in [0,1]} DP\ Disp(f_{\varepsilon_0,\varepsilon_1}^{UFL}) > \min_{\varepsilon_0,\varepsilon_1 \in [0,1]} DP\ Disp(f_{\varepsilon_0,\varepsilon_1}^{FFL\ via\ FedAvg}) = 0.$$

*Proof.* For any $\varepsilon \in [0,1]$, let $\varepsilon_0 = \varepsilon_1 = \varepsilon$. Then the global DP disparity becomes

$$\begin{aligned}
\text{DP Disp}(f_{\varepsilon_0,\varepsilon_1}^{\text{FFL via FedAvg}}) &= |\mathbb{E}_{\text{x|a}=0} f(\text{x},0) - \mathbb{E}_{\text{x|a}=1} f(\text{x},1)| \\
&= |(\int_{\mathcal{X}} f(x,0)\, d\mathcal{P}_0^{(0)} + \int_{\mathcal{X}} f(x,0)\, d\mathcal{P}_0^{(1)})/2 \\
&\quad - (\int_{\mathcal{X}} f(x,1)\, d\mathcal{P}_1^{(0)} + \int_{\mathcal{X}} f(x,1)\, d\mathcal{P}_1^{(1)})/2| \\
&= |\text{MD}_0(f_{\varepsilon_0,\varepsilon_1}^{\text{FFL via FedAvg}})/2 + \text{MD}_1(f_{\varepsilon_0,\varepsilon_1}^{\text{FFL via FedAvg}})/2| \\
&\leq \text{DP Disp}_0(f_{\varepsilon_0,\varepsilon_1}^{\text{FFL via FedAvg}})/2 + \text{DP Disp}_1(f_{\varepsilon_0,\varepsilon_1}^{\text{FFL via FedAvg}})/2 \\
&\leq (\varepsilon_0 + \varepsilon_1)/2 = \varepsilon.
\end{aligned}$$

$\square$

#### A.3.2 THE BEST CLASSIFIER OF FFL VIA FEDAVG

For FFL via FEDAVG, we directly consider multi-client cases. To visualize the gap between UFL, FFL via FEDAVG, and CFL, in our numerical experiments, we draw finite samples from Gaussian distribution, and then we optimize the empirical risk with the fairness constraint to obtain the classifier trained by FFL via FEDAVG. The following lemma provides the solution to FFL via FEDAVG($\varepsilon$) when $\mathcal{X}$ is finite.

**Lemma 16.** *For finite $\mathcal{X}$, the solution to FFL via FEDAVG($\varepsilon$) is given by*

$$f(x,a) = [\![ \sum_{i \in [I]} s_i(x,a) p_a^{(i)}(x) > 0 ]\!],$$

*where $s_i(x,a) = 2\eta(x) - 1 + I\lambda_i \frac{[\![\text{a}=0]\!]}{1-q} - I\lambda_i \frac{[\![\text{a}=1]\!]}{q}$, for certain $\lambda_0, \dots \lambda_{I-1} \in \mathbb{R}$.*

*Proof.* This proof is based on Menon & Williamson (2018). The key idea of this proof is to use the Lagrangian approach. Before we applying the Lagrangian approach, we will show that FFL via FEDAVG($\varepsilon$) is expressible as a linear program, and thus the strong duality holds.

Since $\mathcal{X}$ is finite, $f$ is a vector of finite dimension. Based on the proof of Lemma 5, the error rate can be written as

$$\mathbb{P}(\hat{y} \neq y) = \sum_{x \in \mathcal{X}, a \in \mathcal{A}} f(x, a)(1 - 2\eta(x))\mathbb{P}(x = x, a = a) + \mathbb{P}(y = 1),$$

and the fairness constraints can be written as

$$\mathbb{P}(\hat{y} = 1 \mid a = 0, i = i) - \mathbb{P}(\hat{y} = 1 \mid a = 1, i = i)$$
$$= \sum_{x \in \mathcal{X}} \left[ f(x, 0)\frac{\mathbb{P}(x = x \mid a = 0, i = i)}{\mathbb{P}(a = 0)} - f(x, 1)\frac{\mathbb{P}(x = x \mid a = 1, i = i)}{\mathbb{P}(a = 1)} \right],$$

for $i \in [I]$. Let $u(x, a) = (1 - 2\eta(x))\mathbb{P}(x = x, a = a)$, $u' = \mathbb{P}(y = 1)$, and $v_i(x, a) = (\llbracket a = 0 \rrbracket - \llbracket a = 1 \rrbracket)\mathbb{P}(x = x \mid a = a, i = i)/\mathbb{P}(a = a)$, for $x \in \mathcal{X}, a \in \mathcal{A}, i \in [I]$. Note that $u, v_0, v_1$ are vectors of the same dimension of $f$. For ease of notation, we allow $\leq$ to be applied to pairs of vectors in an element-wise manner. Therefore, the optimization is

$$\min_{f} u^\top f + u'$$
$$\text{s.t. } v_i^\top f \leq \varepsilon_i$$
$$0 \leq f \leq 1,$$

which is a linear objective with linear constraint. Therefore, the strong duality holds for FFL via FEDAVG($\varepsilon$). Next, we apply Lagrangian approach to solve the FFL via FEDAVG($\varepsilon$).

Recall that

$$\mathbb{P}(\hat{y} \neq y) = \frac{1}{I}\mathbb{E}_a\mathbb{E}_{x \sim \mathcal{P}_a^{(i)}} f(x, a)(1 - 2\eta(x)) + \mathbb{P}(y = 1),$$

and

$$\mathbb{P}(\hat{y} = 1 \mid a = 0, i = i) - \mathbb{P}(\hat{y} = 1 \mid a = 1, i = i)$$
$$= \mathbb{E}_a\mathbb{E}_{x \sim \mathcal{P}_a^{(i)}} \left[ f(x, 0)\frac{\llbracket a = 0 \rrbracket}{1 - q} - f(x, 1)\frac{\llbracket a = 1 \rrbracket}{q} \right].$$

By strong duality, for $\lambda'_0, \ldots, \lambda'_{2I-1} \geq 0$, the corresponding Lagrangian version of FFL via FEDAVG($\varepsilon$) is

$$\min_{f \in \mathcal{F}} \mathbb{P}(\hat{y} \neq y) - \sum_{i \in [I]} \Big[ \lambda'_{2i}[\mathbb{P}(\hat{y} = 1 \mid a = 0, i = i) - \mathbb{P}(\hat{y} = 1 \mid a = 1, i = i) - \varepsilon_i] \qquad (12)$$
$$+ \lambda'_{2i+1}[\varepsilon_i - \mathbb{P}(\hat{y} = 1 \mid a = 0, i = i) - \mathbb{P}(\hat{y} = 1 \mid a = 1, i = i)] \Big]$$

Let $\lambda_i = \lambda'_{2i} - \lambda'_{2i+1}$, $i \in [I]$, then we get

$$(12) = \min_{f \in \mathcal{F}} \mathbb{E}_a\mathbb{E}_{x \sim \mathcal{P}_a^{(i)}} \left[ \frac{1}{I}f(x, a)(1 - 2\eta(x)) - \sum_{i \in [I]} \Big( \lambda_i f(x, 0)\frac{\llbracket a = 0 \rrbracket}{1 - q} - \lambda_i f(x, 1)\frac{\llbracket a = 1 \rrbracket}{q} \Big) \right]$$
$$= \min_{f \in \mathcal{F}} \int_{\mathcal{X}} \sum_{a \in \mathcal{A}} -\frac{1}{I}f(x, a)[\sum_{i \in [I]} s_i(x, a)p_a^{(i)}(x)]\mathrm{d}x.$$

where $s_i$ is defined in Lemma 16. Thus the above equation reaches its minimum at

$$f(x, a) = \llbracket \sum_{i \in [I]} s_i(x, a)p_a^{(i)}(x) > 0 \rrbracket.$$

$\square$

**Remark 8.** *Based on the proof of Lemma 16, FFL via* FEDAVG($\varepsilon$) *is equivalent to solving*

$$\min_{f \in \mathcal{F}} \mathbb{P}(\hat{y} \neq y) - \sum_{i=0}^{I-1} \lambda_i (\mathbb{P}(\hat{y} = 1 \mid a = 0, i = i) - \mathbb{P}(\hat{y} = 1 \mid a = 1, i = i)).$$

*Under certain conditions (assumptions 1 to 4 in Li et al. (2020b)), we have solving FFL via* FEDAVG($\varepsilon$) *is equivalent to minimizing*

$$\mathbb{P}(\hat{y} \neq y \mid i = i) - \lambda_i (\mathbb{P}(\hat{y} = 1 \mid a = 0, i = i) - \mathbb{P}(\hat{y} = 1 \mid a = 1, i = i))$$

*locally and applying* FEDAVG *(Theorem 1 in Li et al. (2020b)).*

### A.3.3 COMPARISON OF FFL VIA FEDAVG AND CFL

**Lemma 17** (Formal version of Lemma 4 under two clients cases)**.** *When* $a \mid i = 0 \sim \text{Bern}(0)$, $a \mid i = 1 \sim \text{Bern}(1)$ *and DP Disp*$(f_1^{CFL}) > 0$*, we have*

$$\min_{\varepsilon_0, \varepsilon_1} DP\ Disp(f_{\varepsilon_0, \varepsilon_1}^{FFL\ via\ FedAvg}) = DP\ Disp(f_1^{CFL}) > \min_{\varepsilon} DP\ Disp(f_\varepsilon^{CFL}) = 0.$$

*Proof.* Since $a \mid i = 0 \sim \text{Bern}(0), a \mid i = 1 \sim \text{Bern}(1)$, the constraints in FFL via FEDAVG($\varepsilon$) vanish. When $\varepsilon = 1$, the constraint in CFL($\varepsilon$) always holds and thus also vanishes. Thus in such scenario the solution to FFL via FEDAVG($\varepsilon$) becomes $f_1^{CFL}$. Then from the assumption we have

$$\text{DP Disp}(f_{\varepsilon_0, \varepsilon_1}^{\text{FFL via FedAvg}}) = \text{DP Disp}(f_1^{\text{CFL}}) > \text{DP Disp}(f_0^{\text{CFL}}) = 0.$$

$\square$

### A.4 EXTENSION TO MULTI-CLIENT CASES

In this subsection, we perform the analysis of UFL and FFL via FEDAVG for the multi-client cases. We present a more general version of Lemma 8, Thm. 15 and Lemma 17.

The following lemma shows the fundamental limitation of UFL:

**Lemma 18** (Formal version of Lemma 1)**.** *Let* $q \in (0, 1)$*. Consider a partition which divides* $I$ *clients into two subsets. Denote the mixture distribution of each subset as* $x \mid a = a, j = j \sim \tilde{\mathcal{P}}_a^{(j)}$*, where* $j$ *is the index of the subset, and* $\tilde{\mathcal{P}}_a^{(j)}$ *is a distribution, for* $a, j = 0, 1$*. Similar to two clients case, define* $\tilde{g}_j(\lambda) = \int_{[\eta^{-1}(\frac{1}{2} - \frac{\lambda}{2(1-q)}), +\infty)} \mathrm{d}\tilde{\mathcal{P}}_0^{(j)} - \int_{[\eta^{-1}(\frac{1}{2} + \frac{\lambda}{2q}), +\infty)} \mathrm{d}\tilde{\mathcal{P}}_1^{(j)}$*. Consider the case that* $q_i = q$ *for all* $i \in [I]$ *and* $q \in (0, 1)$*. Denote the proportion of the two subset as* $J_0$ *and* $J_1$*, where* $J_0, J_1 > 0$ *and* $J_0 + J_1 = 1$*. Let* $c = \min\{|\tilde{g}_0(0)|, |\tilde{g}_1(0)|\}$*. Define* $\tilde{\psi} : [0, c] \times [0, c] \to [-1, 1]$ *as*

$$\tilde{\psi}(\tilde{\varepsilon}_0, \tilde{\varepsilon}_1) = J_0 J_1 \tilde{g}_0(\tilde{g}_1^{-1}(\text{sign}(\tilde{g}_1(0))\tilde{\varepsilon}_1)) + J_0 J_1 \tilde{g}_1(\tilde{g}_0^{-1}(\text{sign}(\tilde{g}_0(0))\tilde{\varepsilon}_0))$$
$$+ J_0^2 \text{sign}(\tilde{g}_0(0))\tilde{\varepsilon}_0 + J_1^2 \text{sign}(\tilde{g}_1(0))\tilde{\varepsilon}_1.$$

*If there exists a partition such that* $\tilde{g}_0(0)\tilde{g}_1(0) < 0$ *and* $\tilde{\psi}(\tilde{\varepsilon}_0, \tilde{\varepsilon}_1)(\tilde{g}_0(0) + \tilde{g}_1(0)) > 0$ *for all* $\tilde{\varepsilon}_0, \tilde{\varepsilon}_1 \in [0, c]$*, then for all* $\tilde{\varepsilon}_0, \tilde{\varepsilon}_1 \in [0, 1]$*, DP Disp*$(f_\varepsilon^{UFL}) \geq \tilde{\delta} = \min\{|\tilde{\psi}(\tilde{\varepsilon}_0, \tilde{\varepsilon}_1)| : \tilde{\varepsilon}_0, \tilde{\varepsilon}_1 \in [0, c]\} > 0$*.*

*Proof.* By Lemma 8, we conclude the achievable fairness range of UFL is strictly smaller than that of CFL. Therefore, pooling the datasets in one subset and perform fair learning can clearly achieve a wider range of fairness than perform fair learning on each client individually. Thus, we can consider the two subsets as two clients with uneven amounts of data, which is almost the same case Lemma 8 considers. Therefore, we follow the same proof idea as Lemma 8 to prove our claim.

Denote the assembled classifier trained from two pooled datasets as $\tilde{f}_{\tilde{\varepsilon}_0, \tilde{\varepsilon}_1}^{\text{UFL}}$. Note that the mean difference can be expressed as

$$\text{MD}(\tilde{f}_{\tilde{\varepsilon}_0, \tilde{\varepsilon}_1}^{\text{UFL}}) = J_0^2 \tilde{g}_0(\tilde{\lambda}_{\tilde{\varepsilon}_0}^{\text{UFL}_0}) + J_0 J_1 \tilde{g}_0(\tilde{\lambda}_{\tilde{\varepsilon}_1}^{\text{UFL}_1}) + J_0 J_1 \tilde{g}_1(\tilde{\lambda}_{\tilde{\varepsilon}_0}^{\text{UFL}_0}) + J_1^2 \tilde{g}_1(\tilde{\lambda}_{\tilde{\varepsilon}_1}^{\text{UFL}_1}). \tag{13}$$

In the following proof, we will show, the mean difference cannot reach 0.

Without any loss of generality, assume $|\tilde{g}_0(0)| < |\tilde{g}_1(0)|$ and $\tilde{g}_1(0) > 0$. By $\tilde{g}_0(0)\tilde{g}_1(0) < 0$ and $\tilde{\psi}(\tilde{\varepsilon}_0, \tilde{\varepsilon}_1)(\tilde{g}_0(0) + \tilde{g}_1(0)) > 0$ for all $\tilde{\varepsilon}_0, \tilde{\varepsilon}_1 \in [0, c]$, we have $\tilde{g}_0(0) < 0$ and $\tilde{\psi}(\tilde{\varepsilon}_0, \tilde{\varepsilon}_1) > 0$ for all $\tilde{\varepsilon}_0, \tilde{\varepsilon}_1 \in [0, c]$. Without any loss of generality, assume $J_0 \leq J_1$.

First, we will prove that UFL achieves its lowest mean difference when $\tilde{\varepsilon}_0, \tilde{\varepsilon}_1 \in [0, c]$. In what follows, we consider five different cases to derive the desired result.

**Case 1.** $\tilde{\varepsilon}_0 > |\tilde{g}_0(0)|, \tilde{\varepsilon}_1 > |\tilde{g}_1(0)|$: *ERM is fair on both clients.*

By (6), we have $\tilde{\lambda}_{\tilde{\varepsilon}_0}^{\mathrm{UFL}_0} = \tilde{\lambda}_{\tilde{\varepsilon}_1}^{\mathrm{UFL}_1} = 0$. Recall $\tilde{g}_i(\cdot)$ is a monotone increasing function, combine $\tilde{g}_1(0) > 0$ and Lemma 7, and thus $\tilde{g}_0(\tilde{g}_1^{-1}(0)) < \tilde{g}_0(0) < 0$. Applying the above conclusion yields

$$(13) = J_0\tilde{g}_0(0) + J_1\tilde{g}_1(0) > \frac{1}{J_1}\left(J_0^2\tilde{g}_0(0) + J_1^2\tilde{g}_1(0) + J_0J_1\tilde{g}_0(\tilde{g}_1^{-1}(0))\right) = \frac{1}{J_1}\tilde{\psi}(\tilde{g}_0(0), 0) \geq \tilde{\delta}.$$

**Case 2.** $\tilde{\varepsilon}_0 \leq |\tilde{g}_0(0)|, \tilde{\varepsilon}_1 > |\tilde{g}_1(0)|$: *ERM is unfair on client 0, but fair on client 1.*

Applying (6) results in $\lambda_{\varepsilon_1}^{\mathrm{UFL}_1} = 0$. By the fact that $\tilde{g}_i(\cdot)$ is a strictly monotone increasing function, we have $\tilde{\lambda}_{\tilde{\varepsilon}_0}^{\mathrm{UFL}_0} = \tilde{g}_0^{-1}(-\tilde{\varepsilon}_0) > \tilde{g}_0^{-1}(\tilde{g}_0(0)) = 0$. Applying the above conclusion yields

$$(13) = -J_0^2\tilde{\varepsilon}_0 + J_1^2\tilde{g}_1(0) + J_0J_1\tilde{g}_0(0) + J_0J_1\tilde{g}_1(\tilde{\lambda}_{\tilde{\varepsilon}_0}^{\mathrm{UFL}_0})$$

$$> J_0\tilde{g}_0(0) + J_1\tilde{g}_1(0) > \frac{1}{J_1}\tilde{\psi}(\tilde{g}_0(0), 0) \geq \tilde{\delta}.$$

In the first equality we used $\tilde{\lambda}_{\tilde{\varepsilon}_0}^{\mathrm{UFL}_0} > 0, \tilde{g}_1(\tilde{\lambda}_{\tilde{\varepsilon}_0}^{\mathrm{UFL}_0}) > \tilde{g}_1(0), \tilde{g}_0(0) < -\tilde{\varepsilon}_0$.

**Case 3.** $\tilde{\varepsilon}_0 \leq |\tilde{g}_0(0)|, \tilde{\varepsilon}_1 \leq |\tilde{g}_1(0)|$: *ERM is unfair on both client 0 and client 1.*

Applying (6) we have $\tilde{\lambda}_{\tilde{\varepsilon}_0}^{\mathrm{UFL}_0} = \tilde{g}_0^{-1}(-\tilde{\varepsilon}_0), \tilde{\lambda}_{\tilde{\varepsilon}_1}^{\mathrm{UFL}_1} = \tilde{g}_1^{-1}(\tilde{\varepsilon}_1)$. Then we have

$$(13) = -J_0^2\tilde{\varepsilon}_0 + J_1^2\tilde{\varepsilon}_1 + J_0J_1\tilde{g}_0(\tilde{g}_1^{-1}(\tilde{\varepsilon}_1)) + J_0J_1\tilde{g}_1(\tilde{g}_0^{-1}(-\tilde{\varepsilon}_0))$$

$$\geq -J_0^2\tilde{\varepsilon}_0 + J_1^2\tilde{\varepsilon}_0 + J_0J_1\tilde{g}_0(\tilde{g}_1^{-1}(\tilde{\varepsilon}_0)) + J_0J_1\tilde{g}_1(\tilde{g}_0^{-1}(-\tilde{\varepsilon}_0)) = \tilde{\psi}(\tilde{\varepsilon}_0, \tilde{\varepsilon}_0) \geq \tilde{\delta}.$$

**Case 4.** $\tilde{\varepsilon}_0 > |\tilde{g}_0(0)|, \tilde{\varepsilon}_1 \leq |\tilde{g}_0(0)|$: *ERM is fair on client 0 and very unfair on client 1.*

By (6), we have $\tilde{\lambda}_{\tilde{\varepsilon}_0}^{\mathrm{UFL}_0} = 0, \tilde{\lambda}_{\tilde{\varepsilon}_1}^{\mathrm{UFL}_1} = \tilde{g}_1^{-1}(\tilde{\varepsilon}_1) > \tilde{g}_1^{-1}(0)$. Then we obtain

$$(13) = J_0^2\tilde{g}_0(0) + J_0J_1\tilde{g}_0(\tilde{\lambda}_{\tilde{\varepsilon}_1}^{\mathrm{UFL}_1}) + J_0J_1\tilde{g}_1(0) + J_1^2\tilde{\varepsilon}_1$$

$$> J_0^2\tilde{g}_0(0) + J_0J_1\tilde{g}_0(\tilde{g}_1^{-1}(0)) + J_1^2\tilde{g}_1(0) = \tilde{\psi}(\tilde{g}_0(0), 0) \geq \tilde{\delta}.$$

**Case 5.** $\tilde{\varepsilon}_0 > |\tilde{g}_0(0)|, |\tilde{g}_0(0)| \leq \tilde{\varepsilon}_1 < |\tilde{g}_1(0)|$: *ERM is fair on client 0 and unfair on client 1.*

Applying (6) implies $\tilde{\lambda}_{\tilde{\varepsilon}_1}^{\mathrm{UFL}_1} = \tilde{g}_1^{-1}(\tilde{\varepsilon}_1) > \tilde{g}_1^{-1}(0)$. Therefore,

$$(13) = J_0^2\tilde{g}_0(0) + J_0J_1\tilde{g}_0(\tilde{g}_1^{-1}(\tilde{\varepsilon}_1)) + J_1^2\tilde{\varepsilon}_1 + J_0J_1\tilde{g}_1(0)$$

$$> J_0^2\tilde{g}_0(0) + J_0J_1\tilde{g}_1(0) + J_0J_1\tilde{g}_0(\tilde{g}_1^{-1}(0)) + J_1^2\tilde{\varepsilon}_1 > \tilde{\psi}(\tilde{g}_0(0), 0) \geq \tilde{\delta}.$$

Combining all the cases above, we conclude that when $\tilde{g}_1(0) > 0$, DP $\mathrm{Disp}(\tilde{f}_{\tilde{\varepsilon}_0, \tilde{\varepsilon}_1}^{\mathrm{UFL}}) \geq \tilde{\delta} = \min\{|\tilde{\psi}(\tilde{\varepsilon}_0, \tilde{\varepsilon}_1)| : \tilde{\varepsilon}_0, \tilde{\varepsilon}_1 \in [0, c]\} > 0$ for all $\tilde{\varepsilon}_0, \tilde{\varepsilon}_1 \in [0, 1]$.

$\square$

**Remark 9.** *Based on the proof above, we can conclude, for the cases with multiple clients, the fundamental limitation of UFL still exists.*

The following theorem shows that FFL via FEDAVG can reach any $\varepsilon$ DP disparity:

**Theorem 19** (Generalized version of Thm. 15). *Let $q_i = q \in (0, 1)$ for all $i \in [I]$. For all $\varepsilon \in [0, 1]$, let $\varepsilon_i \leq \varepsilon$ for all $i \in [I]$, then DP $Disp(f_{\boldsymbol{\varepsilon}}^{FFL\ via\ FedAvg}) \leq \varepsilon$. Thus under the condition in Lemma 18, we have*

$$\min_{\boldsymbol{\varepsilon} \in [0,1]^I} DP\ Disp(f_{\boldsymbol{\varepsilon}}^{UFL}) > \min_{\boldsymbol{\varepsilon} \in [0,1]^I} DP\ Disp(f_{\boldsymbol{\varepsilon}}^{FFL\ via\ FedAvg}) = 0.$$

*Proof.* When $\varepsilon_i \leq \varepsilon$ the global DP disparity becomes

$$\text{DP Disp}(f_{\boldsymbol{\varepsilon}}^{\text{FFL via FedAvg}}) = |\mathbb{E}_{\text{x}|\text{a}=0}f(\text{x},0) - \mathbb{E}_{\text{x}|\text{a}=1}f(\text{x},1)|$$

$$= |(\sum_{i=0}^{I-1}\int_{\mathcal{X}} f(x,0)\,d\mathcal{P}_0^{(i)})/I - (\sum_{i=0}^{I-1}\int_{\mathcal{X}} f(x,1)\,d\mathcal{P}_1^{(i)})/I|$$

$$= |\sum_{i=0}^{I-1}\text{MD}_i(f_{\boldsymbol{\varepsilon}}^{\text{FFL via FedAvg}})/I| \leq \sum_{i=0}^{I-1}\text{DP Disp}_i(f_{\boldsymbol{\varepsilon}}^{\text{FFL via FedAvg}})/I \leq \sum_{i=0}^{I-1}\varepsilon_i/I = \varepsilon.$$

$\square$

The following theorem shows the limitation of FFL via FEDAVG:

**Lemma 20** (Generalized version of Lemma 17). *Let* $\text{a} \mid \text{i} = i \sim \text{Bern}(0)$ *or* $\text{a} \mid \text{i} = i \sim \text{Bern}(1)$ *for all* $i \in [I]$. *When* $DP\,Disp(f_1^{CFL}) > 0$, *we have*

$$\min_{\boldsymbol{\varepsilon}\in[0,1]^I} DP\,Disp(f_{\boldsymbol{\varepsilon}}^{FFL\,via\,FedAvg}) = DP\,Disp(f_1^{CFL}) > \min_{\varepsilon} DP\,Disp(f_{\varepsilon}^{CFL}) = 0.$$

*Proof.* Since $\text{a} \mid \text{i} = i \sim \text{Bern}(0)$ or $\text{a} \mid \text{i} = i \sim \text{Bern}(1)$, the constraints in (FFL via FEDAVG($\boldsymbol{\varepsilon}$)) vanish. When $\varepsilon = 1$, the constraint in CFL($\varepsilon$) always holds and thus also vanishes. Thus in such scenario the solution to (FFL via FEDAVG($\boldsymbol{\varepsilon}$)) becomes $f_1^{\text{CFL}}$. Then from the assumption we have

$$\text{DP Disp}(f_{\boldsymbol{\varepsilon}}^{\text{FFL via FedAvg}}) = \text{DP Disp}(f_1^{\text{CFL}}) > \text{DP Disp}(f_0^{\text{CFL}}) = 0.$$

$\square$

# B    APPENDIX - FEDFB ANALYSIS AND ALGORITHM DESCRIPTION

In this section, we provide our bi-level optimization formulation for FEDFB for four fairness notions: demographic parity, equal opportunity, equalized odds and client parity, and design the corresponding update rule. This development can also be applied to centralized case. Then, we provides more details of how we incorporate FB with federated learning.

To explain how to optimize the weights of different groups, we introduce some necessary notations first. Denote the $k$th sample as $(\text{x}_k, \text{y}_k, \text{a}_k, \text{i}_k)$, $k = 1, \ldots, n$. Here $\text{x}_k \in \mathcal{X}$ is the input feature, $\text{y}_k \in \{0,1\}$ is the label, $\text{a}_k \in \mathcal{A} = [A]$ is the sensitive attribute and $\text{i}_k \in [I]$ is the index of the client that the sample belongs to. $A$ represents the total amount of sensitive attribute and $I$ represents the total amount of clients. Denote the number of samples in group $a$ as $n_{\star,a} := |\{k : \text{a}_k = a\}|$. Let $n_{y,a} := |\{k : \text{y}_k = y, \text{a}_k = a\}|$ be the number of samples in group $a$ of label $y$, and $n_{y,a}^{(i)} := |\{k : \text{y}_k = y, \text{a}_k = a, \text{i}_k = i\}|$ be the number of samples belong to client $i$ of label $y$ and sensitive attribute $a$. Define the loss function as $\ell(\text{y}, \hat{\text{y}})$. Let $L_{y,a}(\boldsymbol{w})$ be the empirical risk aggregated over samples subject to $\text{y} = y, \text{a} = a$, *i.e.*, $L_{y,a}(\boldsymbol{w}) := \sum_{k:\text{y}_k=y,\text{a}_k=a} \ell(\text{y}_k, \hat{\text{y}}_k)/n_{y,a}$, where $n_{y,a} := |\{k : \text{y}_k = y, \text{a}_k = a\}|$. We then define the local version of $L_{y,a}$ as $L_{y,a}^{(i)}(\boldsymbol{w}) := \sum_{k:\text{y}_k=y,\text{a}_k=a,\text{i}_k=i} \ell(\text{y}_k, \hat{\text{y}}_k)/n_{y,a}^{(i)}$.

## B.1    FEDFB *w.r.t* DEMOGRAPHIC PARITY

To extend Roh et al. (2021) to multiple groups cases, we propose a different bi-level optimization problem *w.r.t* demographic parity. This development can also be applied to the centralized setting. The following proposition gives a necessary sufficient condition for demographic parity, which can be directly obtained from Roh et al. (2021). For completeness, we also include the proof here.

**Proposition 21** (Necessary sufficient condition for demographic parity, Proposition 2 in Roh et al. (2021)). *Consider 0-1 loss:* $\ell(\text{y}, \hat{\text{y}}) = [\![\text{y} \neq \hat{\text{y}}]\!]$. *Let* $L'_{y,a}(\boldsymbol{w}) := \frac{n_{y,a}}{n_{\star,a}} L_{y,a}(\boldsymbol{w})$, *then*

$$-L'_{0,0}(\boldsymbol{w}) + L'_{1,0}(\boldsymbol{w}) + L'_{0,a}(\boldsymbol{w}) - L'_{1,a}(\boldsymbol{w}) + \frac{n_{0,0}}{n_{\star,0}} - \frac{n_{0,a}}{n_{\star,a}} = 0 \qquad (14)$$

*for all* $a \in [A]$ *is a necessary sufficient condition for demographic parity.*

*Proof.* We denote by $\mathbb{P}$ the empirical probability. The demographic parity is satisfied when $\mathbb{P}(\hat{y} = 1 \mid a = 0) = \mathbb{P}(\hat{y} = 1 \mid a = a)$ holds for all $a \in [A]$. Thus,

$$\mathbb{P}(\hat{y} = 1, y = 0 \mid a = 0) + \mathbb{P}(\hat{y} = 1, y = 1 \mid a = 0)$$
$$= \mathbb{P}(\hat{y} = 1, y = 0 \mid a = a) + \mathbb{P}(\hat{y} = 1, y = 1 \mid a = a)$$

For 0-1 loss, we have $\ell(|1 - y|, \cdot) = 1 - \ell(y, \cdot)$, thus

$$\frac{1}{n_{\star,0}} \sum_{k:y_k=0, a_k=0} (1 - \ell(y_k, \hat{y}_k)) + \frac{1}{n_{\star,0}} \sum_{k:y_k=1, a_k=0} \ell(y_k, \hat{y}_k)$$
$$= \frac{1}{n_{\star,a}} \sum_{k:y_k=0, a_k=a} (1 - \ell(y_k, \hat{y}_k)) + \frac{1}{n_{\star,a}} \sum_{k:y_k=1, a_k=a} \ell(y_k, \hat{y}_k).$$

By replacing $\sum_{k:y_k=y, a_k=a} \ell(y_k, \hat{y}_k) = n_{y,a} L_{y,a}(\boldsymbol{w})$, we have

$$\frac{n_{0,0}}{n_{\star,0}}(1 - L_{0,0}(\boldsymbol{w})) + \frac{n_{1,0}}{n_{\star,0}} L_{1,0}(\boldsymbol{w})$$
$$= \frac{n_{0,a}}{n_{\star,a}}(1 - L_{0,a}(\boldsymbol{w})) + \frac{n_{1,a}}{n_{\star,a}} L_{1,a}(\boldsymbol{w}).$$

$\square$

**Remark 10.** *Proposition 21 provides us a way to measure demographic disparity using group-specific losses. However, since 0-1 loss is noncontinuous, in practice, we use the "continuous surrogate" of it, which is cross entropy.*

The necessary sufficient condition for demographic parity (14) inspires us to achieve demographic parity by connecting one group with all the other groups for non-binary sensitive attribute settings. To achieve demographic parity, we introduce another parameter: $\boldsymbol{\lambda} = (\lambda_0, \dots, \lambda_{A-1})$, the weights attached to samples. We formalize the reweighting task into the following bi-level optimization problem, where the outer objective function captures the demographic parity criterion:

$$\min_{\boldsymbol{\lambda} \in \Lambda} F_{\mathrm{dp}}(\boldsymbol{\lambda}) := \min_{\boldsymbol{\lambda} \in \Lambda} \underbrace{\sum_{a=1}^{A-1} \left( -L'_{0,0}(\boldsymbol{w}_{\boldsymbol{\lambda}}) + L'_{1,0}(\boldsymbol{w}_{\boldsymbol{\lambda}}) + L'_{0,a}(\boldsymbol{w}_{\boldsymbol{\lambda}}) - L'_{1,a}(\boldsymbol{w}_{\boldsymbol{\lambda}}) + \frac{n_{0,0}}{n_{\star,0}} - \frac{n_{0,a}}{n_{\star,a}} \right)^2}_{(15)_{\mathrm{outer}}}$$

$$\boldsymbol{w}_{\boldsymbol{\lambda}} := \arg\min_{\boldsymbol{w}} L(\boldsymbol{w}, \boldsymbol{\lambda}) = \arg\min_{\boldsymbol{w}} \underbrace{\sum_{a=0}^{A-1} \left[ \lambda_a L'_{0,a}(\boldsymbol{w}) + (2\frac{n_{\star,a}}{n} - \lambda_a)L'_{1,a}(\boldsymbol{w}) \right]}_{(15)_{\mathrm{inner}}}, \qquad (15)$$

where $\Lambda = [0, 2\frac{n_{\star,0}}{n}] \times \cdots \times [0, 2\frac{n_{\star,A-1}}{n}]$.

We make the following assumption to our loss function to have the decreasing direction in Lemma 22.

**Assumption 1.** $L'_{y,a}(\cdot)$ *is twice differentiable for all* $y \in \{0, 1\}$, $a \in [A]$, *and*

$$\sum_{a=0}^{A-1} \left[ \lambda_a \nabla^2 L'_{0,a}(\boldsymbol{w}) + (2\frac{n_{\star,a}}{n} - \lambda_a)\nabla^2 L'_{1,a}(\boldsymbol{w}) \right] \succ 0 \text{ for all } \lambda \in \Lambda. \qquad (16)$$

If $L'_{y,a}(\boldsymbol{w}_{\boldsymbol{\lambda}})$ is convex for all $y \in \{0, 1\}, a \in [A]$, the condition (16) holds unless for all $a$, $L'_{0,a}(\cdot), L'_{1,a}(\cdot)$ share their stationary points, which is very unlikely (see Remark 1 in Roh et al. (2021)).

The following lemma provides a decreasing direction of the outer objective function $F_{\mathrm{dp}}$, which inspired us to design the update rule of $\boldsymbol{\lambda}$.

**Lemma 22** (Decreasing direction of $F_{\mathrm{dp}}$). *If Assumption 1 holds, then on the direction* $\boldsymbol{\mu}(\boldsymbol{\lambda}) = (\mu_0(\boldsymbol{\lambda}), \dots, \mu_{A-1}(\boldsymbol{\lambda}))$ *where*

$$\mu_0(\boldsymbol{\lambda}) = -\sum_{a=1}^{A-1} \left( -L'_{0,0}(\boldsymbol{w}_{\boldsymbol{\lambda}}) + L'_{1,0}(\boldsymbol{w}_{\boldsymbol{\lambda}}) + L'_{0,a}(\boldsymbol{w}_{\boldsymbol{\lambda}}) - L'_{1,a}(\boldsymbol{w}_{\boldsymbol{\lambda}}) + \frac{n_{0,0}}{n_{\star,0}} - \frac{n_{0,a}}{n_{\star,a}} \right),$$

$$\mu_a(\boldsymbol{\lambda}) = -L'_{0,0}(\boldsymbol{w}_{\boldsymbol{\lambda}}) + L'_{1,0}(\boldsymbol{w}_{\boldsymbol{\lambda}}) + L'_{0,a}(\boldsymbol{w}_{\boldsymbol{\lambda}}) - L'_{1,a}(\boldsymbol{w}_{\boldsymbol{\lambda}}) + \frac{n_{0,0}}{n_{\star,0}} - \frac{n_{0,a}}{n_{\star,a}}$$

$$(17)$$

*for all $a \in \{1, \dots, A-1\}$, we have $\boldsymbol{\mu}(\boldsymbol{\lambda}) \cdot \nabla F_{\mathrm{dp}}(\boldsymbol{\lambda}) \leq 0$, and the equality holds if only if $\boldsymbol{\mu}(\boldsymbol{\lambda}) = \mathbf{0}$.*

*Proof.* We compute the derivative as

$$\frac{\partial F_{\mathrm{dp}}(\boldsymbol{\lambda})}{\partial \lambda_j} = 2 \sum_{a=1}^{A-1} \left[ \left( -L'_{0,0}(\boldsymbol{w_\lambda}) + L'_{1,0}(\boldsymbol{w_\lambda}) + L'_{0,a}(\boldsymbol{w_\lambda}) - L'_{1,a}(\boldsymbol{w_\lambda}) + \frac{n_{0,0}}{n_{\star,0}} - \frac{n_{0,a}}{n_{\star,a}} \right) \right.$$
$$\left. \left( -\nabla L'_{0,0}(\boldsymbol{w_\lambda}) + \nabla L'_{1,0}(\boldsymbol{w_\lambda}) + \nabla L'_{0,a}(\boldsymbol{w_\lambda}) - \nabla L'_{1,a}(\boldsymbol{w_\lambda}) \right) \right] \frac{\partial \boldsymbol{w_\lambda}}{\partial \lambda_j}. \tag{18}$$

Note that $\boldsymbol{w_\lambda}$ is the minimizer to $(15)_{\mathrm{inner}}$, we have

$$\sum_{a=0}^{A-1} \left[ \lambda_a \nabla L'_{0,a}(\boldsymbol{w_\lambda}) + (2\frac{n_{\star,a}}{n} - \lambda_a) \nabla L'_{1,a}(\boldsymbol{w_\lambda}) \right] = 0.$$

We take the $\lambda_j$ derivative to the above equation and have

$$\nabla L'_{0,j}(\boldsymbol{w_\lambda}) - \nabla L'_{1,j}(\boldsymbol{w_\lambda}) + \sum_{a=0}^{A-1} \left[ \lambda_a \nabla^2 L'_{0,a}(\boldsymbol{w_\lambda}) + (2\frac{n_{\star,a}}{n} - \lambda_a) \nabla^2 L'_{1,a}(\boldsymbol{w_\lambda}) \right] \frac{\partial \boldsymbol{w_\lambda}}{\partial \lambda_j} = 0.$$

Thus we get

$$\frac{\partial \boldsymbol{w_\lambda}}{\partial \lambda_j} = \left( \sum_{a=0}^{A-1} \left[ \lambda_a \nabla^2 L'_{0,a}(\boldsymbol{w_\lambda}) + (2\frac{n_{\star,a}}{n} - \lambda_a) \nabla^2 L'_{1,a}(\boldsymbol{w_\lambda}) \right] \right)^{-1} [\nabla L'_{1,j}(\boldsymbol{w_\lambda}) - \nabla L'_{0,j}(\boldsymbol{w_\lambda})]. \tag{19}$$

Then on the direction $\boldsymbol{\mu}(\boldsymbol{\lambda})$ given by (17), we combine (18) and (19) to have

$$\boldsymbol{\mu}(\boldsymbol{\lambda}) \cdot \nabla F_{\mathrm{dp}}(\boldsymbol{\lambda})$$
$$= 2 \left( \sum_{a=1}^{A-1} \left[ \left( -L'_{0,0}(\boldsymbol{w_\lambda}) + L'_{1,0}(\boldsymbol{w_\lambda}) + L'_{0,a}(\boldsymbol{w_\lambda}) - L'_{1,a}(\boldsymbol{w_\lambda}) + \frac{n_{0,0}}{n_{\star,0}} - \frac{n_{0,a}}{n_{\star,a}} \right) \right. \right.$$
$$\left. \left. \left( -\nabla L'_{0,0}(\boldsymbol{w_\lambda}) + \nabla L'_{1,0}(\boldsymbol{w_\lambda}) + \nabla L'_{0,a}(\boldsymbol{w_\lambda}) - \nabla L'_{1,a}(\boldsymbol{w_\lambda}) \right) \right] \right)$$
$$\left( \sum_{a=0}^{A-1} \left[ \lambda_a \nabla^2 L'_{0,a}(\boldsymbol{w_\lambda}) + (2\frac{n_{\star,a}}{n} - \lambda_a) \nabla^2 L'_{1,a}(\boldsymbol{w_\lambda}) \right] \right)^{-1}$$
$$\left( -\sum_{a=1}^{A-1} \left[ \left( -L'_{0,0}(\boldsymbol{w_\lambda}) + L'_{1,0}(\boldsymbol{w_\lambda}) + L'_{0,a}(\boldsymbol{w_\lambda}) - L'_{1,a}(\boldsymbol{w_\lambda}) + \frac{n_{0,0}}{n_{\star,0}} - \frac{n_{0,a}}{n_{\star,a}} \right) \right. \right.$$
$$\left. \left. \left( -\nabla L'_{0,0}(\boldsymbol{w_\lambda}) + \nabla L'_{1,0}(\boldsymbol{w_\lambda}) + \nabla L'_{0,a}(\boldsymbol{w_\lambda}) - \nabla L'_{1,a}(\boldsymbol{w_\lambda}) \right) \right] \right) \leq 0$$

where we have used Assumption 1, and the equality holds only when $\boldsymbol{\mu}(\boldsymbol{\lambda}) = \mathbf{0}$. $\qquad \square$

Inspired by Lemma 22, in each communication round $t = 0, 1, \dots$, we design update rule for $\boldsymbol{\lambda}$ as:

$$\lambda_a^{(t+1)} = \lambda_a^{(t)} + \frac{\alpha_t}{\|\boldsymbol{\mu}(\boldsymbol{\lambda}^{(t)})\|_2} \mu_a(\boldsymbol{\lambda}^{(t)}), \text{ for } a \in [A], \tag{20}$$

where $\alpha_t$ is the step size.

Now we introduce how clients collaborate to solve the bi-level optimization problem (15).

First, we focus on the outer objective function $(15)_{\mathrm{outer}}$ and introduce how clients collaborate to update the weight $\boldsymbol{\lambda}$. Note that the central server can compute $L'_{y,a}(\boldsymbol{w}_{\boldsymbol{\lambda}^{(t)}})$ by weight-averaging the local group loss $L^{(i)}_{y,a}(\boldsymbol{w}_{\boldsymbol{\lambda}^{(t)}})$ sent from clients at communication rounds as

$$L'_{y,a}(\boldsymbol{w}_{\boldsymbol{\lambda}^{(t)}}) = \sum_{i \in [I]} \frac{n^{(i)}_{y,a}}{n_{\star,a}} L^{(i)}_{y,a}(\boldsymbol{w}_{\boldsymbol{\lambda}^{(t)}}),$$

thereby obtain $\boldsymbol{\mu}(\boldsymbol{\lambda}^{(t)})$ and update $\boldsymbol{\lambda}^{(t)}$ by (20).

Next, we focus on the inner objective function $(15)_{\text{inner}}$ and introduce how clients collaborate to update the model parameters $\boldsymbol{w_\lambda}$ using FEDAVG. Note that we can decompose the objective function $L(\boldsymbol{w}, \boldsymbol{\lambda})$ into $L(\boldsymbol{w}, \boldsymbol{\lambda}) = \sum_{i \in [I]} L^{(i)}(\boldsymbol{w}, \boldsymbol{\lambda})$, where $L^{(i)}(\boldsymbol{w}, \boldsymbol{\lambda}) := \sum_{a=0}^{A-1} [\lambda_a n_{0,a}^{(i)} L_{0,a}^{(i)} / n_{\star,a} + (2\frac{n_{\star,a}}{n} - \lambda_a) n_{1,a}^{(i)} L_{1,a}^{(i)} / n_{\star,a}]$ is the client objective function of client $i$. The global objective can be seen as a weighted sum of the client objective function. Therefore, we can use FEDAVG to solve the inner optimization problem.

We present the pseudocode of FEDFB *w.r.t* demographic parity in Algorithm 2.

---

**Algorithm 2:** FEDFB *w.r.t* Demographic Parity

---

**Server executes:**

    **input** :Learning rate $\{\alpha_t\}_{t \in \mathbb{N}}$;

    Initialize $\lambda_a$ as $\frac{n_{\star,a}}{n}$ for all $a \in [A] \backslash \{0\}$;

    **for** *each iteration* $t = 1, 2, \dots$ **do**

        Clients perform updates;

        $\boldsymbol{w_\lambda} \leftarrow \text{SecAgg}\left\{\boldsymbol{w}^{(i)}\right\}$ for all $i$;

        $L_{y,a} \leftarrow \text{SecAgg}\left\{L_{y,a}^{(i)}\right\}$ for all $(y, a)$;

        $L'_{y,a} \leftarrow \frac{n_{y,a}}{n_{\star,a}} L_{y,a}$ for all $y \in \{0,1\}, a \in [A]$;

        $\mu_0 \leftarrow -\sum_{a=1}^{A-1}\left(-L'_{0,0} + L'_{1,0} + L'_{0,a} - L'_{1,a} + \frac{n_{0,0}}{n_{\star,0}} - \frac{n_{0,a}}{n_{\star,a}}\right)$;

        $\mu_a \leftarrow -L'_{0,0} + L'_{1,0} + L'_{0,a} - L'_{1,a} + \frac{n_{0,0}}{n_{\star,0}} - \frac{n_{0,a}}{n_{\star,a}}, a \in [A] \backslash \{0\}$;

        $\lambda_a \leftarrow \lambda_a + \frac{\alpha_t}{\|\boldsymbol{\mu}\|_2} \mu_a$, for all $a \in [A]$;

        Broadcast $\boldsymbol{w_\lambda}$ and $\boldsymbol{\lambda}$ to clients;

    **end**

    **output**:$w_\lambda$

**ClientUpdate**$(i, \boldsymbol{w}, \boldsymbol{\lambda})$**:**

    $\boldsymbol{w}^{(i)} \leftarrow$ Gradient descent *w.r.t* objective function

    $\sum_{a=0}^{A-1}\left[\lambda_a L_{0,a}'^{(i)}(\boldsymbol{w}) + (2\frac{n_{\star,a}}{n} - \lambda_a) L_{1,a}'^{(i)}(\boldsymbol{w})\right]$;

    Send $\boldsymbol{w}^{(i)}, L_{0,a}^{(i)}(\boldsymbol{w}), L_{1,a}^{(i)}(\boldsymbol{w})$ for all $a \in [A]$ to server via a SecAgg protocol;

---

Next, we analyze the convergence performance of FEDFB. We need to make the following assumptions on the objective function $L^{(i)}(\boldsymbol{w}, \boldsymbol{\lambda})$, $i \in [I]$. For simplicity, we drop the $\boldsymbol{\lambda}$ here and use the notations $L^{(i)}(\boldsymbol{w})$ and $L(\boldsymbol{w})$ instead. We use $\boldsymbol{w}_t^{(i)}$ to denote the model parameters at $t$-th iteration in $i$-th client. The assumptions below are proposed by work Li et al. (2020b):

**Assumption 2** (Strong convexity, Assumption 1 in Li et al. (2020b)). *$L^{(i)}(\boldsymbol{w})$ is $\mu$-strongly convex for $i \in [I]$, i.e., for all $\boldsymbol{v}$ and $\boldsymbol{w}$, $\boldsymbol{w}$, $L^{(i)}(\boldsymbol{v}) \geq L^{(i)}(\boldsymbol{w}) + (\boldsymbol{v} - \boldsymbol{w})^\top \nabla L^{(i)}(\boldsymbol{w}) + \frac{\mu}{2}\|\boldsymbol{v} - \boldsymbol{w}\|_2^2$.*

**Assumption 3** (Smoothness, Assumption 2 in Li et al. (2020b)). *$L^{(i)}(\boldsymbol{w})$ is $L$-smooth for $i \in [I]$, i.e., for all $\boldsymbol{v}$ and $\boldsymbol{w}$, $L^{(i)}(\boldsymbol{v}) \leq L^{(i)}(\boldsymbol{w}) + (\boldsymbol{v} - \boldsymbol{w})^\top \nabla L^{(i)}(\boldsymbol{w}) + \frac{L}{2}\|\boldsymbol{v} - \boldsymbol{w}\|_2^2$.*

**Assumption 4** (Bounded variance, Assumption 3 in Li et al. (2020b)). *Let $\xi_t^{(i)}$ be sampled from $i$-th device's local data uniformly at random, where $t \in [T]$, and $T$ is the total number of every client's SGDs. The variance of stochastic gradients in each device is bounded: $\mathbb{E}\left\|\nabla L^{(i)}(\boldsymbol{w}_t^{(i)}, \xi_t^{(i)}) - \nabla L^{(i)}(\boldsymbol{w}_t^{(i)})\right\|^2 < \infty$ for $i \in [I]$.*

**Assumption 5** (Bounded gradients, Assumption 4 in Li et al. (2020b)). *The expected squared norm of stochastic gradients is uniformly bounded, i.e., $\mathbb{E}\left\|\nabla L^{(i)}(\boldsymbol{w}_t^{(i)}, \xi_t^{(i)})\right\| < \infty$ for all $i \in [I], t \in [T]$.*

In FEDAVG, first, the central server broadcasts the lastest model to all clients, then, every client performs local updates for a number of iterations, last, the central server aggregates the local models to produce the new global model(see Algorithm Description in Li et al. (2020b) for more detailed explanation).

We denote $E$ as the number local iterations performed in a client between two communications, and $R$ be the total number of every client's iteration. Thus $\frac{R}{E}$ is the number of communications.

With the above assumptions, the following theorem shows the convergence of FedFB in the case of two clients.

**Theorem 23.** *Consider the case of $A = 2$. Let Assumption 2, 3, 4, 5 on $\{L^{(i)}(\cdot, \boldsymbol{\lambda})\}_{i \in [I]}$ and Assumption 1 on $\{L'_{y,a}(\cdot)\}_{y,a \in \{0,1\}}$ hold. Choose $\{\alpha_t\}_{t=1}^{\infty}$ such that $\lim_{t \to \infty} \alpha_t = 0, \sum_{t=1}^{\infty} \alpha_t = \infty$. Suppose we use FEDAVG with $R$ and $E$ satisfying $\frac{E^2}{R} \to 0$ to solve $(15)_{inner}$ between two $\boldsymbol{\lambda}$ update rounds. We can find sufficiently large $T$ such that with high probability, applying update rule (20) leads to $|\lambda_a^{(T)} - \lambda_a^{\star}| \le \max\left\{|\lambda_a^{(0)} - \lambda_a^{\star}| - \sum_{t=1}^{T} \alpha_t, \alpha_T\right\} \to 0$, where $\boldsymbol{\lambda}^{\star}$ is the local minimizer on direction $\boldsymbol{\mu}(\boldsymbol{\lambda})$.*

*Proof.* We first derive the update rule in the case of $A = 2$. From (20), we have $\mu_0 = -\mu_1$. Denote

$$f(\boldsymbol{\lambda}^{(t)}) = -L'_{0,0}(\boldsymbol{w}_{\boldsymbol{\lambda}^{(t)}}) + L'_{1,0}(\boldsymbol{w}_{\boldsymbol{\lambda}}^{(t)}) + L'_{0,1}(\boldsymbol{w}_{\boldsymbol{\lambda}}^{(t)}) - L'_{1,1}(\boldsymbol{w}_{\boldsymbol{\lambda}}^{(t)}) + \frac{n_{0,0}}{n_{\star,0}} - \frac{n_{0,1}}{n_{\star,1}}.$$

Then the update rule becomes

$$\begin{aligned}
\lambda_0^{(t+1)} &= \lambda_0^{(t)} - \frac{\sqrt{2}}{2} \alpha_t \text{sign}(f(\boldsymbol{\lambda}^{(t)})) \\
\lambda_1^{(t+1)} &= \lambda_1^{(t)} + \frac{\sqrt{2}}{2} \alpha_t \text{sign}(f(\boldsymbol{\lambda}^{(t)})).
\end{aligned} \tag{21}$$

Then we apply FEDAVG with $R$ number of total iterations to solve $\min_{\boldsymbol{w}} L(\boldsymbol{w}; \boldsymbol{\lambda}^{(t)})$, and obtain $\boldsymbol{w}_R^{(t)}$. Then in the update round from $\boldsymbol{\lambda}^{(t)}$ to $\boldsymbol{\lambda}^{(t+1)}$, by Thm. 1 in Li et al. (2020b), we have

$$\begin{aligned}
\mathbb{E}[L(\boldsymbol{w}_R^{(t)}; \boldsymbol{\lambda}^{(t)})] - L(\boldsymbol{w}_{\boldsymbol{\lambda}^{(t)}}; \boldsymbol{\lambda}^{(t)}) &= O\left(\frac{E^2}{R}\right), \\
\mathbb{E}\left\|\boldsymbol{w}_R^{(t)} - \boldsymbol{w}_{\boldsymbol{\lambda}^{(t)}}\right\|^2 &= O\left(\frac{E^2}{R}\right),
\end{aligned}$$

where $\boldsymbol{w}_{\boldsymbol{\lambda}^{(t)}} = \arg\min_{\boldsymbol{w}} L(\boldsymbol{w}; \boldsymbol{\lambda}^{(t)})$. By Markov's inequality, with probability $1 - \delta$,

$$\begin{aligned}
L(\boldsymbol{w}_R^{(t)}; \boldsymbol{\lambda}^{(t)}) - L(\boldsymbol{w}_{\boldsymbol{\lambda}^{(t)}}; \boldsymbol{\lambda}^{(t)}) &= O\left(\frac{E^2}{\delta R}\right), \\
\left\|\boldsymbol{w}_R^{(t)} - \boldsymbol{w}_{\boldsymbol{\lambda}^{(t)}}\right\|^2 &= O\left(\frac{E^2}{\delta R}\right).
\end{aligned}$$

Then taking the union bound over $T$ updating iterations of $\boldsymbol{\lambda}^{(t)}$, the conclusions above hold with probability at least $1 - T\delta$ for all $\boldsymbol{\lambda}^{(t)}, t = 1, 2, \cdots, T$. Therefore, with sufficiently large $R = R(\delta)$ such that $\frac{E^2}{\delta R} \to 0$, for all $\boldsymbol{\lambda}^{(t)}$ in the $T$ iteration, $|L(\boldsymbol{w}_R^{(t)}; \boldsymbol{\lambda}^{(t)}) - L(\boldsymbol{w}_{\boldsymbol{\lambda}^{(t)}}; \boldsymbol{\lambda}^{(t)})| \ll 1$, $\|\boldsymbol{w}_R^{(t)} - \boldsymbol{w}_{\boldsymbol{\lambda}^{(t)}}\| \ll 1$.

By Lemma 22, $F_{\text{dp}}(\boldsymbol{\lambda})$ has a local minimizer $\boldsymbol{\lambda}^{\star}$ on direction $\boldsymbol{\mu}(\boldsymbol{\lambda})$. By update rule (21), we can find large $T > 0$ to have

$$|\lambda_a^{(T)} - \lambda_a^{\star}| \le \max\left\{|\lambda_a^{(0)} - \lambda_a^{\star}| - \sum_{t=1}^{T} \alpha_t, \alpha_T\right\} \to 0,$$

where $a \in \{0, 1\}$. □

**Remark 11.** *Note that Thm. 23 assumes FEDFB does not update $\boldsymbol{\lambda}$ in each communication round and there are infinite rounds of aggregations between two $\boldsymbol{\lambda}$ updating round. However, for computation efficiency, we update $\boldsymbol{\lambda}$ at every communication round in practice.*

### B.2 FEDFB *w.r.t* EQUAL OPPORTUNITY

Similar to Proposition 21, we design the following bi-level optimization problems to capture equal opportunity:

$$
\min_{\boldsymbol{\lambda} \in \Lambda} F_{\mathrm{eo}}(\boldsymbol{\lambda}) = \min_{\boldsymbol{\lambda} \in \Lambda} \sum_{a=1}^{A-1} (L_{1,a}(\boldsymbol{w}_{\boldsymbol{\lambda}}) - L_{1,0}(\boldsymbol{w}_{\boldsymbol{\lambda}}))^2
$$

$$
\boldsymbol{w}_{\boldsymbol{\lambda}} = \arg\min_{\boldsymbol{w}} \sum_{a=1}^{A-1} \lambda_a L_{1,a}(\boldsymbol{w}) + (\frac{n_{1,\star}}{n} - \sum_{a=1}^{A-1} \lambda_a) L_{1,0}(\boldsymbol{w}) + \frac{n_{0,\star}}{n} L_{0,\star}(\boldsymbol{w}). \tag{22}
$$

Here $\Lambda = \{(\lambda_1, \ldots, \lambda_{A-1}) : \lambda_1 + \cdots + \lambda_{A-1} \leq \frac{n_{1,\star}}{n}, \lambda_a \geq 0 \text{ for all } a = 1, \ldots, A-1\}$.

For equal opportunity, we make the following assumption:

**Assumption 6.** $L_{y,a}(\cdot)$ *is twice differentiable for all* $y \in \{0,1\}$, $a \in [A]$, *and*

$$
\sum_{a=1}^{A-1} \lambda_a \nabla^2 L_{1,a}(\boldsymbol{w}_{\boldsymbol{\lambda}}) + (\frac{n_{1,\star}}{n} - \sum_{a=1}^{A-1} \lambda_a) \nabla^2 L_{1,0}(\boldsymbol{w}_{\boldsymbol{\lambda}}) + \frac{n_{0,\star}}{n} \nabla^2 L_{0,\star}(\boldsymbol{w}_{\boldsymbol{\lambda}}) \succ 0
$$

*for all* $\lambda \in \Lambda$.

With the above assumption, the following lemma provides the update rule:

**Lemma 24.** *If Assumption 6 holds, then on the direction*

$$
\boldsymbol{\mu}(\boldsymbol{\lambda}) = (L_{1,1}(\boldsymbol{w}_{\boldsymbol{\lambda}}) - L_{1,0}(\boldsymbol{w}_{\boldsymbol{\lambda}}), \ldots, L_{1,A-1}(\boldsymbol{w}_{\boldsymbol{\lambda}}) - L_{1,0}(\boldsymbol{w}_{\boldsymbol{\lambda}})), \tag{23}
$$

*we have* $\boldsymbol{\mu}(\boldsymbol{\lambda}) \cdot \nabla F_{\mathrm{eo}}(\boldsymbol{\lambda}) \leq 0$, *and the equality holds if only if* $\boldsymbol{\mu}(\boldsymbol{\lambda}) = \boldsymbol{0}$.

**Update rule for equal opportunity:**

$$
\lambda_a^{(t+1)} = \lambda_a^{(t)} + \frac{\alpha_t}{\|\boldsymbol{\mu}(\boldsymbol{\lambda}^{(t)})\|_2} (L_{1,a}(\boldsymbol{w}_{\boldsymbol{\lambda}^{(t)}}) - L_{1,0}(\boldsymbol{w}_{\boldsymbol{\lambda}^{(t)}})).
$$

*Proof of Lemma 24.* We compute the derivative as

$$
\frac{\partial F_{\mathrm{eo}}(\boldsymbol{\lambda})}{\partial \lambda_j} = 2 \sum_{a=1}^{A-1} (L_{1,a}(\boldsymbol{w}_{\boldsymbol{\lambda}}) - L_{1,0}(\boldsymbol{w}_{\boldsymbol{\lambda}}))(\nabla L_{1,a}(\boldsymbol{w}_{\boldsymbol{\lambda}}) - \nabla L_{1,0}(\boldsymbol{w}_{\boldsymbol{\lambda}})) \frac{\partial \boldsymbol{w}_{\boldsymbol{\lambda}}}{\partial \lambda_j}. \tag{24}
$$

Note that $\boldsymbol{w}_{\boldsymbol{\lambda}}$ is the minimizer to (22), we have

$$
\sum_{a=1}^{A-1} \lambda_a \nabla L_{1,a}(\boldsymbol{w}_{\boldsymbol{\lambda}}) + (\frac{n_{1,\star}}{n} - \sum_{a=1}^{A-1} \lambda_a) \nabla L_{1,0}(\boldsymbol{w}_{\boldsymbol{\lambda}}) + \frac{n_{0,\star}}{n} \nabla L_{0,\star}(\boldsymbol{w}_{\boldsymbol{\lambda}}) = 0.
$$

We take the $\lambda_j$ derivative to the above equation and have

$$
\Big[ \sum_{a=1}^{A-1} \lambda_a \nabla^2 L_{1,a}(\boldsymbol{w}_{\boldsymbol{\lambda}}) + (\frac{n_{1,\star}}{n} - \sum_{a=1}^{A-1} \lambda_a) \nabla^2 L_{1,0}(\boldsymbol{w}_{\boldsymbol{\lambda}}) + \frac{n_{0,\star}}{n} \nabla^2 L_{0,\star}(\boldsymbol{w}_{\boldsymbol{\lambda}}) \Big]
$$

$$
\nabla L_{1,j}(\boldsymbol{w}_{\boldsymbol{\lambda}}) - \nabla L_{1,0}(\boldsymbol{w}_{\boldsymbol{\lambda}}) + \frac{\partial \boldsymbol{w}_{\boldsymbol{\lambda}}}{\partial \lambda_j} = 0.
$$

Thus we get

$$
\frac{\partial \boldsymbol{w}_{\boldsymbol{\lambda}}}{\partial \lambda_j} = \Big[ \sum_{a=1}^{A-1} \lambda_a \nabla^2 L_{1,a}(\boldsymbol{w}_{\boldsymbol{\lambda}}) + (\frac{n_{1,\star}}{n} - \sum_{a=1}^{A-1} \lambda_a) \nabla^2 L_{1,0}(\boldsymbol{w}_{\boldsymbol{\lambda}}) + \frac{n_{0,\star}}{n} \nabla^2 L_{0,\star}(\boldsymbol{w}_{\boldsymbol{\lambda}}) \Big]^{-1}
$$

$$
[\nabla L_{1,0}(\boldsymbol{w}_{\boldsymbol{\lambda}}) - \nabla L_{1,j}(\boldsymbol{w}_{\boldsymbol{\lambda}})]. \tag{25}
$$

Then on the direction $\boldsymbol{\mu}(\boldsymbol{\lambda})$ given by (23), we combine (24) and (25) to have

$$\boldsymbol{\mu}(\boldsymbol{\lambda}) \cdot \nabla F_{\mathrm{eo}}(\boldsymbol{\lambda})$$

$$= 2\Big[ \sum_{a=1}^{A-1} (L_{1,a}(\boldsymbol{w_\lambda}) - L_{1,0}(\boldsymbol{w_\lambda}))(\nabla L_{1,a}(\boldsymbol{w_\lambda}) - \nabla L_{1,0}(\boldsymbol{w_\lambda}))\Big]$$

$$\Big[ \sum_{a=1}^{A-1} \lambda_a \nabla^2 L_{1,a}(\boldsymbol{w_\lambda}) + (\frac{n_{1,\star}}{n} - \sum_{a=1}^{A-1} \lambda_a)\nabla^2 L_{1,0}(\boldsymbol{w_\lambda}) + \frac{n_{0,\star}}{n}\nabla^2 L_{0,\star}(\boldsymbol{w_\lambda})\Big]^{-1}$$

$$\Big[ \sum_{a=1}^{A-1} (L_{1,a}(\boldsymbol{w_\lambda}) - L_{1,0}(\boldsymbol{w_\lambda}))(\nabla L_{1,0}(\boldsymbol{w_\lambda}) - \nabla L_{1,a}(\boldsymbol{w_\lambda}))\Big] \leq 0,$$

where we have used Assumption 6, the equality holds only when $\boldsymbol{\mu}(\boldsymbol{\lambda}) = \boldsymbol{0}$.

$\square$

We present the FEDFB algorithm *w.r.t* equal opportunity in Algorithm 3.

---

**Algorithm 3:** FEDFB *w.r.t* Equal Opportunity

---

**Server executes:**
    **input** : Learning rate $\{\alpha_t\}_{t\in\mathbb{N}}$;
    Initialize $\lambda_a$ as $\frac{n_{1,a}}{n}$ for all $a \in [A] \setminus \{0\}$;
    **for** *each iteration* $t = 1, 2, \ldots$ **do**
        Clients perform updates;
        $\boldsymbol{w_\lambda} \leftarrow \mathrm{SecAgg}\left\{\boldsymbol{w}^{(i)}\right\}$ for all $i$;
        $L_{y,a} \leftarrow \mathrm{SecAgg}\left\{L_{y,a}^{(i)}\right\}$ for all $y \in \{0,1\}, a \in [A] \setminus \{0\}$;
        $\mu_a \leftarrow L_{1,a} - L_{1,0}, a \in [A] \setminus \{0\}$;
        $\lambda_a \leftarrow \lambda_a + \frac{\alpha_t}{\|\boldsymbol{\mu}\|_2}\mu_a$, for all $a \in [A] \setminus \{0\}$;
        Broadcast $\boldsymbol{w_\lambda}$ and $\boldsymbol{\lambda}$ to clients;
    **end**
    **output** : $w_\lambda$
**ClientUpdate**$(i, \boldsymbol{w}, \boldsymbol{\lambda})$**:**
    $\boldsymbol{w}^{(i)} \leftarrow$ Gradient descent *w.r.t* objective function
    $\sum_{a=1}^{A-1} \lambda_a L_{1,a}^{(i)}(\boldsymbol{w}) + (\frac{n_{1,\star}}{n} - \sum_{a=1}^{A-1} \lambda_a)L_{1,0}^{(i)}(\boldsymbol{w}) + \frac{n_{0,\star}}{n}L_{0,\star}^{(i)}(\boldsymbol{w})$;
    Send $\boldsymbol{w}^{(i)}, L_{0,a}^{(i)}(\boldsymbol{w}), L_{1,a}^{(i)}(\boldsymbol{w})$ for all $a \in [A]$ to server via a SecAgg protocol;

---

### B.3 FEDFB *w.r.t* EQUALIZED ODDS

For equalized odd, we design the following bi-level optimization problem:

$$\min_{\boldsymbol{\lambda}\in\Lambda} F_{\mathrm{eod}}(\boldsymbol{\lambda}) = \min_{\boldsymbol{\lambda}\in\Lambda} \sum_{a=1}^{A-1} \left[(L_{1,a}(\boldsymbol{w_\lambda}) - L_{1,0}(\boldsymbol{w_\lambda}))^2 + (L_{0,a}(\boldsymbol{w_\lambda}) - L_{0,0}(\boldsymbol{w_\lambda}))^2\right]$$

$$\boldsymbol{w_\lambda} = \arg\min_{\boldsymbol{w}} \sum_{a=1}^{A-1} (\lambda_{0,a}L_{0,a}(\boldsymbol{w}) + \lambda_{1,a}L_{1,a}(\boldsymbol{w}))$$

$$+ (\frac{n_{0,\star}}{n} - \sum_{a=1}^{A-1} \lambda_{0,a})L_{0,0}(\boldsymbol{w}) + (\frac{n_{1,\star}}{n} - \sum_{a=1}^{A-1} \lambda_{1,a})L_{1,0}(\boldsymbol{w}). \tag{26}$$

Here

$$\Lambda = \{(\lambda_{0,1}, \ldots, \lambda_{0,A-1}, \lambda_{1,1}, \ldots, \lambda_{1,A-1}) : \sum_{a=1}^{A-1} \lambda_{0,a} \leq \frac{n_{0,\star}}{n}, \quad \sum_{a=1}^{A-1} \lambda_{1,a} \leq \frac{n_{1,\star}}{n},$$

$$\lambda_{0,a}, \lambda_{1,a} \geq 0, \quad \text{for all } a = 1, \ldots, A-1\}.$$

For equalized odds, we make the following assumption:

**Assumption 7.** *$L_{y,a}(\cdot)$ is twice differentiable for all $y \in \{0,1\}$, $a \in [A]$, and*

$$\sum_{a=1}^{A-1} \Big[ (\lambda_{1,a}\nabla^2 L_{1,a}(\boldsymbol{w_\lambda}) + \lambda_{0,a}\nabla^2 L_{0,a}(\boldsymbol{w_\lambda}) + (\frac{n_{0,\star}}{n} - \sum_{a=1}^{A-1}\lambda_{0,a})\nabla^2 L_{0,0}(\boldsymbol{w_\lambda})$$

$$+ (\frac{n_{1,\star}}{n} - \sum_{a=1}^{A-1}\lambda_{1,a})\nabla^2 L_{1,0}(\boldsymbol{w_\lambda})) \Big] \succ 0$$

*for all $\lambda \in \Lambda$.*

With the above assumption, the following lemma provides the update rule:

**Lemma 25.** *If Assumption 7 holds, then on the direction*

$$\boldsymbol{\mu}(\boldsymbol{\lambda}) = (\mu_{0,1}(\boldsymbol{\lambda}), \ldots, \mu_{0,A-1}(\boldsymbol{\lambda}), \mu_{1,1}(\boldsymbol{\lambda}), \ldots, \mu_{1,A-1}(\boldsymbol{\lambda})), \text{ with}$$

$$\mu_{y,a}(\boldsymbol{\lambda}) = L_{y,a}(\boldsymbol{w_\lambda}) - L_{y,0}(\boldsymbol{w_\lambda}), \quad y \in \{0,1\}, \quad a \in [A], \tag{27}$$

*we have $\boldsymbol{\mu}(\boldsymbol{\lambda}) \cdot \nabla F_{\mathrm{eod}}(\boldsymbol{\lambda}) \leq 0$, and the equality holds if only if $\boldsymbol{\mu}(\boldsymbol{\lambda}) = \boldsymbol{0}$.*

**Update rule for equalized odd:**

$$\lambda_{y,a}^{(t+1)} = \lambda_{y,a}^{(t)} + \frac{\alpha_t}{\|\boldsymbol{\mu}(\boldsymbol{\lambda}^{(t)})\|_2}(L_{y,a}(\boldsymbol{w}_{\boldsymbol{\lambda}^{(t)}}) - L_{y,0}(\boldsymbol{w}_{\boldsymbol{\lambda}^{(t)}})) \text{ for } y \in \{0,1\}, a \in [A].$$

*Proof of Lemma 25.* We compute the derivative as

$$\frac{\partial F_{\mathrm{eod}}(\boldsymbol{\lambda})}{\partial \lambda_{y,j}} = 2\sum_{a=1}^{A-1} \Big[ (L_{1,a}(\boldsymbol{w_\lambda}) - L_{1,0}(\boldsymbol{w_\lambda}))(\nabla L_{1,a}(\boldsymbol{w_\lambda}) - \nabla L_{1,0}(\boldsymbol{w_\lambda}))$$

$$+ (L_{0,a}(\boldsymbol{w_\lambda}) - L_{0,0}(\boldsymbol{w_\lambda}))(\nabla L_{0,a}(\boldsymbol{w_\lambda}) - \nabla L_{0,0}(\boldsymbol{w_\lambda})) \Big] \frac{\partial \boldsymbol{w_\lambda}}{\partial \lambda_{y,j}}. \tag{28}$$

Note that $\boldsymbol{w_\lambda}$ is the minimizer to (26), we have

$$\sum_{a=1}^{A-1} (\lambda_{0,a}\nabla L_{0,a}(\boldsymbol{w_\lambda}) + \lambda_{1,a}\nabla L_{1,a}(\boldsymbol{w_\lambda}))$$

$$+ (\frac{n_{0,\star}}{n} - \sum_{a=1}^{A-1}\lambda_{0,a})\nabla L_{0,0}(\boldsymbol{w_\lambda}) + (\frac{n_{1,\star}}{n} - \sum_{a=1}^{A-1}\lambda_{1,a})\nabla L_{1,0}(\boldsymbol{w_\lambda}) = 0.$$

We take the $\lambda_{y,j}$ derivative to the above equation and have

$$\nabla L_{y,j}(\boldsymbol{w_\lambda}) - \nabla L_{y,0}(\boldsymbol{w_\lambda}) + \Big[ \sum_{a=1}^{A-1} (\lambda_{0,a}\nabla^2 L_{0,a}(\boldsymbol{w_\lambda}) + \lambda_{1,a}\nabla^2 L_{1,a}(\boldsymbol{w_\lambda}))$$

$$+ (\frac{n_{0,\star}}{n} - \sum_{a=1}^{A-1}\lambda_{0,a})\nabla^2 L_{0,0}(\boldsymbol{w_\lambda}) + (\frac{n_{1,\star}}{n} - \sum_{a=1}^{A-1}\lambda_{1,a})\nabla^2 L_{1,0}(\boldsymbol{w_\lambda}) \Big] \frac{\partial \boldsymbol{w_\lambda}}{\partial \lambda_{y,j}} = 0.$$

Thus we get

$$\frac{\partial \boldsymbol{w_\lambda}}{\partial \lambda_{y,j}} = \Big[ \sum_{a=1}^{A-1} (\lambda_{0,a}\nabla^2 L_{0,a}(\boldsymbol{w_\lambda}) + \lambda_{1,a}\nabla^2 L_{1,a}(\boldsymbol{w_\lambda}))$$

$$+ (\frac{n_{0,\star}}{n} - \sum_{a=1}^{A-1}\lambda_{0,a})\nabla^2 L_{0,0}(\boldsymbol{w_\lambda}) + (\frac{n_{1,\star}}{n} - \sum_{a=1}^{A-1}\lambda_{1,a})\nabla^2 L_{1,0}(\boldsymbol{w_\lambda}) \Big]^{-1}$$

$$[\nabla L_{y,0}(\boldsymbol{w_\lambda}) - \nabla L_{y,j}(\boldsymbol{w_\lambda})]. \tag{29}$$

Then on the direction $\boldsymbol{\mu}(\boldsymbol{\lambda})$ given by (27), we combine (28) and (29) to have

$$\boldsymbol{\mu}(\boldsymbol{\lambda}) \cdot \nabla F_{\text{eod}}(\boldsymbol{\lambda})$$

$$= 2\Big[ \sum_{a=1}^{A-1} [(L_{1,a}(\boldsymbol{w}_{\boldsymbol{\lambda}}) - L_{1,0}(\boldsymbol{w}_{\boldsymbol{\lambda}}))(\nabla L_{1,a}(\boldsymbol{w}_{\boldsymbol{\lambda}}) - \nabla L_{1,0}(\boldsymbol{w}_{\boldsymbol{\lambda}}))$$

$$+ (L_{0,a}(\boldsymbol{w}_{\boldsymbol{\lambda}}) - L_{0,0}(\boldsymbol{w}_{\boldsymbol{\lambda}}))(\nabla L_{0,a}(\boldsymbol{w}_{\boldsymbol{\lambda}}) - \nabla L_{0,0}(\boldsymbol{w}_{\boldsymbol{\lambda}}))]\Big]$$

$$\Big[ \sum_{a=1}^{A-1} (\lambda_{0,a} \nabla^2 L_{0,a}(\boldsymbol{w}_{\boldsymbol{\lambda}}) + \lambda_{1,a} \nabla^2 L_{1,a}(\boldsymbol{w}_{\boldsymbol{\lambda}}))$$

$$+ (\frac{n_{0,\star}}{n} - \sum_{a=1}^{A-1} \lambda_{0,a}) \nabla^2 L_{0,0}(\boldsymbol{w}_{\boldsymbol{\lambda}}) + (\frac{n_{1,\star}}{n} - \sum_{a=1}^{A-1} \lambda_{1,a}) \nabla^2 L_{1,0}(\boldsymbol{w}_{\boldsymbol{\lambda}})\Big]^{-1}$$

$$\Big[ \sum_{a=1}^{A-1} [(L_{1,a}(\boldsymbol{w}_{\boldsymbol{\lambda}}) - L_{1,0}(\boldsymbol{w}_{\boldsymbol{\lambda}}))(\nabla L_{1,0}(\boldsymbol{w}_{\boldsymbol{\lambda}}) - \nabla L_{1,a}(\boldsymbol{w}_{\boldsymbol{\lambda}}))$$

$$+ (L_{0,a}(\boldsymbol{w}_{\boldsymbol{\lambda}}) - L_{0,0}(\boldsymbol{w}_{\boldsymbol{\lambda}}))(\nabla L_{0,0}(\boldsymbol{w}_{\boldsymbol{\lambda}}) - \nabla L_{0,a}(\boldsymbol{w}_{\boldsymbol{\lambda}}))]\Big] \leq 0$$

where we have used Assumption 7, and the equality holds only when $\boldsymbol{\mu}(\boldsymbol{\lambda}) = \boldsymbol{0}$.

$\square$

The full procedure is described in Algorithm 4.

---

**Algorithm 4:** FEDFB *w.r.t* Equalized Odds

---

**Server executes:**
    **input** :Learning rate $\{\alpha_t\}_{t \in \mathbb{N}}$;
    Initialize $\lambda_{y,a}$ as $\frac{n_{y,a}}{n}$ for all $y \in \{0, 1\}, a \in [A] \backslash \{0\}$;
    **for** *each iteration* $t = 1, 2, \ldots$ **do**
        Clients perform updates;
        $\boldsymbol{w}_{\boldsymbol{\lambda}} \leftarrow \text{SecAgg} \{\boldsymbol{w}^{(i)}\}$ for all $i$;
        $L_{y,a} \leftarrow \text{SecAgg} \{L_{y,a}^{(i)}\}$ for all $y \in \{0, 1\}, a \in [A] \backslash \{0\}$;
        $\mu_{y,a} \leftarrow L_{y,a} - L_{y,0}$ for all $y \in \{0, 1\}, a \in [A] \backslash \{0\}$;
        $\lambda_{y,a} \leftarrow \lambda_{y,a} + \frac{\alpha_t}{\|\boldsymbol{\mu}\|_2} \mu_{y,a}$, for all $a \in [A] \backslash \{0\}$;
        Broadcast $\boldsymbol{w}_{\boldsymbol{\lambda}}$ and $\boldsymbol{\lambda}$ to clients;
    **end**
    **output** :$w_{\boldsymbol{\lambda}}$
**ClientUpdate**$(i, \boldsymbol{w}, \boldsymbol{\lambda})$:
    $\boldsymbol{w}^{(i)} \leftarrow$ Gradient descent *w.r.t* objective function $\sum_{a=1}^{A-1} \left( \lambda_{0,a} L_{0,a}^{(i)}(\boldsymbol{w}) + \lambda_{1,a} L_{1,a}^{(i)}(\boldsymbol{w}) \right) +$
    $(\frac{n_{0,\star}}{n} - \sum_{a=1}^{A-1} \lambda_{0,a}) L_{0,0}^{(i)}(\boldsymbol{w}) + (\frac{n_{1,\star}}{n} - \sum_{a=1}^{A-1} \lambda_{1,a}) L_{1,0}^{(i)}(\boldsymbol{w})$;
    Send $\boldsymbol{w}^{(i)}, L_{0,a}^{(i)}(\boldsymbol{w}), L_{1,a}^{(i)}(\boldsymbol{w})$ for all $a \in [A]$ to server via a SecAgg protocol;

---

### B.4 FEDFB *w.r.t* CLIENT PARITY

For client parity, we slightly abuse the notation and define the loss over client $i$ as $L^{(i)}(\boldsymbol{w}) := \sum_{i_k=i} \ell(y_k, \hat{y}_k)/n^{(i)}$, with $n^{(i)} = |\{k : i_k = i\}|$. Note that the $L^{(i)}(\boldsymbol{w})$ here is different from the one in Sec. B.1. We design the following bi-level optimization problem:

$$\min_{\boldsymbol{\lambda} \in \Lambda} F_{\text{cp}}(\boldsymbol{\lambda}) = \min_{\boldsymbol{\lambda} \in \Lambda} \sum_{i=1}^{I-1} \left( L^{(i)}(\boldsymbol{w}_{\boldsymbol{\lambda}}) - L^{(0)}(\boldsymbol{w}_{\boldsymbol{\lambda}}) \right)^2,$$

$$\boldsymbol{w}_{\boldsymbol{\lambda}} = \arg\min_{\boldsymbol{w}} \sum_{i=1}^{I-1} \lambda^{(i)} L^{(i)}(\boldsymbol{w}) + (1 - \sum_{i=1}^{I-1} \lambda^{(i)}) L^{(0)}(\boldsymbol{w}). \tag{30}$$

Here

$$\Lambda = \{(\lambda^{(1)}, \ldots, \lambda^{(I-1)}) : 0 \leq \lambda^{(i)} \leq 1, \quad \sum_{i=1}^{I-1} \lambda^{(i)} \leq 1\}.$$

For client parity, we make the following assumption:

**Assumption 8.** $L^{(i)}(\cdot)$ *is twice differentiable for all* $i = 1, \ldots, I-1$*, and*

$$\sum_{i=1}^{I-1} \lambda^{(i)} \nabla^2 L^{(i)}(\boldsymbol{w}_{\boldsymbol{\lambda}}) + (1 - \sum_{i=1}^{I-1} \lambda^{(i)}) \nabla^2 L^{(0)}(\boldsymbol{w}_{\boldsymbol{\lambda}}) \succ 0.$$

With the above assumption, the following lemma provides the update rule:

**Lemma 26.** *If Assumption 8 holds, then on the direction*

$$\boldsymbol{\mu}(\boldsymbol{\lambda}) = (L^{(1)}(\boldsymbol{w}_{\boldsymbol{\lambda}}) - L^{(0)}(\boldsymbol{w}_{\boldsymbol{\lambda}}), \ldots, L^{(I-1)}(\boldsymbol{w}_{\boldsymbol{\lambda}}) - L^{(0)}(\boldsymbol{w}_{\boldsymbol{\lambda}})), \tag{31}$$

*we have* $\boldsymbol{\mu}(\boldsymbol{\lambda}) \cdot \nabla F_{\mathrm{cp}}(\boldsymbol{\lambda}) \leq 0$*, and the equality holds if and only if* $\boldsymbol{\mu}(\boldsymbol{\lambda}) = \boldsymbol{0}$*.*

Then we update $\boldsymbol{\lambda}^{(t)} = (\lambda^{(1)^{(t)}}, \ldots, \lambda^{(I-1)^{(t)}})$ as follows:

**Update rule for client parity:**

$$\lambda^{(i)^{(t+1)}} = \lambda^{(i)^{(t)}} + \frac{\alpha_t}{\|\boldsymbol{\mu}(\boldsymbol{\lambda}^{(t)})\|_2} (L^{(i)}(\boldsymbol{w}_{\boldsymbol{\lambda}^{(t)}}) - L^{(0)}(\boldsymbol{w}_{\boldsymbol{\lambda}^{(t)}})) \text{ for } i = 1, \ldots, I-1.$$

*Proof of Lemma 26.* We compute the derivative as

$$\frac{\partial F_{\mathrm{cp}}(\boldsymbol{\lambda})}{\partial \lambda^{(j)}} = \Big(2 \sum_{i=1}^{I-1} [L^{(i)}(\boldsymbol{w}_{\boldsymbol{\lambda}}) - L^{(0)}(\boldsymbol{w}_{\boldsymbol{\lambda}})][\nabla L^{(i)}(\boldsymbol{w}_{\boldsymbol{\lambda}}) - \nabla L^{(0)}(\boldsymbol{w}_{\boldsymbol{\lambda}})]\Big) \frac{\partial \boldsymbol{w}_{\boldsymbol{\lambda}}}{\partial \lambda^{(j)}}. \tag{32}$$

Note that $\boldsymbol{w}_{\boldsymbol{\lambda}}$ is the minimizer to (30), we have

$$\sum_{i=1}^{I-1} \lambda^{(i)} \nabla L^{(i)}(\boldsymbol{w}_{\boldsymbol{\lambda}}) + (1 - \sum_{i=1}^{I-1} \lambda^{(i)}) \nabla L^{(0)}(\boldsymbol{w}_{\boldsymbol{\lambda}}) = 0.$$

We take the $\lambda^{(j)}$ derivative to the above equation and have

$$\nabla L^{(j)}(\boldsymbol{w}_{\boldsymbol{\lambda}}) + \sum_{i=1}^{I-1} \lambda^{(i)} \nabla^2 L^{(i)}(\boldsymbol{w}_{\boldsymbol{\lambda}}) \frac{\partial \boldsymbol{w}_{\boldsymbol{\lambda}}}{\partial \lambda_j} - \nabla L^{(0)}(\boldsymbol{w}_{\boldsymbol{\lambda}}) + (1 - \sum_{i=1}^{I-1} \lambda^{(i)}) \nabla^2 L^{(0)}(\boldsymbol{w}_{\boldsymbol{\lambda}}) \frac{\partial \boldsymbol{w}_{\boldsymbol{\lambda}}}{\partial \lambda_j} = 0.$$

Thus we get

$$\frac{\partial \boldsymbol{w}_{\boldsymbol{\lambda}}}{\partial \lambda^{(j)}} = \Big[\sum_{i=1}^{I-1} \lambda^{(i)} \nabla^2 L^{(i)}(\boldsymbol{w}_{\boldsymbol{\lambda}}) + (1 - \sum_{i=1}^{I-1} \lambda^{(i)}) \nabla^2 L^{(0)}(\boldsymbol{w}_{\boldsymbol{\lambda}})\Big]^{-1} [\nabla L^{(0)}(\boldsymbol{w}_{\boldsymbol{\lambda}}) - \nabla L^{(j)}(\boldsymbol{w}_{\boldsymbol{\lambda}})]. \tag{33}$$

Then on the direction given by (31), we combine (32) and (33) to have

$$\boldsymbol{\mu}(\boldsymbol{\lambda}) \cdot \nabla F_{\mathrm{cp}}(\boldsymbol{\lambda})$$

$$= 2\Big(\sum_{i=1}^{I-1} [L^{(i)}(\boldsymbol{w}_{\boldsymbol{\lambda}}) - L^{(0)}(\boldsymbol{w}_{\boldsymbol{\lambda}})][\nabla L^{(i)}(\boldsymbol{w}_{\boldsymbol{\lambda}}) - \nabla L^{(0)}(\boldsymbol{w}_{\boldsymbol{\lambda}})]\Big)$$

$$\Big[\sum_{i=1}^{I-1} \lambda^{(i)} \nabla^2 L^{(i)}(\boldsymbol{w}_{\boldsymbol{\lambda}}) + (1 - \sum_{i=1}^{I-1} \lambda^{(i)}) \nabla^2 L^{(0)}(\boldsymbol{w}_{\boldsymbol{\lambda}})\Big]^{-1}$$

$$\sum_{i=1}^{I-1} \Big([L^{(i)}(\boldsymbol{w}_{\boldsymbol{\lambda}}) - L^{(0)}(\boldsymbol{w}_{\boldsymbol{\lambda}})][\nabla L^{(0)}(\boldsymbol{w}_{\boldsymbol{\lambda}}) - \nabla L^{(i)}(\boldsymbol{w}_{\boldsymbol{\lambda}})]\Big) \leq 0,$$

where we have used Assumption 8, and the equality holds only when $\boldsymbol{\mu}(\boldsymbol{\lambda}) = \boldsymbol{0}$.

$\square$

The Algorithm 5 gives the full description.

---

**Algorithm 5:** FEDFB *w.r.t* Client Parity

---

**Server executes:**

    **input** :Learning rate $\{\alpha_t\}_{t\in\mathbb{N}}$;

    Initialize $\lambda^{(i)}$ as $\frac{n^{(i)}}{n}$ for all $i \in [I]\setminus\{0\}$;

    **for** *each iteration* $t = 1, 2, \ldots$ **do**

        Clients perform updates;

        $\boldsymbol{w_\lambda} \leftarrow \text{SecAgg}\left\{\boldsymbol{w}^{(i)}\right\}$ for all $i$;

        $\mu^{(i)} \leftarrow L^{(i)} - L^{(0)}, i \in [I]\setminus\{0\}$;

        $\lambda^{(i)} \leftarrow \lambda^{(i)} + \frac{\alpha_t}{\|\boldsymbol{\mu}\|_2}\mu^{(i)}$, for all $i \in [I]\setminus\{0\}$;

        Broadcast $\boldsymbol{w_\lambda}$ and $\boldsymbol{\lambda}$ to clients;

    **end**

    **output** : $w_\lambda$

**ClientUpdate**$(i, \boldsymbol{w}, \boldsymbol{\lambda})$**:**

    $\boldsymbol{w}^{(i)} \leftarrow$ Gradient descent *w.r.t* objective function

    $[\![i \neq 0]\!]\lambda^{(i)}L^{(i)}(\boldsymbol{w}) + [\![i = 0]\!](1 - \sum_{j=1}^{I-1}\lambda^{(ij)})L^{(0)}(\boldsymbol{w})$;

    Send $\boldsymbol{w}^{(i)}$ to server via a SecAgg protocol; Send $L^{(i)}(\boldsymbol{w})$ to server;

---

## C  APPENDIX - EXPERIMENTS

We continue from Sec. 5 and provide more details on experimental settings, and other supplementary experiment results.

### C.1  EXPERIMENT SETTING

The reported statistics are computed on the test set, and we set 10 communication rounds and 30 local epochs for all federated learning algorithms except AGNOSTICFAIR. We run 300 epochs for other methods. For all tasks, we randomly split data into a training set and a testing set at a ratio of 7:3. The batch size is set to be 128. For all methods, we choose learning rate froom $\{0.001, 0.005, 0.01\}$. We solve FEDFB with sample weight learning rates $\alpha \in \{0.001, 0.05, 0.08, 0.1, 0.2, 0.5, 1, 2\}$ in parallel and select the $\alpha$ which achieves the highest fairness. All benchmark models are tuned according to the hyperparameter configuration suggested in their original works. We perform cross-validation on the training sets to find the best hyperparameter for all algorithms.

### C.2  SYNTHETIC DATASET GENERATION

We generate a synthetic dataset of 5,000 examples with two non-sensitive attributes $(x_1, x_2)$, a binary sensitive $a$, and a binary label $y$. A tuple $(x_1, x_2, y)$ is randomly generated based on the two Gaussian distributions: $(x_1, x_2) \mid y \sim \mathcal{N}([-2; -2], [10, 1; 1, 3])$ and $(x_1, x_2) \mid y = 1 \sim \mathcal{N}([2; 2], [5, 1])$, where $y \sim \text{Bern}(0.6)$. For the sensitive attribute $a$, we generate biased data using an unfair scenario $p_{x_1, x_2}((x_1', x_2') \mid y = 1)/[p_{x_1, x_2}((x_1', x_2') \mid y = 0) + p_{x_1, x_2}((x_1', x_2') \mid y = 1)]$, where $p_{x_1, x_2}$ is the pdf of $(x_1, x_2)$. We split the data into three clients in a non-iid way. We randomly assign $50\%, 30\%, 20\%$ of the samples from group 0 and $20\%, 40\%, 40\%$ of the samples from group 1 to 1st, 2nd, and 3rd client, respectively. To study the empirical relationship between the performance of FEDFB and data heterogeneity, we split the dataset into other ratios to obtain the desired level of data heterogeneity. To get the dataset with low data heterogeneity, we draw samples from each group into three clients in a ratio of $33\%, 33\%, 34\%$ and $33\%, 33\%, 34\%$. For dataset with high data heterogeneity, the ratio we choose is $70\%, 10\%, 20\%$ and $10\%, 80\%, 10\%$.

### C.3  SUPPLEMENTARY EXPERIMENT RESULTS

Fig. 8 shows the accuracy versus fairness violation for the 3 clients' cases. We see a clear advantage of FFL via FEDAVG over the UFL in both 2 clients' cases and 3 clients' cases. The achievable fairness range of FFL via FEDAVG is much wider than that of UFL, though the accuracy of FEDAVG is not guaranteed when the data heterogeneity increases.

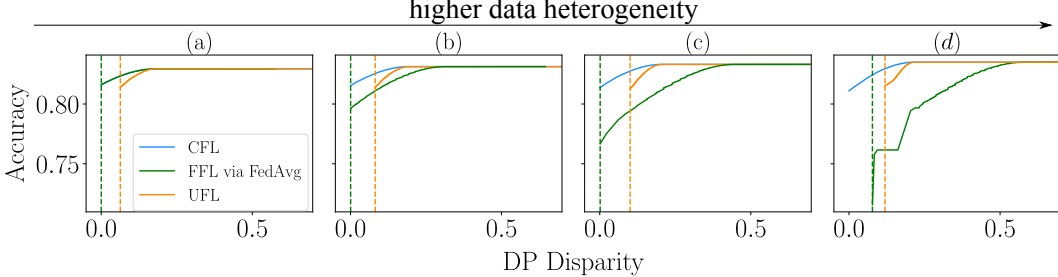

**Figure 8: Accuracy-Fairness tradeoff curves of CFL, FFL via FedAvg, and UFL for three clients cases.** The data heterogeneity is increasing from left to right. The green dotted vertical line describes the lower bound of unfairness FFL via FedAvg can achieve, and and the orange dotted vertical line describes the lower bound of unfairness UFL can achieve. Here the distribution setting is $x \mid a = 0, i = 0 \sim \mathcal{N}(3,1), x \mid a = 1, i = 0 \sim \mathcal{N}(5,1), x \mid a = 0, i = 1 \sim \mathcal{N}(1,1), x \mid a = 1, i = 1 \sim \mathcal{N}(-1,1), x \mid a = 0, i = 2 \sim \mathcal{N}(1,1), x \mid a = 1, i = 2 \sim \mathcal{N}(2,1), a \mid i = i \sim \text{Bern}(q_i)$ for $i = 0, 1, 2$. The data heterogeneity here is captured by $|q_2 - q_0|$. (a) $q_0 = q_1 = q_2 = 0.5$. (b) $q_0 = 0.4, q_1 = 0.5, q_2 = 0.6$. (c): $q_0 = 0.3, q_1 = 0.5, q_2 = 0.7$. (d): $q_0 = 0.2, q_1 = 0.5, q_2 = 0.8$.

**Table 4: Comparison of accuracy and fairness in the synthetic, Adult, COMPAS, and Bank datasets *w.r.t* demographic parity (DP) on logistic regression.** The implementation of UFL, FFL via FedAvg, and CFL are all based on FB.

| | SYNTHETIC | | ADULT | | COMPAS | | BANK | |
|---|---|---|---|---|---|---|---|---|
| METHOD | ACC.(↑) | DP DISP.(↓) | ACC.(↑) | DP DISP.(↓) | ACC.(↑) | DP DISP.(↓) | ACC.(↑) | DP DISP.(↓) |
| FEDAVG | .884±.001 | .419±.006 | .837±.007 | .144±.015 | .658±.006 | .149±.022 | .900±.000 | .026±.001 |
| UFL | .712±.198 | .266±.175 | .819±006. | .032±.031 | .606±.018 | .089±.058 | .888±.005 | .008±.007 |
| FFL VIA FEDAVG | .789±.138 | .301±.192 | .828±.002 | .098±.008 | .560±.000 | .030±.002 | .892±.000 | .013±.000 |
| FEDFB | .756±.001 | .085±.001 | .820±.000 | **.002±.001** | .550±.000 | **.009±.000** | .890±.001 | .011±.002 |
| AGNOSTICFAIR | .622±.051 | **.028±.008** | .768±.000 | .003±.000 | .568±.018 | .034±.023 | .883±.000 | **.000±.000** |
| CFL | .662±.039 | .077±.020 | .810±.009 | .054±.027 | .587±.003 | .032±.002 | .883±.000 | **.000±.000** |

We also compare FEDAVG, UFL, FFL via FEDAVG, FEDFB, AGNOSTICFAIR and CFL on logistic regression. Table 4 shows that our method outperforms UFL and FFL via FEDAVG, while achieving similar performance as AGNOSTICFAIR and CFL but ensuring higher privacy.

## C.4 EVALUATION OF DIFFERENTIALLY PRIVATE FEDFB

Since FEDFB exchanges more information than FFL via FEDAVG, we employ differential privacy to decrease information leakage. We apply the Laplace mechanism to make the information exchanged in each communication round $\varepsilon$-differentially private. We report the test accuracy and DP disparity of FFL via FEDAVG and FEDFB with different differential privacy levels in Table 5. Interestingly, we observe that a higher level of differential privacy even helps to improve fairness (see FFL via FEDAVG column), though the accuracy is decreased. This phenomenon is to be expected since larger noise helps to protect sensitive information, thereby improving fairness and lower accuracy. Still, Table 5 implies that FEDFB still outperforms FFL via FEDAVG with restrictions on the information exchange.

## C.5 EMPIRICAL RELATIONSHIP BETWEEN ACCURACY, FAIRNESS, AND THE NUMBER OF CLIENTS

We investigate the empirical relationship between accuracy, fairness, and the number of clients. We generate three synthetic datasets of 3,333, 5,000, and 6,667 samples, and split them into two, three, and four clients, respectively. Table 6 shows that our method outperforms under the three cases.

**Table 5:** Performance comparison under $\varepsilon$-differential private information exchange in each communication round on synthetic dataset.

| $\varepsilon$ | FFL via FedAvg | | FedFB | |
|---|---|---|---|---|
| | Acc.($\uparrow$) | DP Disp.($\downarrow$) | Acc.($\uparrow$) | DP Disp.($\downarrow$) |
| 0.1 | .583$\pm$.258 | .185$\pm$.254 | .517$\pm$.180 | .086$\pm$.192 |
| 1 | .582$\pm$.261 | .189$\pm$.249 | .508$\pm$.191 | .103$\pm$.188 |
| 10 | .645$\pm$.315 | .373$\pm$.217 | .616$\pm$.169 | .091$\pm$.204 |

**Table 6: Comparison of accuracy and fairness in the synthetic datasets with different numbers of clients *w.r.t* demographic parity (DP).** The implementation of UFL, FFL via FEDAVG, and CFL are all based on FB.

| Method | TWO CLIENTS | | THREE CLIENTS | | FOUR CLIENTS | |
|---|---|---|---|---|---|---|
| | Acc.($\uparrow$) | DP Disp.($\downarrow$) | Acc.($\uparrow$) | DP Disp.($\downarrow$) | Acc.($\uparrow$) | DP Disp.($\downarrow$) |
| FEDAVG | .879$\pm$.004 | .360$\pm$.013 | .883$\pm$.003 | .402$\pm$.018 | .879$\pm$.003 | .382$\pm$.005 |
| UFL | .780$\pm$.090 | .161$\pm$.157 | .729$\pm$.195 | .256$\pm$.193 | .720$\pm$.192 | .246$\pm$.180 |
| FFL VIA FEDAVG | .866$\pm$.002 | .429$\pm$.017 | .746$\pm$.307 | .478$\pm$.079 | .608$\pm$.367 | .491$\pm$.102 |
| FEDFB | .679$\pm$.007 | **.047$\pm$.012** | .613$\pm$.007 | **.011$\pm$.009** | .705$\pm$.004 | **.005$\pm$.003** |
| CFL | .668$\pm$.028 | .063$\pm$.035 | .693$\pm$.030 | .051$\pm$.020 | .670$\pm$.035 | .064$\pm$.020 |

## C.6 COMPARISON WITH FAIRFED

**Table 7:** Comparison of accuracy and fairness in the synthetic and Adult dataset *w.r.t* demographic parity (DP) on logistic regression.

| Method | SYNTHETIC | | ADULT | |
|---|---|---|---|---|
| | Acc.($\uparrow$) | DP Disp.($\downarrow$) | Acc.($\uparrow$) | DP Disp.($\downarrow$) |
| FAIRFED | .509$\pm$.092 | .092$\pm$.127 | .757$\pm$.003 | .105$\pm$.009 |
| **FEDFB (OURS)** | .756$\pm$.001 | **.085$\pm$.001** | .820$\pm$.000 | **.002$\pm$.001** |

As suggested in Ezzeldin et al. (2021), we use logistic regression to compare our approach with FAIRFED. Since FAIRFED is only applicable to single binary sensitive attribute cases, we report the performance of FAIRFED on synthetic and Adult datasets in Table 7. We observe that our FEDFB outperforms FAIRFED in terms of both accuracy and fairness, while FAIRFED also outperforms FFL via FEDAVG thanks to the additional client reweighting step.

**Table 8: Comparison of accuracy and fairness under the same setting as Ezzeldin et al. (2021).** The statistics of FAIRFED are from Ezzeldin et al. (2021).

| | Method | ADULT | | | | | COMPAS | | | | |
|---|---|---|---|---|---|---|---|---|---|---|---|
| | | HETEROGENEITY LEVEL $\alpha$ | | | | | HETEROGENEITY LEVEL $\alpha$ | | | | |
| | | 0.1 | 0.2 | 0.5 | 10 | 5000 | 0.1 | 0.2 | 0.5 | 10 | 5000 |
| Acc.($\uparrow$) | FAIRFED | .775 | .794 | .819 | .824 | .824 | .594 | .586 | .608 | .636 | .640 |
| | **FEDFB (OURS)** | .764 | .761 | .762 | .764 | .759 | .668 | .655 | .541 | .666 | .666 |
| $|SPD|$($\downarrow$) | FAIRFED | .021 | .037 | .061 | .065 | .065 | .048 | .040 | .072 | .108 | .115 |
| | **FEDFB (OURS)** | **.003** | **.003** | **.003** | **.000** | **.009** | **.009** | **.017** | **.000** | **.050** | **.006** |

To make fairer comparison, we follow the exact same setting as Ezzeldin et al. (2021) to re-split Adult and COMPAS dataset, employ $|SPD| = |\mathbb{P}(\hat{y} = 1 \mid a = 0) - \mathbb{P}(\hat{y} = 1 \mid a = 1)|$ as unfairness metric and present the results in Table 8. We observe that FEDFB achieves higher fairness than FAIRFED while being robust to data heterogeneity.

## C.7 COMPARISON WITH AGNOSTICFAIR

Lastly, we compare our method with AGNOSTICFAIR (Du et al., 2021), which exchanges the model parameters and the other information after every gradient update to mimic the performance of CFL with FAIRNESSCONSTRAINT implementation (Zafar et al., 2017c). Table 9 shows that our FEDFB achieves similar performance as AGNOSTICFAIR, at much lower cost of privacy.

**Table 9:** Comparison of accuracy and fairness in the synthetic, Adult, COMPAS and Bank datasets *w.r.t* demographic parity (DP) on multilayer perceptron.

| METHOD | SYNTHETIC | | ADULT | | COMPAS | | BANK | |
|---|---|---|---|---|---|---|---|---|
| | ACC.(↑) | DP DISP.(↓) | ACC.(↑) | DP DISP.(↓) | ACC.(↑) | DP DISP.(↓) | ACC.(↑) | DP DISP.(↓) |
| **FEDFB (OURS)** | .613±.007 | **.011±.009** | .765±.001 | **.001±.001** | .542±.001 | .001±.001 | .883±.000 | **.000±.000** |
| AGNOSTICFAIR | .657±.029 | .032±.044 | .767±.004 | .003±.005 | .541±.000 | **.000±.000** | .883±.000 | **.000±.000** |

