# OpenReview forum: "Improving Fairness via Federated Learning"
_ICLR.cc/2022/Conference — ICLR 2022 Submitted_

### Official Review · Reviewer_6XNC · 2021-10-21

**Correctness:** 4
**Technical Novelty And Significance:** 4
**Empirical Novelty And Significance:** 4
**Recommendation:** 6
**Confidence:** 4

**Main Review:**

Strength:
- The topic of fair federated learning is important and useful.
- This paper first gives a theoretical analysis of existing algorithms for fair federated learning. I enjoy reading this part.
- The empirical results are complete and confusing.

Question:
-

**Summary Of The Paper:**

This paper investigates how one can achieve group fairness under a decentralized setting. The authors develop a theoretical framework for decentralized fair learning algorithms and analyzed the performance of existing approaches including UFL, FFL via FedAvg, and CFL. They provide novel insights showing that UFL<FFL via FedAvg<CFL. They also propose a new federated fair learning algorithm FEDFB by letting each client share extra information about the unfairness of its local classifier with the server, which then computes the optimal samples weights for the following round of local training. The experimental results demonstrate that FEDFB achieves state-of-the-art performance, while still ensuring data privacy.


**Summary Of The Review:**

Overall, I recommend the paper. It considers an important problem, fair federated learning, analyzes and compares existing approaches, and proposes a convincing framework. The writing is nice and clear, especially the discussion in Section 3.

---

> ### Author Response · Authors · 2021-11-23
> **Thanks for your encouraging review!**
>
> Thank you so much for appreciating our result and for the positive scoring. We agree with your understanding of the main contributions of the paper.
>
> Besides, we are happy to update you that we have made more effort to strengthen our paper:
>
> 1. We developed a more private variant of FedFB, which only exchanges a small number of bits in each communication round. Our experiment results show that this new variant enjoys as good performance as plain FedFB. See Figure 5 on P8 and Figure 6 on P9 for more details.
>
> 2. We draw the accuracy-fairness tradeoff curves to better compare FedFB with other baseline methods in Figure 6 on P9.
>
> Overall, we are excited that you find our idea is important and useful. Thanks again for your careful reading of our work and for providing encouraging feedback!

---

### Official Review · Reviewer_pTFp · 2021-11-01

**Correctness:** 3
**Technical Novelty And Significance:** 2
**Empirical Novelty And Significance:** 2
**Recommendation:** 3
**Confidence:** 4

**Main Review:**

1. The 0-1 loss is a non-continuous function. Thus the most assumptions in B.1 are valid. For example, the convexity and twice differential assumptions. In addition, could you elaborate more on how to perform client updates in Algorithm 1? Given the noncontinuity of the 0-1 loss function, do you perform any approximation for the client updates? If there is no approximation, how can it be optimized? In addition, how can the DEMOGRAPHIC PARITY constraint be satisfied in FedFB?
2. The experiments in the paper are not extensive. The proposed methods have not been compared with some recent papers on this topic. For example,
a. Du W, Xu D, Wu X, Tong H. Fairness-aware Agnostic Federated Learning. InProceedings of the 2021 SIAM International Conference on Data Mining (SDM) 2021 (pp. 181-189). Society for Industrial and Applied Mathematics.
The authors use the data setting in the above paper but fail to compare it.
3. What is the data distribution among the clients in the experiments? It seems the authors use the iid setting across the clients and it is not a practical setting.
4. The notations in the main body and appendix are not consistent. For example, in the main body  $\cal A$ is used for the sensitivity attributes set but $[A]$ is used in the appendix. This type of inconsistency makes the paper really hard to follow. If I misunderstand something, please clarify these notations in the paper.
5. The authors may also discuss the privacy of the proposed method. Is there any increased risk of leaking the information from passing $L$ other than only transmitting model weights $w$.

**Summary Of The Paper:**

This paper tried to propose a generic federated learning method to train a fairness model. It compared the

**Summary Of The Review:**

1. The paper is not well-written and hard to follow. Some of the details are missing, which makes me hard to justify the claims.
2. The complete convergence analysis of the proposed algorithm is missing.
3. The experiments are not performed extensively. Thus, the claims in the paper are not fully supported.

---

> ### Author Response · Authors · 2021-11-23
> **Thank you for your valuable feedback. We have addressed all your concerns!**
>
> We appreciate the careful reading and thoughtful interchange, which aided us greatly in improving the clarity of our writing.
>
> --------------------------------------------------------------------------------------------------------------------
>
> Q. (paraphrased) 0-1 loss-related questions. 0-1 loss is not continuous -- how does FedFB compute the gradient and minimize the demographic disparity?
>
> A. We derive the update rules of the FedFB algorithm by analyzing the 0-1 loss case. However, when we actually apply the FedFB algorithm, we do not use the 0-1 loss-based demographic disparity. Instead, we use the “continuous surrogate” of it, which is the cross-entropy loss. We note that this design flow is identical to the design approach adopted in the original FB paper [1]. We clarified this in our revision -- please see Remark 10 in P29.
>
> --------------------------------------------------------------------------------------------------------------------
>
> Q. The proposed methods have not been compared with some recent papers on this topic.
>
> A. Thanks for a great comment. As per your suggestion, we implemented and evaluated the performance of AgnosticFair [2], the baseline algorithm suggested by the reviewer.  We corresponded with the author of the paper to get the official implementation.  We observed that AgnosticFair achieves almost identical performances with CFL + Fairness Constraints [3]. This is expected as it exchanges the model parameters and the other information after every gradient update, at the cost of low privacy.  Note that we also confirmed with the author of the AgnosticFair paper about our evaluation and interpretation.
>
> Furthermore, we also added the performance comparison with recent concurrent work [4].  The added experiments can be found at the end of the revised paper.
>
> --------------------------------------------------------------------------------------------------------------------
>
> Q. It seems the authors use iid setting across the clients and it is not a practical setting.
>
> A. We did not assume the iid settings across the clients, and we considered the heterogeneous data settings. For instance, in Table 2, we observed the performance trade-off as a function of data heterogeneity. Please let us know if there were any misleading statements in our paper -- we are happy to revise them in our final version.
>
> --------------------------------------------------------------------------------------------------------------------
>
> Q. Some notations are inconsistent. For example, we used $\mathcal{A}$ in the main body and used $[A]$ in the appendix.
>
> A. Fixed. Thanks for your careful reading.
>
> --------------------------------------------------------------------------------------------------------------------
>
> Q. Is there any increased risk of leaking the information from exchanging more information?
>
> A. Thanks for engaging so deeply and asking such a good question! The reviewer is correct that our FedFB algorithm leaks more information than the standard FedAvg protocol. To fully quantify the increased risk of privacy leakage, we need to employ the mathematical notion of privacy such as differential privacy. Doing so is one of our future research directions.
>
> Instead, we ran the following interesting experiment and added the results to our revised paper.  To limit the amount of additional information sent from the clients to the master, we evaluated the performance of a variant of FedFB, where clients only exchange the “quantized” loss values.  By setting the appropriate quantization levels, we showed that FedFB can still maintain the same performance while communicating only “a few more bits” in every communication round. See Sec. 4 in P7 for more details.
>
> --------------------------------------------------------------------------------------------------------------------
>
> We hope that our response and the commitment to (a) clarifying the logic flow of FedFB development and experiment setting, (b) adding extra recent related work to our experiments, and (c) developing a more private variant of FedFB make you consider increasing your score and support accepting our paper. Thanks again for your careful reading of our work and for providing valuable feedback!
>
> References:
>
> [1] Roh, Y., Lee, K., Whang, S. E., & Suh, C. (2020). Fairbatch: Batch selection for model fairness. arXiv preprint arXiv:2012.01696.
>
> [2] Du, W., Xu, D., Wu, X., & Tong, H. (2021). Fairness-aware Agnostic Federated Learning. In Proceedings of the 2021 SIAM International Conference on Data Mining (SDM) (pp. 181-189). Society for Industrial and Applied Mathematics.
>
> [3] Zafar, M. B., Valera, I., Rogriguez, M. G., & Gummadi, K. P. (2017, April). Fairness constraints: Mechanisms for fair classification. In Artificial Intelligence and Statistics (pp. 962-970). PMLR.
>
> [4] Ezzeldin, Y. H., Yan, S., He, C., Ferrara, E., & Avestimehr, S. (2021). FairFed: Enabling Group Fairness in Federated Learning. arXiv preprint arXiv:2110.00857.

---

### Official Review · Reviewer_q1zx · 2021-11-02

**Correctness:** 2
**Technical Novelty And Significance:** 1
**Empirical Novelty And Significance:** Not applicable
**Recommendation:** 5
**Confidence:** 4

**Main Review:**

The authors proposition appears to be novel: fairness in Federated Learning is currently understudied, despite being a critical aspect for a wider adoption. Authors did a great job at introducing their work and rigorously used a mathematical definition for fairness evaluation (Demographic Parity) which is often lacking in this field.
- Could you please explain in more details UFL ?
UFL accounts for the case where clients do not participate to FL. With UFL, every client trains a fair model on its data. However, could you elaborate why considering a random classifier that makes a prediction by randomly selecting a client classifier makes a good baseline?
From a theoretical point of view, I do not see yet the use of UFL nor what the resulting model means. Finally, I do not see what the disparity of UFL is measuring. Giving some context would be very appreciated.
Unfortunately this work should have prioritized its main content over appendix: this work is not clear enough and always requires going to the appendix for full understanding of what the authors are claiming. Therefore, unless UFL is strongly important to the conclusions of this work, Section 3.1 could have been suppressed to focus on other sections instead.

- Could you please elaborate on the statements of Lemma 4? If we consider a setting where clients return the same model, e.g. clients have identical data, wouldn’t FFL via FedAvg and CFL be identical problems while $q_0 \neq q_1$?

- Section 4 would really benefit from having the code of FedFB and explanation of the theoretical work in Section B.1. You state that “it [FedFB] provably converges under some mild technical conditions […] See Theorem 23 for more details”. I have the following questions regarding this theorem:
  - This work seems to rely on Assumption 2-5 for the local loss functions that are standard in Federated Learning and on Assumption 1 “to have a decreasing direction in Lemma 22”. Could you please cite previous work using such an Assumption? Also, you say that with convexity Assumption 1 holds. Could you please prove it or provide references for the proof of that statement?
  - Theorem 23 relies on the work of Li et al. to bound the impact of FedAvg over one iteration. As a result, the obtained bound only depends on the amount of local work $R$. It is standard to consider that the amount of aggregations $T$ goes to infinity but not for the amount of local work $R$. Am I missing something here? Please elaborate or point me to a reference? Especially considering that most work done on FL optimization (including the one of Li et al., or Wang et al. Tackling the Objective Inconsistency Problem in Heterogeneous Federated Optimization) shows that the convergence rate and thus the asymptotic convergence bound are proportional to $R$
  - Can you elaborate on the steps to obtain the bound obtained on $\lambda_a$? It seems to me that the first term in the maximum is always smaller than the left term of the inequality as $\alpha_t$ is positive.
  - Lastly, do you know under which condition we could generalize to A>2?

 - The experiments consider 10 communication rounds and 30 local epochs when considering two clients. Also, the values of some hyperparameters are missing. The experiments like the theory are very specific and do not seem to cover realistic FL cases: could you communicate other parameters lr, batch size, $\epsilon$ ?

 - Also, do you have an explanation why CFL is providing worse fairness than FedFB? The authors speak about extending with their work FB. It would be interesting to compare FedFb with FB where every clients data is centralized on the server.


**Summary Of The Paper:**

The authors propose a novel algorithm to train statistical models respecting fairness criteria like client parity or demographic parity. This problem has been investigated in the literature for the centralized setting and the authors propose to extend FairBatch (FB) to the federated setting. Finally, the authors show experimentally that their algorithm FedFB provides better fairness than when a server centralizes all the clients data. This work is quite novel but its theoretical statements solely rely on very specific use cases and lack clarity.

**Summary Of The Review:**

The authors proposition appears to be novel and with a genuine interest. Authors did a great job at scoping their work and rigorously used a mathematical definition for fairness evaluation.
However, paper structure needs some rework: some keys components, required for the reader to understand the paper, are not in the main content but are in the appendix. Some elements (e.g. UFL) are presented as keys, but their impact on the overall demonstration is not clear and may be moved to appendix. Also, some claims are not fully supported yet.
Finally, experimental cases, and some proofs, are not straightforward to generalize enough mainly due to binary class problems only and federating only two clients.

---

> ### Author Response · Authors · 2021-11-23
> **Thank you for your valuable feedback. All your concerns are addressed.**
>
> We would like to thank you for your thorough reading of our manuscript including the appendix, for your suggestions to improve the writing, and for the detailed review.
>
> --------------------------------------------------------------------------------------------------------------------
>
> Q. The authors' proposition appears to be novel and with a genuine interest.
> A. Thank you so much for appreciating the ideas in our paper.
>
> --------------------------------------------------------------------------------------------------------------------
>
> Q. Why does considering a random classifier make a prediction by randomly selecting a client classifier is a good baseline? Giving some context would be very appreciated.
>
> A. Thanks for a great question. The UFL scheme corresponds to the most private but least performant scheme among all the schemes we considered here, hence playing the role of a good baseline to be studied. We also added Fig. 2 that visualizes the privacy/performance trade-off curve and compares the pros and cons of most of the baselines studied in this paper.
>
> --------------------------------------------------------------------------------------------------------------------
>
> Q. (paraphrased) If clients have identical data, wouldn't FFL via FedAvg and CFL be identical problems? Doesn’t this contradict Lemma 4, which claims that FFL via FedAvg is strictly worse than CFL?
>
> A. The reviewer is correct that FFL via FedAvg becomes identical to CFL when clients have identical data. However, this does not contradict Lemma 4.  It is straightforward to see that for all data distributions, Fairness(FFL via FedAvg) <= Fairness(CFL). What’s unclear is whether there exists any data distribution where the strict inequality holds. What Lemma 4 asserts is that there exist certain “heterogeneous” data distributions such that the strict inequality will hold. We clarified our lemma and added a remark right after the lemma to avoid further confusion.
>
> --------------------------------------------------------------------------------------------------------------------
>
> Q. Section 4 would really benefit from having the code of FedFB.
>
> A. Added. Thanks for your suggestion.
>
> --------------------------------------------------------------------------------------------------------------------
>
> Q. Could you please cite previous work for using assumptions or statements from other work?
>
> A. Fixed. Thanks for your detailed reading. [Please see P29 & P31]
>
> --------------------------------------------------------------------------------------------------------------------
>
> Q. (paraphrased) It is standard to consider that the amount of aggregations $T$ goes to infinity but not for the amount of local work $R$.  Theorem 23 seems to assume that the amount of local work $R$ goes to infinity.  Is that correct?
>
> A. No, Theorem 23 does not assume the infinite amount of local work.  Instead, it follows the standard assumptions that the reviewer mentioned -- finite local work and infinite rounds of aggregations.  We believe that this confusion was due to our choice of notation, so we revised the theorem statement to avoid further confusion.  Please find our revised Theorem statement and let us know if you have any questions.
>
> --------------------------------------------------------------------------------------------------------------------
>
> Q. Could you elaborate on the steps to obtain the bound obtained on $\lambda_a$? It seems to me that the first term in the maximum is always smaller than the left term in the inequality as $\alpha_t$ is positive.
>
> A. Thanks for pointing out this typo. The first term in the maximum should have been $|\lambda_a^{0} - \lambda_a^{\star}|$. We have fixed it now.
>
> --------------------------------------------------------------------------------------------------------------------
>
> Q. Which condition we could generalize Theorem 23 to $A>2$?
>
> A. Unfortunately, our theorem does not generalize beyond the binary setting. The whole analysis becomes highly intractable, and our current proof techniques do not apply anymore when $A>2$.  Generalizing our theory is one of our future directions.  However, since this work is the first theoretical work that characterizes the fundamental gap between various baselines for federated fair learning, we believe that our theory still has enough values and brings new ideas to society.
>
> We remark that our algorithm (FedFB) is readily generalizable to the A>2 settings, and indeed, some of our experimental results are run for those settings (e.g., COMPAS: A = 4, Bank: A = 5).

---

> > ### Author Response · Authors · 2021-11-23
> > **Response (continued)**
> >
> >
> >
> >
> > --------------------------------------------------------------------------------------------------------------------
> >
> > Q. It seems the experiments only consider two clients.
> >
> > A. We considered more than two clients in our experiments. For instance, in Table 6, we studied the performance of FedFB in the synthetic dataset with different numbers of clients. Please let us know if there were any misleading statements in our paper -- we are happy to revise them in our final version.
> >
> > --------------------------------------------------------------------------------------------------------------------
> >
> > Q. The values of some hyperparameters are missing.
> >
> > A. Thanks for pointing it out. We added the details of hyperparameter tuning in Appendix C.1 (P38).
> >
> > --------------------------------------------------------------------------------------------------------------------
> >
> > Q. Could you communicate other parameters lr, batch size, $\epsilon$?
> >
> > A. We assume hyperparameters are only exchanged once at the beginning of the training phase.
> >
> > --------------------------------------------------------------------------------------------------------------------
> >
> > Q. Do you have an explanation why CFL is providing worse fairness than FedFB.
> >
> > A. Thank you for pointing it out. It was because (1) tables only showed one operating point on the trade-off curve, and (2) our specific cross-validation algorithm for hyperparameter selection was highly sensitive to randomness.  To avoid further confusion on CFL, we updated Table 1, 2 and added Figure 5 which depicts the complete trade-off curves with carefully chosen hyperparameters.
> >
> > --------------------------------------------------------------------------------------------------------------------
> >
> > Q. It would be interesting to compare FedFB with FB where every clients’ data is centralized on the server.
> >
> > A. Our current CFL scheme is exactly what the reviewer suggested -- it runs FB on every clients’ data in a centralized manner.
> >
> > --------------------------------------------------------------------------------------------------------------------
> >
> > Final Notes: We want to thank you for such exemplary and detailed comments. Given your understanding of the novelty and significance of our work and that we have addressed your major concerns, it would be very helpful if you are willing to increase your score and support accepting our paper.
> >
> > References:
> >
> > [1] Yue, X., Nouiehed, M., & Kontar, R. A. (2021). Gifair-fl: An approach for group and individual fairness in federated learning. arXiv preprint arXiv:2108.02741.
> >
> > [2] Li, T., Hu, S., Beirami, A., & Smith, V. (2021, July). Ditto: Fair and robust federated learning through personalization. In International Conference on Machine Learning (pp. 6357-6368). PMLR.
> >
> > [3] Li, T., Sanjabi, M., Beirami, A., & Smith, V. (2019). Fair resource allocation in federated learning. arXiv preprint arXiv:1905.10497.

---

> > > ### Comment · Reviewer_q1zx · 2021-11-29
> > > **Final feedback**
> > >
> > > I want to thank the authors for their answers. Added references are relevant for this work.
> > > UFL relevancy is still not justified enough for being considered as a baseline.
> > > Also, I am still confused by Theorem 23 despite the rewriting. Indeed, Li et al. prove convergence for a whole FL process, and I do not see how it can be used when bounded between two communication rounds. Using three aggregation notations ($R$, $T$, $E$) is confusing.
> > >
> > > The authors added a statement as part of their last update:
> > > "Our setting is more appropriate in domains such as criminal justice and social welfare", this applicability of the author's proposal may raise some ethical concerns and should be supported.
> > > Experimental scenarios are still not fully described preventing the complete understanding and reproducibility needed for validating their work.
> > > As my key concerns remain, therefore I will not change my score.
> > > I encourage authors to update their work to improve clarity regarding both theoretical (mainly Theorem 23) and experimental descriptions.

---

> > > > ### Author Response · Authors · 2021-12-09
> > > > **Thanks for your feedback.**
> > > >
> > > > Dear Reviewer:
> > > >
> > > > Thank you for the questions and suggestions.
> > > >
> > > > ----------------------------------------------------------------------------
> > > > Q. Theorem 23 is still confusing.
> > > >
> > > > A. Thanks for your feedback. We will take your advice and make it more clear.
> > > >
> > > > ----------------------------------------------------------------------------
> > > > Q. Concerns related to ethics and experiments.
> > > >
> > > > A. We will describe the potential ethical concerns related to these applications, and specify more experiment details.

---

### Official Review · Reviewer_V46Y · 2021-11-02

**Correctness:** 3
**Technical Novelty And Significance:** 2
**Empirical Novelty And Significance:** 2
**Recommendation:** 3
**Confidence:** 4

**Main Review:**

This paper provides a novel analysis on fair training in three different settings. I'm concerned about part of this paper's main theoretically claim that UFL provides weaker guarantee then FFL via FedAvg. It is unclear to me why it is necessary to compute the \hat{y}|x,a across all clients instead of evaluating Demographic Parity just separately on individual clients. Basically in this scenario local clients are not collaborating and they are just training separate global model. In fact, client 0 and client 1 even use different models to produce \hat{y}. Why does studying the joint distribution of \hat{y}|x,a matters here?

It is also not clear how do the authors optimize the FairBatch objective in the FL setting. I assume the step of solving the optimal global to follow the standard FedAvg algorithm. It is not clear, however, how the group-specific loss is calculated. From Figure 4 it seems that the authors are summing over that loss over all the clients? Could you explain on this part?

This paper also compares the empirical results of FedFB with other fair FL training methods and demonstrates an advantage of using FedFB over other methods for fairness. However, the results of several baselines in this paper look confusing to me. In the section of client parity, the qffl accuracy is surprisingly low for adult, which does not match up with the results in the original qffl paper. It is also not clear to me why FedFB outperforms CFL in some cases. Could you explain why this happens?


**Summary Of The Paper:**

This paper compares and analyzes the fairness guarantee(Demographic Parity in particular) provided in three different settings: training local models, training FedAvg with local fair training, training fair global model. This paper provides a method on how to enforce group fairness into federated learning by applying a current fair training method, FairBatch, into FL.

**Summary Of The Review:**

This paper focuses on fairness in FL from an interesting angle. However, the current comparison between UFL and FFL via FedAvg doesn't make sense to me. More explanations are also needed for both algorithm and empirical results.

---

> ### Author Response · Authors · 2021-11-23
> **Thank you for your detailed feedback. We have addressed all your comments!**
>
> First, we thank the reviewer for the detailed review and very helpful suggestions.
>
> --------------------------------------------------------------------------------------------------------------------
>
> Q. This paper focuses on fairness in FL from an interesting angle.
>
> A. We thank the reviewer for appreciating the ideas in our paper.
>
> --------------------------------------------------------------------------------------------------------------------
>
> Q. Why does studying the global fairness (joint distribution of $\hat{\mathrm{y}} \mid \mathrm{x,a}$) matter here?
>
> A. Thank you for asking a great question.  Indeed, there are two different settings studied in the literature.  The first one is the setting we considered -- we want to find a classifier that satisfies “the global constraint” and performs well on the overall data distribution.  The same setting was employed in [1--3].  The second one, which is less studied in the literature, is the setting where one wants to find a classifier that satisfies each of the local constraints, i.e., to find a classifier that is local on each client’s dataset.  This setting was recently studied in [4].
>
> The two settings have different purposes.  The former is useful when one wants to find a single classifier that can be applied to the entire population as a whole, while the other is useful when one needs to satisfy local constraints for some other reasons, say legal constraints.  In our revised paper, we added this discussion together with more specific examples where our problem setting is more relevant.
>
> --------------------------------------------------------------------------------------------------------------------
>
> Q. It is not clear how the group-specific loss is calculated.
>
> A. Each client computes the total loss specific to each group, i.e., $\sum_{j: \mathrm{y}_j = y, \mathrm{a}_j = a} \ell(\mathrm{y}_j, \hat{\mathrm{y}}_j)$. To avoid further confusion, we added more details about our FedFB algorithm to the main body, including the aforementioned definition.
>
> --------------------------------------------------------------------------------------------------------------------
>
> Q. q-FFL accuracy is surprisingly low for Adult, which does not match up with the results in the original q-FFL paper.
>
> Q. It is not clear why FedFB outperforms CFL in some cases.
>
> A. Thank you for pointing these out. It was because (1) our original tables only showed one operating point on the trade-off curve, and (2) our specific cross-validation algorithm for hyperparameter selection was highly sensitive to randomness.  To avoid further confusion on q-FFL and other algorithms for Client Parity, we replaced Table 3 with Figure 6 which depicts the complete trade-off curves with carefully chosen hyperparameters.  To do the same for FedFB vs CFL, we added Figure 5 to provide a more clear comparison.
>
> In the new figures, one can see that (1) our algorithm closely matches the CP-accuracy trade-offs of the existing CP algorithms but does *not* necessarily outperform them, and (2) CFL provides a strict upper bound on the performance of FedFB. We hope this resolved the reviewer’s concern.
>
> --------------------------------------------------------------------------------------------------------------------
>
> Final Notes: We want to thank you for such helpful comments. Given your understanding of the novelty of our work and that we have addressed your major concerns, we will highly appreciate it if you can consider increasing your score and support accepting our paper.
>
> References:
>
> [1] Rodríguez-Gálvez, B., Granqvist, F., van Dalen, R., & Seigel, M. (2021). Enforcing fairness in private federated learning via the modified method of differential multipliers. arXiv preprint arXiv:2109.08604.
>
> [2] Ezzeldin, Y. H., Yan, S., He, C., Ferrara, E., & Avestimehr, S. (2021). FairFed: Enabling Group Fairness in Federated Learning. arXiv preprint arXiv:2110.00857.
>
> [3] Chu, L., Wang, L., Dong, Y., Pei, J., Zhou, Z., & Zhang, Y. (2021). FedFair: Training Fair Models In Cross-Silo Federated Learning. arXiv preprint arXiv:2109.05662.
>
> [4] Cui, S., Pan, W., Liang, J., Zhang, C., & Wang, F. (2021, May). Addressing Algorithmic Disparity and Performance Inconsistency in Federated Learning. In Thirty-Fifth Conference on Neural Information Processing Systems.

---

> > ### Comment · Reviewer_V46Y · 2021-11-30
> > **Still have concerns regarding the joint distribution**
> >
> > Dear authors:
> >
> > Thanks for your detailed response. I'm still not convinced by what you said about why global fairness matters in training local models. I understand the difference between evaluating fairness globally and evaluating fairness locally. However, as suggested by yourself, the former is useful when one wants to find a **single** classifier, which is not the case in training separate local models. Therefore, I still feel that it is not fair to evaluate global fairness for separate local models. (I do think measuring global fairness for CFL, FFL via FedAvg makes sense.) Could you further explain why you perform the comparison here?

---

> > > ### Author Response · Authors · 2021-12-06
> > > **Motivation for UFL**
> > >
> > > Dear reviewer:
> > >
> > > Thank you for the questions! One of the main goals of our paper is to show that federated learning is necessary for improving fairness. Therefore, we need one baseline method which may learn fair classifiers from decentralized data privately without incorporating federated learning. As a result, we proposed UFL. Actually, UFL is equivalent to assembling the local classifiers to one single global classifier (f = f1/2 + f2/2).
> > >
> > > Thanks for letting us know the confusion here. We will certainly add the discussion in the final revision.

---

### Public Comment · ~Sen_Cui1 · 2021-11-20
**A nice work on fairness in federated learning**

This work studies the fairness issues in federated learning. There is a very relevant reference that is missed. "Addressing Algorithmic Disparity and Performance Inconsistency in Federated Learning" with arxiv link https://arxiv.org/pdf/2108.08435.pdf

***

Many thanks for you. After careful learning, I'm sure this is a very valuable study, especially its theoretical contribution. It emphasizes that decentralization itself does not exacerbate unfairness.  Though it was unfortunately rejected this time, it has already given us a deeper understanding of fairness in federated learning.

---

> ### Author Response · Authors · 2021-11-23
> **Response**
>
> Hi Sen,
>
> Thank you for bringing your interesting work to our attention. We found that this work proposed an interesting solution to a highly related but slightly different problem -- the goal of your paper is to find a classifier that satisfies each of the local constraints, while our goal is to find a classifier that satisfies the global constraint, which is the aggregated version of the local constraints. We added your work to the related work section of our revised paper.
>
>
> Best regards,
>
> Authors

---

### Decision · Program_Chairs · 2022-01-20

**Decision:**

Reject

**Comment:**

The authors consider the problem of training a fair classifier on decentralized data, and compare three methods: training locally, training the proposed FedAvg algorithm with local fairness, and a global fairness approach.

The reviewers agreed that the setting was interesting and novel, but had concerns about the writing quality, experimental setup, and, most importantly, the organization of the paper, with several reviewers complaining that necessary information was relegated to the appendix.

Overall, this work is not quite ready for publication. With that said, the reviewers agreed that it was interesting and highly promising (it just needs refinement). Please seriously consider the reviewers' recommendations, which on the whole were very constructive and, if followed, should lead to a significant improvement in your manuscript.